**Extreme flood events reconstruction spanning the last century in the El Bibane lagoon**
**(Southeast of Tunisia): a multi-proxy approach**
**A. Affouri[a,b], L. Dezileau[b] and N. Kallel[a]**
a : Laboratoire Georessources, Matériaux, Environnements et changements globaux,
LR13ES23 (GEOGLOB), Faculté des Sciences de Sfax, BP1171, Sfax 3000, Université de
Sfax, Tunisie.
b : Geosciences Montpellier, CNRS/INSU, UMR 5243, Université Montpellier, Montpellier,
France.
*Corresponding authors:* **aidaemna@yahoo.fr** (*A. Affouri*) and **dezileau@gm.univ-**
**montp2.fr** (*L. Dezileau*)
**Abstract**
Climate models project that rising atmospheric carbon dioxide concentrations will increase
the frequency and the severity of some extreme weather events. The flood events represent a
major risk for populations and infrastructures settled on coastal lowlands. Recent studies of
lagoon sediments have enhanced our knowledge on extreme hydrological events such as
paleo-storms and on their relation with climate change over the last millennium. However few
studies have been undertaken to reconstruct past flood events from lagoon sediments. Here,
the past flood activity was investigated using a multi-proxy approach combining
sedimentological and geochemical analysis of surfaces sediments from the Southeast of
Tunisia catchment in order to trace the origin of sediment deposits in the El Bibane lagoon.
Three sediment sources were identified: marine, fluvial, and aeolian. When applying this
multi-proxy approach on the core BL12-10, recovered from the El Bibane lagoon, we can see
that finer material, high content of the clay and silt, and high content of the elemental ratios
(Fe/Ca and Ti/Ca) characterize the sedimentological signature of the paleoflood levels
identified in the lagoonal sequence. For the last century which is the period covered by the

BL12-10 short core, three paleo-flood events were identified. The age of these flood events have been determined by $^{210}$Pb and $^{137}$Cs chronology and give age of AD 1995 ± 6, AD 1970 ± 9 and AD 1945 ± 9. These results show a good temporal correlation with historical flood events recorded in the Southern of Tunisia in the last century (A.D 1932, A.D 1969, A.D 1979 and A.D 1995). Our finding suggests that reconstruction of the history of the hydrological extreme events during the upper Holocene is possible in this location, by the use of the sedimentary archives.

**Keywords:** El Bibane Lagoon; watershed basin; surface sediments; geochemistry; grain size; paleo-floods, upper Holocene, Southeast Tunisia.

## 1. Introduction

The Mediterranean region has experienced numerous extreme coastal events, such as flood events which caused casualties and economic damages (Lionello et al., 2006). However, the meteorological instrumental records are limited to only a few decades, especially in Southern Mediterranean countries. Geological data offer a way to reconstruct the historical records of intense flood events. Deciphering records of extreme precipitation and damaging floods preserved in geologic archives enables society to understand and plan for floods in the future (Parris et al., 2010). The importance of studying trees, river and lake sediments has already been shown for reconstructing extreme flooding events (Baker, 1989; Ely et al., 1993; Brown et al., 2000; Benito et al., 2003; Wolfe et al., 2006; Moreno et al., 2008; Wilhelm et al., 2012; St. George and Nielsen, 2003; Gilli et al., 2013). Few studies have been undertaken to reconstruct past flood events from lagoon sediments (Raji, 2014). Most of the studies were interested to flooding associated with both hurricanes and tsunamis where overwash deposits are preserved within back-barrier lagoons and salt ponds can provide a mean for documenting previous flooding activity (Liu & Fearn, 1993; Donnelly and Woodruff, 2007; Sabatier et al., 2008; Dezileau et al., 2011, 2016; Raji et al., 2015; Degeai et al., 2015). Heavy rain flooding

events recorded within these lagoon environments are still poorly documented. Moreover, reconstruction of past flood events from sedimentary archives has been poorly studied in Tunisia. Some fluvial archives have been used to reconstruct past flood events in the northern part of Tunisia (Zielhofer et al. 2004; Zielhofer and Faust; 2008) but not in the southern part. In this study we tried to reveal the importance of lagoonal archives to reconstruct past flood activities under a semi-arid environment in southern part of Tunisia, studying the paleo-floods from high resolution geochemical and sedimentogical analyses. The first aim of this study was to identify the different sediment sources and to retrace the marine, the fluvial and the aeolian contributions to the sedimentation in the El Bibane Lagoon. The second aim was to reconstruct flood events from the lagoonal archives during the last century. To reach these objectives, we undertook the calibration of the sedimentological and geochemical proxy data with historical flood records.

## 2. Study site: El Bibane Lagoon and its watershed

Morphologically, Southern Tunisia known as the Tunisian platform includes two distinguished morpho-tectonic domains (Fig. 1) namely: The Djeffara (Inner domain) and the Dahar (Outer domain). The Djeffara extends over all the coastal plain from Gabes (Southeastern Tunisia) to the Libyan borders. It is limited to the west by the Matmata and the Dahar mountains and to the east by the Gulf of Gabes and the Mediterranean Sea. The Dahar belongs to the Saharan platform domain and is constituted by successions sequences ranging in age from the Late Permian to the Late Cretaceous (Fig. 1). The lithostratigraphic successions could be summarized as follows: The Early–Middle Triassic sequence in the Dahar plateau is mainly constituted by continental sandstone, conglomerate and clay; whereas the Late Triassic outcrops exhibit shallow marine carbonate (Busson, 1967). The Jurassic series are represented by a thick Liassic evaporitic sequence, Dogger marine carbonate and late Jurassic–Neocomian mixed facies with continental predominance (Bouaziz et al., 2002).

The Cretaceous series represents a general succession from neritic, lagoonal and continental
facies (Mejri et al., 2006). The Late Cretaceous is characterized by thick shallow marine
carbonates-marl sequences and covered by sand dunes of the Eastern Saharan Erg.
The Mio-Pliocene series represent the substratum of the coastal plain of Djeffara. Jedoui
et al. (1998) subdivided these series into two principal facies: (1) the red coloured clays rich
in gypsum and (2) the sands which locally associated with conglomerates and grey clays. The
Pleistocene marine deposits of the Southeast Tunisian coastal zone assigned to the
"Tyrrhenian" (marine isotopic stage 5e) overly unconformably the Mio-Pliocene. These
deposits form a ridge parallel to the actual coast. They show the superposition of two units
described by Jedoui et al. (2002) as the lower "quartz-rich unit" and the upper "carbonate
unit" with *Strombus bubonius*.
The study area is focused on the El Bibane Lagoon and its watershed (El Bibane Lagoon:
33° 15' 01"N-11° 15' 41"E; Fig. 1). This lagoon which has an elongated elliptic form (33 x10
km) and a major WNW-ESE axis covers an area of about 230 $Km^2$. It has a maximum water
depth of 6m in the middle part of the basin (Guélorget et al., 1982; Medhioub, 1984). The
Eastern periphery of the EBL is partially separated from the Mediterranean Sea (Gulf of
Gabes) by two peninsulas namely El Gharbi (western) and Ech Chargui (eastern), each of
about twelve kilometres long (Medhioub, 1979). These two peninsulas, called slobs, are cut at
their mid-part by nine small islets and channels: the zone of connection with the
Mediterranean waters (Medhioub & Perthuisot, 1981). The two slobs are represented by
emerged Tyrrhenian aeolian littoral dunes and carbonate sand beach (Jedoui, 2000; Jedoui et
al., 2002). The El Bibane Lagoon has a microtidal regime where tidal amplitude varies from
0.8 to 1.5 m (Davaud and Septfontaine, 1995; Sammari et al., 2006). The intertidal flats are
flooded and exposed daily at regular intervals during the periodically rising and retreating
tide. Supratidal flats are flooded at irregular intervals during spring tides or strong onshore

winds (Bouougri & Porada, 2012). The El Bibane lagoon is relatively unaffected by human activities (Pilkey, 1989; Ounalli, 2001) where it is only exploited by traditional fisheries (Guélorget et al., 1982).

**3.   Climate and hydrology**

The southeastern Tunisia region is characterized by a pre-Saharan and arid to semi-arid climate. The hot season extends beyond the summer (Amari 1984; Ferchichi, 1996; Hamza, 2003) and the number of sunny days may reach 64.4%. The rainfall is low with an annual average that does not exceed 200 mm (Hamza, 2003). Furthermore, rainfall is very fluctuating with high inter-annual variability and intensity. Most of the rainfall is concentrated within 30 days/year (Genin and Sghaier, 2003) leading to high fluctuations in water discharge. The highest precipitation occurs mainly in October to March while in the summer months there are drought conditions.

The annual precipitations of Medenine and Tataouine stations during the last century were obtained from the Tunisian General Administration of Water Resources (DGRE, 2010, Fig.2). Five major enhanced precipitation events were recorded from these two stations (i.e. A.D 1932, A.D 1969, A.D 1979, A.D 1984 and A.D 1995). These pluvial episodes have induced large flood events in the Fessi River watershed (Poncet, 1970; Bonvallot, 1979; Oueslati, 1999; Boujarra and Ktita 2009; Fehri, 2014).

**4.  Materials and Methods**

**4.1. Materials**

Eighteen surface sediment samples were collected from the watershed (Jerba, Zarzis, Medenine, Tataouine and Ben Guerdane localities) in order to assess the origin of the material transported into lagoon (Fig. 3). The location of all sampling stations was recorded by GPS (GPSmap 60, Garmin, Table 1). The main potential sediment sources were sampled in order to characterize their sedimentological and chemical signatures as follows:

-    three samples from the beach area (S1, S2 and S3) representing the marine source,
-    ten samples (S7 to S16) from Fessi River catchment representing the fluvial/river

sources,

-    two dune samples (S17 and S18) representing the eolian component.
-    three surface samples (S4 to S6) from El Bibane lagoon have been selected to

represent the present-day sedimentation. The S6 representing the first three

centimeters of a lagoon sediment core BL12-10 was used to characterize the surface

sediments samples.

Moreover, to reconstruct recent flood events occurred in the studied area, a short
sediment core (BL12-10, 40 cm length; Latitude: 33°14'58.7"; Longitude: 11°10'3.7" Fig.3)
was recovered from the El Bibane Lagoon (EBL) by a hand corer 75mm diameter PVC tube
in the southern part of the lagoon, at 35 km from the Fessi River delta and 14 Km from the
connection with the sea.
**4.2. Analytical methods**
**4.2.1. Sedimentological and geochemical analysis**
The BL12-10 core was first split, photographed and logged in detail. Elemental
geochemical analyses by energy-dispersive X-ray fluorescence spectrometry were undertaken
with a hand-held Niton XL3t. Measurements were realized on the watershed surface samples
and each 2 cm along the BL12-10 core. BL12-10 core and surface samples had been covered
with a 4µm thin Ultralene film to avoid contamination of the XRF measurement unit and the
desiccation of the sediment (Richter et al., 2006). The elemental analyses from XRF
measurement were performed in mining type ModCF prolene mode. These data show directly
concentrations in ppm or percentage values. This is a semi-quantitative measurement.
International powder standards (NIST2702 and NIST2781) were used to assess the analytical
error and accuracy of measurement, which are lower than 5% for Ti, Cr, Fe, Zn, Pb, between

5 and 15% for Ca, Mn, As, Rb, Sr, and between ca. 15 and 25% for K and Co.

Laser grain-size analyses were achieved with a Beckmann-Coulter LS13320 Particle Size Analyser (Geosciences Montpellier). Grain-size analyses were performed on surface samples and on the BL12-10 sequence with an average interval of 1 cm. Each sample was sieved through a 1 mm mesh, suspended in deionised water and gently shaken to achieve disaggregation. Ultrasound was used to avoid particles flocculation of sediment in the fluid module of the granulometer. For each sample, a small homogeneous amount of sediment was mixed in deionized water, then sieved at 1.5 mm diameter before pouring in the Fluid Module of the Particle Sizer until to obtain an optimal obscuration rate between 7 and 12% in the Fraunhofer optical cell. The time of background and sample measurement was set to 90 s and sonication was applied during the measurement of the sample in order to improve the dispersion of fine particles in the fluid. Each sample was measured twice and the good repeatability of measurement was verified according to the statistics from the international standard ISO 13320-1.

GRADISTAT program version 4.0 (Blott, 2000) was used for grain size statistical analysis. The following sample statistics are calculated using the Method of Moments in Microsoft Visual Basic programming language: mean, mode(s), sorting (standard deviation), skewness and kurtosis. Grain size parameters are calculated arithmetically, geometrically (in microns) and logarithmically (using the phi scale) (Krumbein and Pettijohn, 1938). Linear interpolation is also used to calculate statistical parameters by the Folk and Ward (1957) graphical method and derive physical descriptions (such as "very coarse sand" and "moderately sorted").

Finally, the percentage of the granulometric classes <2µm, 2-63µm and 63-2000µm, which stand for clay, silt and sand fractions, respectively, were calculated.

**4.2.2. BL12-10 core dating**

Dating of sedimentary layers was carried out using $^{210}$Pb and $^{137}$Cs methods on a
centennial timescale. The $^{137}$Cs and $^{210}$Pb$_{ex}$ activities analyses were performed on the fraction
< 150μm by gamma spectrometry using a CANBERRA Broad Energy Ge (BEGe) detector
(CANBERRA BEGe 3825). The sediment was then finely crushed after drying, and
transferred into small tubes (diameter 14 mm), and stored for more than 3 weeks to ensure
equilibrium between $^{226}$Ra and $^{222}$Rn. Generally, counting times of 24 to 48 h were required to
reach a statistical error of less than 10% for $^{210}$Pb$_{ex}$ in the deepest samples and for the 1963
$^{137}$Cs peak. Activities of $^{210}$Pb were determined by integrating the area of the 46.5-keV photo-
peak. $^{226}$Ra activities were determined from the average of values derived from the 186.2-keV
peak of $^{226}$Ra and the peaks of its progeny in secular equilibrium with $^{214}$Pb (295 and 352
keV) and $^{214}$Bi (609 keV). In each sample, the ($^{210}$Pb unsupported)$_{ex}$ activities were calculated
by subtracting the ($^{226}$Ra supported) activity from the total ($^{210}$Pb) activity. We then used the
Constant Flux/Constant Sedimentation (CFCS) model and the decrease in $^{210}$Pb$_{ex}$ to calculate
the sedimentation rate (Goldberg, 1963). The uncertainty of the sedimentation rate obtained
by this method was derived from the standard error of the linear regression of the CFCS
model.
$^{137}$Cs was studied on the core BL12- 10 in order to assess sediment accumulation rates
and chronology of the first 30 centimetres of the core. $^{137}$Cs (t1/2 = 30.1 yr) is an
anthropogenic radionuclide. It entered the environment in response to atmospheric nuclear
tests from 1954 to 1980 AD that induced global fallouts (the first year of atmospheric releases
was 1953 AD, whereas the maximum atmospheric production is reached in 1963 AD. $^{137}$Cs
depth profiles have been extensively used in various environments to assess sediment
accumulation rates (Nittrouer et al., 1984; He and Walling, 1996; Radakovitch et al., 1999;
Frignani et al., 2004).
**4.2.3 Statistical analyses**
Statistical methods were applied to complete and refine the analysis. Principal
Component Analysis (PCA) is widely used statistical techniques in environmental
geochemistry. This multivariate approaches is used to reduce the large number of variable that
result from XRF analysis. Principal Component Analysis (PCA) was applied to elements in
order to distinguish the different sediment sources of surface sediments and link them to the
geochemical processes or proprieties. In the present work, the dataset contains 18 samples,
each of which includes concentration of 8 elements (Ca, Sr, Fe, K, Al, Ti, Si and Zr). Data are
presented in the form of elemental concentration (8 variables). In this study, a statistical
analysis was performed using the STATITCF (1987) which is based on variables and it is
suitable for identifying the associations of variables with a set of observations. A
representation quality of the parameters (positions in the factorial plane) was then performed.
**5. Results**
**5.1. Surface sediments**
**5.1.1. Sediment description:  grain size and morphology**
Grain size analysis and binocular observation of the surface sediment samples have
permitted to characterize three groups of sediments as follows, depending on the
environmental setting: Marine, Fluvial and Aeolian sources (Fig. 4 and 5). The first group
encompasses sediment samples (S1, S2 and S3) collected along the coastal zone from Jerba to
Zarzis beaches and the lido of El Bibane Lagoon. In this marine area, surface sediments are
composed of a mixture of coarse sub-rounded quartz grains, mollusc shells and foraminifera
(Fig. 4). The grain size analysis (Table 2) of samples S1 and S2 show unimodal distributions
in 169µm and 203µm, respectively indicating moderately sorted fine sand sediments (Folk,
1954; Folk and Ward, 1957; Fig. 5).  The sample S3 is muddy sand namely very coarse silty
to coarse sand sediment with unimodal distribution in 518µm.
The second group of samples (S7, S8, S9, S10, S11, S12, S13, S14, S15 and S16) came
from the El Bibane delta and the Fessi River. It is assigned as the fluvial source. Binocular
observations of the samples reveal reddish-brown heterogeneous particles composed mainly
of shiny angular to sub angular quartz grains. Some grains display rust colour with iron oxide
(Fig. 4). Figure 5 displays that the fluvial source has a unimodal to multimodal distribution
with two or three modes. In order to obtain the best resolution in the identification of the
fluvial source, we choose to use the sediment samples which were collected only along the
River Fessi: S9, S10, S12 and S13. These surface sediment samples show a decrease in the
mean grain size from upstream to downstream of the River Fessi watershed (Fig. 6). The
decrease in the mean grain size could be explained by a strong change of the topographic
slope around Tataouine (located at approximately 85 km from the lagoon). Here, the coarser
material is deposited and the finer material is transported further by the river. These finer
sediments are deposited in the low plain of the river and in the El Bibane lagoon. Therefore,
we suggest that S9 and S10 (collected between Tataouine and the lagoon) characterize the
fluvial component in the lagoon. The grain size distribution for S9 is unimodal with a mean
grain size around 96 µm indicating a moderately sorted muddy sand. The corresponding size
range very coarse silty/very fine sand. Sample S10 is fine silt with trimodal distribution in
7µm, 26µm and 73µm, and poorly sorted mud sediment type. These characteristics will serve
to identify the fluvial source into the lagoon.
The third group consists of two samples (S17 and S18) recovered in the Aeolian sand
dunes of southern Tunisia. They are composed of homogenous dark yellow sand with angular
grains; some of them are coated by iron oxide (Fig. 4). Unimodal distribution in 116µm
(Table 2) characterizes the aeolien samples S17 and S18. These samples are well (S18) to
very well sorted (S17) and correspond to very fine sand. The characteristics of this group will
serve to identify the aeolian sand dune source.
The El Bibane Lagoon surface sediments samples S4, S5 and S6 were characterized by
multimodal grain size distribution (Table 2, Fig. 5). The grain size distribution of sample S4
shows very poorly sorted sandy mud with trimodal distribution at 154µm, 96µm and 31µm,
which indicates a very fine sand/very coarse silt. The sample S5 is very coarse silty/very fine
sand sediment, with a bimodal distribution in 106µm and 429µm, poorly sorted muddy sand.
The sample S6 is unimodal, with a mode in 116µm. It is moderately sorted very coarse
silty/fine sand sediment with a muddy sand texture (Folk, 1954; Folk and Ward, 1957).
**5.1.2. Distribution of major and trace elements**
The spatial distribution of major and trace elements in surface sediments collected in the
El Bibane lagoon and in all the area mainly along the Fessi River are displayed in figure 7.
The iron (Fe) shows its highest percentages in the Fessi River samples (0.53-1.52%).
Lower values characterise the aeolian dunes (0.38-0.4%) whereas this element is totally
absent in marine sediments (Table 3). The same distribution pattern is also observed for Ti, K
and Al. The highest contents of these elements in the Fessi River samples contrast with the
lowest ones retrieved in the marine surface sediment. Aeolian dunes are characterised by
intermediate values. These four elements will thus be used as indicators of terrigenous input
of material to the lagoon.
Calcium (Ca) and Strontium (Sr) in the sediment are usually associated to the carbonate
fraction, which can be either of allochtonous or autochtonous origin. In the sediments,
carbonates are mainly of biogenic origin. In fact, due to its compatible ionic radius, Sr can
replace Ca in calcite, but remains however as trace element (Fig.7). Nevertheless, both
elements show the same distribution pattern. Marine surface sediments are associated with the
highest values (Ca ≈ 14, 7%; Sr ≈ 1548 ppm) whereas the lowest values and thus the lowest
calcite contents are retrieved in dune samples (Ca ≈ 0.8%; Sr ≈ 52 ppm). Intermediate
concentrations are associated with the Fessi River catchment (Ca ≈ 7%; Sr ≈ 150 ppm) (Table

3).

Silicon (Si) and Zircon (Zr) follow similar spatial distribution pattern (Fig. 7). Higher

content of these elements are observed in the River catchment samples (Si ≈20 %; Zr ≈ 300
ppm) and in the aeolian dune samples (Si ≈33%; Zr ≈ 400 ppm), whereas marine sediments
show generally lower contents (Si ≈ 10%; Zr ≈ 41 ppm) (Table 3).
**5.1.3. Principal component analysis (PCA)**

We used Principal Component Analysis (PCA) to identify the main factors controlling

the chemical composition of the catchment and El Bibane lagoon surface sediments and to
identify different groups of common origin and process. Application of Principal Component
Analysis (PCA) varimax rotation has permitted to identify two components that explained
83% of the total variance (Fig. 8). Factor 1 account for 64.46% of total variance. It is
characterized by high positive loadings for Fe, Ti, K, and Al which indicates the dominance
of alumino-silicates minerals in surface sediments (Spagnoli et al., 2008; Plewa et al., 2012).
These elements are prevailing in the river surface samples and their granulometric
distributions display that their grain sizes are in the range of clay and silt. Zr and Si display a
moderate positive loading in factor 1 and are high in the Aeolian surface sediments. Zr and Si
are associated to silicates originating either from adjacent desert areas by erosion or from
western Saharan dunes by storms.

Factor 2 account for 17.73% of the total variance (Fig. 8). It shows positive loading for

Ca, Sr, Fe and K, whereas Ti, Al, Zr and Si have negative loadings. Ca is high in the marine
samples. The high percentage of Ca in these samples is related to both the significant presence
of biogenic material and also probably the precipitation of authigenic carbonate. These results
corroborate the marine origin of these sediments as revealed by the binocular observations
mainly due to the existence of shell debris and confirmed by the grain size distributions.
Therefore, we suggested that the first component agreed with the fine fraction of the
sediment, which is mainly composed of various types of clay minerals, usually abundant in
surface sediments (De Lazzari et al., 2004). On the other hand, factor 2 (Fig. 8) provides a
better definition of the relatively carbonate fraction of the sediments. Consequently, these two
factors differentiated carbonates from both sand and clay sediments. This method allowed us
to label elements of terrigeneous source (Fe, Ti, K and Al) from those from in situ marine
origin (Ca and Sr). These proxies will be used to reconstruct past flood and storm events with
the help of sedimentary archives.
**5.1.4. El Bibane lagoon: Main sediment sources**
Geochemical parameters as well as grain size data are useful indicators for the
detection of significant facies changes in the stratigraphical record (Vött et al., 2002, Zhu &
Weindorf, 2009). Statistical analyses of geochemical data have permitted to characterise the
different sediment sources around El Bibane lagoon. Ca, Ti and Fe elements have been
chosen in order to recognize the contribution of these sources to the surface sediments of the
Lagoon. Ca displays its highest abundances in marine area and is lower in sand dunes and
river samples. By contrast, Ti characterises the continental source (see section 5.1.2) and
shows low contents in marine samples. On the other hand, Fe is present as a maximum in the
river samples and as a trace element in marine samples. Taking into account this geographic
distribution, Fe/Ca as well as Ti/Ca ratios values would be higher in the continental supply
(fluvial and aeolian samples) and lower in the marine source. High Fe/Ca values due to high
iron content may also reflect dominating subaerial weathering and oxidation. The Fe/Ca and
Ti/Ca ratio values and the position on a Fe/Ca vs. Ti/Ca diagram (Fig. 9) of El Bibane Lagoon
surface sediments (samples S4, S5 and S6) are intermediate between the marine and fluvial
source.  Accordingly, higher Fe/Ca and Ti/Ca ratio in the lagoon sediments would be a signal
of more sediment contribution from fluvial source to the lagoon during flooding. As shown

before, the Fessi River sediments were characterized by fine material with a grain size which

does not exceed 63 µm (case of S9 and S10) (See Chap.5.1.1, page 10).

**5.2 Core BL12-10**

**5.2.1. $^{210}$Pb and $^{137}$Cs dating**

The measured $^{210}$Pb values in the uppermost 30 cm of the BL12-10 core range from

14.5 to 0.1 mBq /g (Table 4). In general, the down core distribution of $^{210}$Pb$_{ex}$ values follows a

relatively exponential decrease with depth and the "Constant flux: Constant Supply" (CF/CS)

sedimentation model was applied. The calculated sedimentation rate (SR) is about 0.48 cm/

year. The down core $^{137}$Cs activity profile (Fig. 10) shows a maximum at 18 cm depth (Table

4). We attributed this maximum to the period of maximum radionuclide fallout in the

Northern Hemisphere associated with the peak of atomic weapons testing in 1963. The $^{137}$Cs-

derived SR (0.37 cm/ year) is lower than that of the $^{210}$Pb (Fig. 10). The difference between

the two methods could be explained by a change of the accumulation rate between the

beginning and the last part of the 20$^{th}$ century.

**5.2.2 Sedimentary and geochemistry**

The sediment sequence from El Bibane lagoon presented in this study come from the

core BL12-10 recovered in the nearest part of the delta of Fessi River in May 2012. This

study proposes the preliminary analyses performed on the first 30 cm only although the whole

BL12-10 core length is 90 cm. The BL12-10 core is composed of coarse-grained layers of

siliciclastic sand and shell fragments inter-bedded with organic rich dark grey fine grained

sediment (mud) of clay and silt (Fig. 11). These coarse layers are interbedded with three mud

layers from 6 to 10 cm, 14 to 18 cm and 26 to 30 cm core depth (Fig. 11). The thickest fine

grained layers are typically composed of clay and silt sediments. The core BL12-10 is

dominated by the bimodal and trimodal grain size distributions. These distributions were

labeled as very coarse silty to very fine sand, poorly to very poorly sorted, fine skewed with

leptokurtic distribution (Table 5). Down-core profiles of heavy and light elements through the
depth also delineate the different units distinguished by sedimentological analysis (Fig.11).
Based on their profiles, the first group composed by Fe, Ti, K and Al exhibit similar
variations, concentration values are mainly high in fine-grained intervals and are low in
coarse-grained intervals. These high values are probably due to high inputs from the Fessi
River. The Si and Zr which characterized the second group display a different behaviour than
the first group (Fig.11). These two elements are high in the fine sandy intervals. This probably
suggests that their highest values are related to aeolian inputs in the lagoon. The Ca and Sr
characterised the third group show a reverse distribution pattern by comparison to the first
group with higher values in the coarse grained intervals and lower values in the fine grained
intervals (Fig.11). Single element concentrations may be sensitive to dilution effects to allow
reliable reconstructions of terrestrial climate, elemental ratios often better reflect the origin of
the sedimentary material. The measured elemental ratios Fe/Ca and Ti/Ca will be used to
reconstruct pas flood events (Fig. 9). A higher Fe/Ca and Ti/Ca ratio in the lagoon sediments
would be a signal of more sediment contribution from the Fessi River during flooding.
**6. Discussion**
**6.1 Paleoflood reconstructions**
In order to identify the paleo-flood events of the El Bibane Lagoon, we applied these
previously discussed proxies to BL12-10 core samples. The BL12-10 core shows 3 mud
layers (clay and silt mixture) preserved in the core which seems to be flood layers, i.e.,
coming from fluvial incursions during intense flood events. Multiproxy analysis on these mud
layers show that they are characterized by high content in clay+silt, as well as high Fe/Ca and
Ti/Ca elemental ratios which represent the sedimentological signature of the River Fessi. The
combination of geochemical and grain size data suggest that the BL12-10 core deposits had

registered three flood events namely FL1, FL2 and FL3 (Fig. 12). These flood deposits have a

thickness of 5cm, 4cm and 2.5cm respectively.

Our paleoflood reconstruction has been compared with historical rainfall data of Tataouine and Medenine (DGRE, 2000; Fehri, 2014). A good correlation is observed between instrumental rainfall records and past flood events recorded in the El Bibane lagoon. Based on our age model, FL1 would have occurred around AD 1995 ± 6 yrs (Fig. 12). This sediment deposit could correspond to the 1995 flood event recorded in hydrological data (Fehri, 2014) and which affected the entire Tataouine region. This flood reached a maximum discharge of 1200 m$^3$/s due to a heavy precipitation event during 24 hours (Boujarra and ktita, 2009). These events provoked heavy losses in human lives and agricultural goods (Boujarra and Ktita, 2009). Using the same approach, FL2 would have occurred around AD 1970±9 yrs, i.e. between AD 1965 to 1980 (Fig.12). Between these dates, two historical extreme flood events are known (AD.1969 and AD.1979) and one flood event of lower magnitude (AD.1972). The 1969 flood event is characterized by a heavy precipitation (400 to 600 mm) during 24 to 48 (Pias et Stuckmann, 1970, Kallel et al., 1972 and Boujarra and Ktita, 2009). The 1979 flood event is characterized by a heavy precipitation during 4 days (Bonvallot, 1979). Only one horizon corresponds to these events in the BL12-10 core. Consequently, we assume that this unique flood deposit registers a period during which these three high precipitation events occurred (i.e. AD.1969, AD.1972 and AD.1979). The activity of $^{210}$Pb in this flood deposit is not disturbed; it is homogeneous (Fig. 10). For this reason we assume that no significant erosion happened in the lagoon during this period. During these heavy precipitation events, most of the sedimentary material was deposited in the floodplain, in the lagoon and probably transported to the Mediterranean Sea through the passes. The sedimentation rate corresponding to these events is not very high. The thickness of the sediment layer associated with these flood events is low, i.e. about 5 cm. The grain size and geochemical values of this

flood deposit are rather homogeneous. This homogeneity is probably linked to the action of weak bottom currents within the El Bibane lagoon. Finally, since these three extreme flood events are very close together in time (1969-1979) and the sedimentation rate is low, they are recorded as only one sedimentary deposit (FL2) in our archive. The third flood event FL3 was dated at A.D 1945±9 (Fig. 12). It could be associated to the 1932 flood event (Fehri, 2014). This event was characterized by a flash flood event with a precipitation of 449 mm in few days. Bonvallot (1979) demonstrated that this event presents a similar characteristic than that of 1979.

El Bibane flood record shows temporal correspondence of flood layers to historical heavy precipitation events. Considering the historical data, we can assume that FL3 flood deposit corresponds to A.D 1932 flood.  FL2 flood deposit is associated to A.D 1969, A.D 1972 and A.D 1969 flood events. FL1 flood deposit could be associated to the A.D 1995 flood event (Fig. 12). In this lagoonal environment, one flood deposit is not always associated to a single event but sometimes to two or three events especially when heavy precipitation events are close together in time (i.e. FL2 flood deposit). Moreover these data demonstrate that finer material with a high content of mud (clay+silt), and high ratios of Fe/Ca and Ti/Ca are associated to flood events in the lagoonal sequence. The association of these proxies in the sedimentary sequence of the El Bibane lagoon can therefore be used to reconstruct flood activities in Southeastern Tunisia.

**6.2. The El Bibane lagoon: A key region for paleohydrological reconstructions**

Lagoon records shows that such costal environments are good study areas to record past climatic and environmental changes, and extreme sea events. These fields of research were successfully applied in the western North Atlantic (Donnelly and Woodruff, 2007), Northwest Florida (Liu and Fearn, 2000; Lane et al., 2011; Das et al., 2013), the Northeastern United States (Parris et al., 2010), the Central Pacific (Toomey et al., 2013), Southern Japan

(Woodruff et al., 2009), Western Australia (Nott, 2011), Northeastern New Zealand (Page et
al., 2010), Northern Europe (Sorrel et al., 2012), or the Western Mediterranean (Dezileau et
al., 2011, 2016; Sabatier et al., 2012; Raji et al., 2015; Degeai et al., 2015). Such studies are
still scarce in southern Tunisia, despite the importance of these topics in Mediterranean
coastal areas. The El Bibane lagoon is different from the other studied lagoons because it
cannot record coastal overwash events. Such particularity is linked to the morphology of
barriers that separate this lagoon from the open sea. These barriers consist of two narrow
fossil carbonate consolidated peninsula formed during the last interglacial period and reaching
10 m elevation (Medhioub, 1979; Jedoui, 2000). Thus they cannot not be over-washed during
extreme sea events. However, we have demonstrated from this study that this lagoon could
record past flood events during exceptional heavy precipitation episodes that punctuated the
recent meteorological and climatic history of Tunisia and North Africa. Tramblay et al.,
(2013) have analysed the influence of large-scale atmospheric circulation, including the North
Atlantic Oscillation (NAO), Mediterranean Oscillation (MO), El Nino-Southern Oscillation
(ENSO) and Western Mediterranean Oscillation (WEMO) on precipitations and extreme
events in 22 stations located in Algeria, Morocco and Tunisia for the last 50 years. Although
some spatial patterns for the different precipitation indices have been identified over Maghreb
countries the southern part of Tunisia was only represented by one meteorological station
(Gabes). This clearly avoid to identify an homogeneous climatic region, there is a need to
include more stations with longer record length. El Bibane lagoon paleoflood record can be
of great importance to better understand the physical mechanism responsible for the changes
in the frequency and/or the intensity of extreme events in the southern part of Tunisia. It will
be interesting to study the natural variability of past flood events in this semi-arid
environment through contrasting climatic periods (cold and warm periods). Further coming
investigations on long core sediments could clarify the relationship between large-scale
atmospheric circulation reconstructions and the major flood periods (Affouri et al., data in
progress). Additionally, such studies could be a crucial tool to evaluate the role of
Mediterranean paleo-climate on the development and growth of human society.
**Conclusion**
This study focuses on the sedimentological and geochemical characterization of the main
surface sediments sources of El Bibane Lagoon (southeast Tunisia) and its watershed in order
to identify the specific signature of paleoflood events recorded in the sedimentary core
archives. We used Principal Component Analysis (PCA) to identify the main factors
controlling the chemical composition of the catchment and El Bibane lagoon surface
sediments and to discriminate between the sources of detrital inputs into the lagoon. Three
sediments sources were identified: Marine, fluvial and Aeolian. Our results display that El
Bibane Lagoon surface sediment characteristics are situated between marine and river
sources. The application of this multi-proxy analysis on the BL12-10 core shows that finer
material, high content of mud (clay+silt), as well as high elemental ratios (Fe/Ca and Ti/Ca)
typify the sedimentological signature of flood events in the lagoonal sequence. The BL12-10
age model based on $^{210}$Pb and $^{137}$Cs activity profiles have allowed us to identify three periods
of past flood events dated at AD 1995±6, AD 1970±9, and 1945±9. The good agreement
between our estimated ages and the historical flood events suggests that sedimentological and
geochemical data of lagoon sediment cores could be used to reconstruct paleoflood history in
South-eastern Tunisia in arid and semi-arid environment during the upper Holocene.
**Acknowledgments**
Our thanks go to Dr. M. Ouaja, Ph. Blanchemache and J.P. Degai for their help on the field.
We also thank Pr. Y. Jedoui and G. Siani for their fruitful suggestions in the discussions. This
study is funded by the MISTRALS PALEOMEX and the PHC-UTIQUE N° 14G1002
projects. We are particularly grateful for editor comments of Dr Nathalie Combourieu-
Nebout. We thank Maria-Angella Bassetti, MCF-HDR Assistant Professor at the University
of Perpignan, France and anonymous reviewers for their helpful comments and their criticism,
which led to a considerable improvement of the manuscript.

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

**Figures Captions**

**Figure 1.** Location of the study area of El Bibane Lagoon (EBL) South East of Tunisia (A) and the geological map of South Eastern Tunisia (Modified from the Geological map of Tunisia 1/500000 after Ben Haj Ali et al., 1985) (B).

**Figure 2.** Variation of the annual precipitations of the Medenine and Tataouine meteorological stations during the period between 1900 and 2000 (DGRE, 2010). Dashed line: mean annual precipitation.

**Figure 3.** Location of the investigated surface samples from the catchment basin and from the El Bibane Lagoon.

**Figure 4.** Microtextural photos under binocular observation of five representative samples from the catchment basin of El Bibane Lagoon. S3 Marine sample; S8 and S11: Fessi River samples; S17 and S18: Dunes samples (Diameter of the photos: 3 cm; G x 6.5).

**Figure 5.** Particle size distributions (<2000µm) of representative samples from the catchment basin and the El Bibane Lagoon.

**Figure 6**: Distribution of the mean size of the samples collected in the Fessi River

**Figure 7.** Distribution map of major and trace elements in surface sediments from catchment basin and the El Bibane lagoon.

**Figure 8.** Principal Component Analysis (PCA) loadings plot of major and trace elements concentrations displaying the three main sources: marine, fluvial and aeolian sand dune.

**Figure 9.** Distribution of the investigated surface samples from the watershed and the El Bibane Lagoon on a cross-plot Fe/Ca *versus* Ti/Ca

**Figure 10.** $^{210}Pb_{ex}$ and $^{137}Cs$ activity-depth profiles along the core BL12-10. SR: sedimentation rate (cm yr$^{-1}$).

**Figure 11.** Records of eight geochemical elements (expressed in percentage or ppm) *versus* depth in core BL12-10.

**Figure 12.** (a) Paleoflood records in sedimentary archive of core BL12-10 based on elemental
ratios of Fe/Ca and Ti/Ca and grain size analysis (clay + silt ; fraction <63µm). Triangles
indicate the age control obtained using $^{210}$Pb and $^{137}$Cs along the core. Colored areas display
the three periods of floods recorded in the core (FL1, FL2 and FL3). (b) Observed rainfall
record since, 1932 in Medenine and Tataouine stations, is also shown.
**Tables captions**
**Table 1.** Geographic location and GPS coordinate of the studied samples
**Table 2.** Grain size statistical analysis of surface samples from the watershed of the El Bibane
Lagoon.
**Table 3.** XRF analysis results of the major and trace element in studied samples.
**Table 4.** Activities of radionuclides $^{210}$Pb, $^{137}$Cs and $^{226}$Ra along the core BL12-10.
**Table 5.** Grain size statistical analysis along the core BL12-10.













**Figure 1**

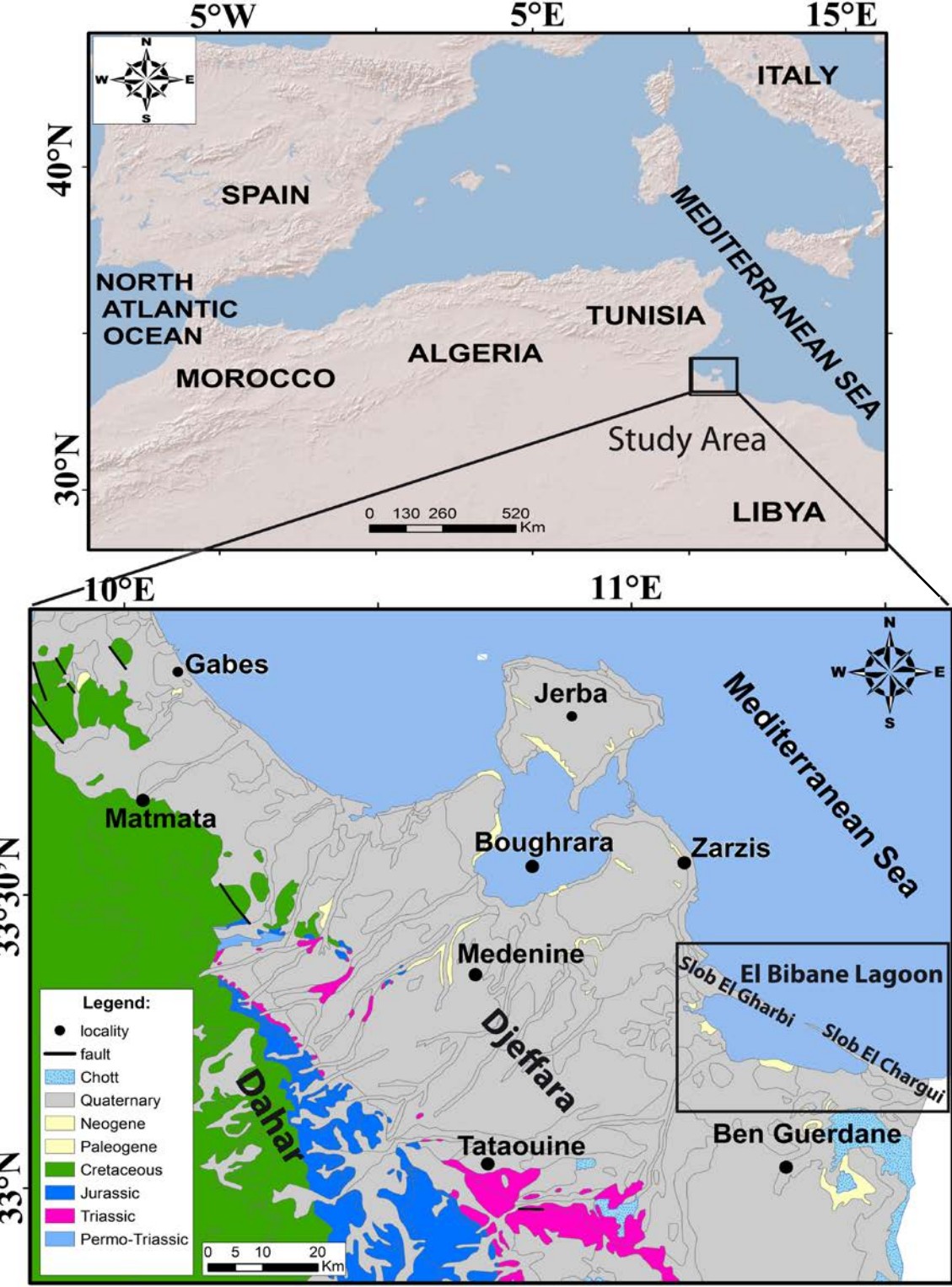




**Figure 2**

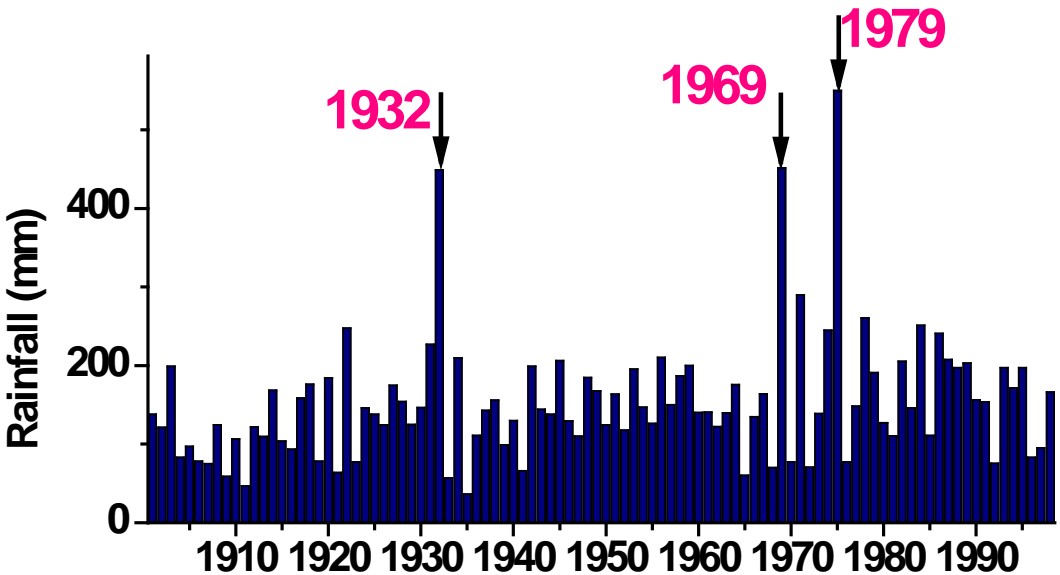

**Annual Rainfall of the Medenine watershed**

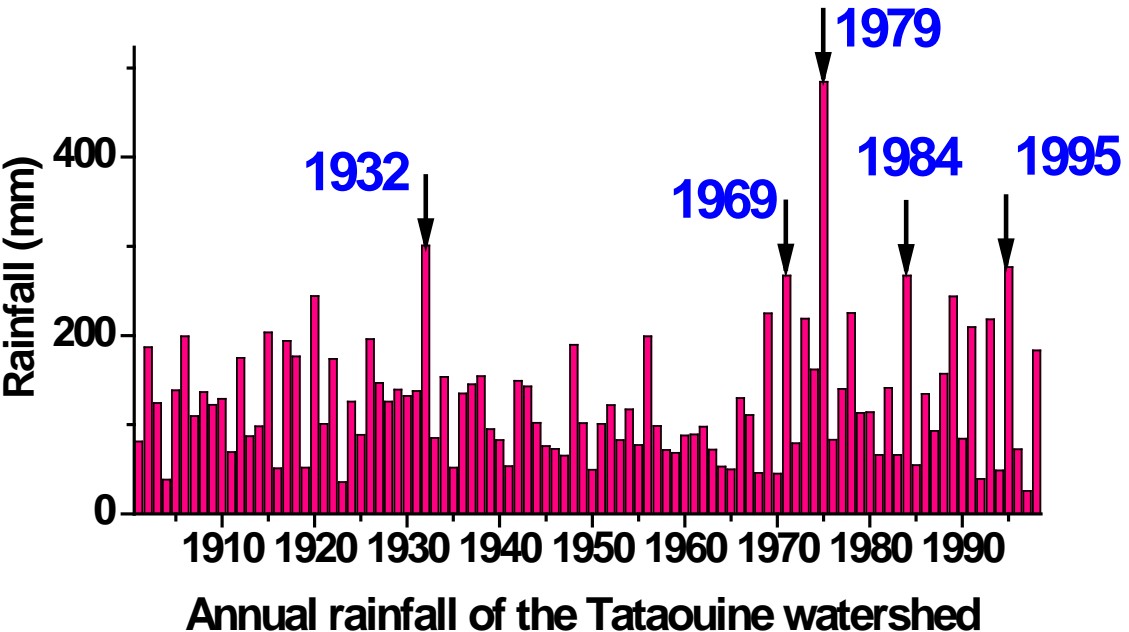

**Annual rainfall of the Tataouine watershed**






**Figure 3**

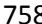





**Figure 4**

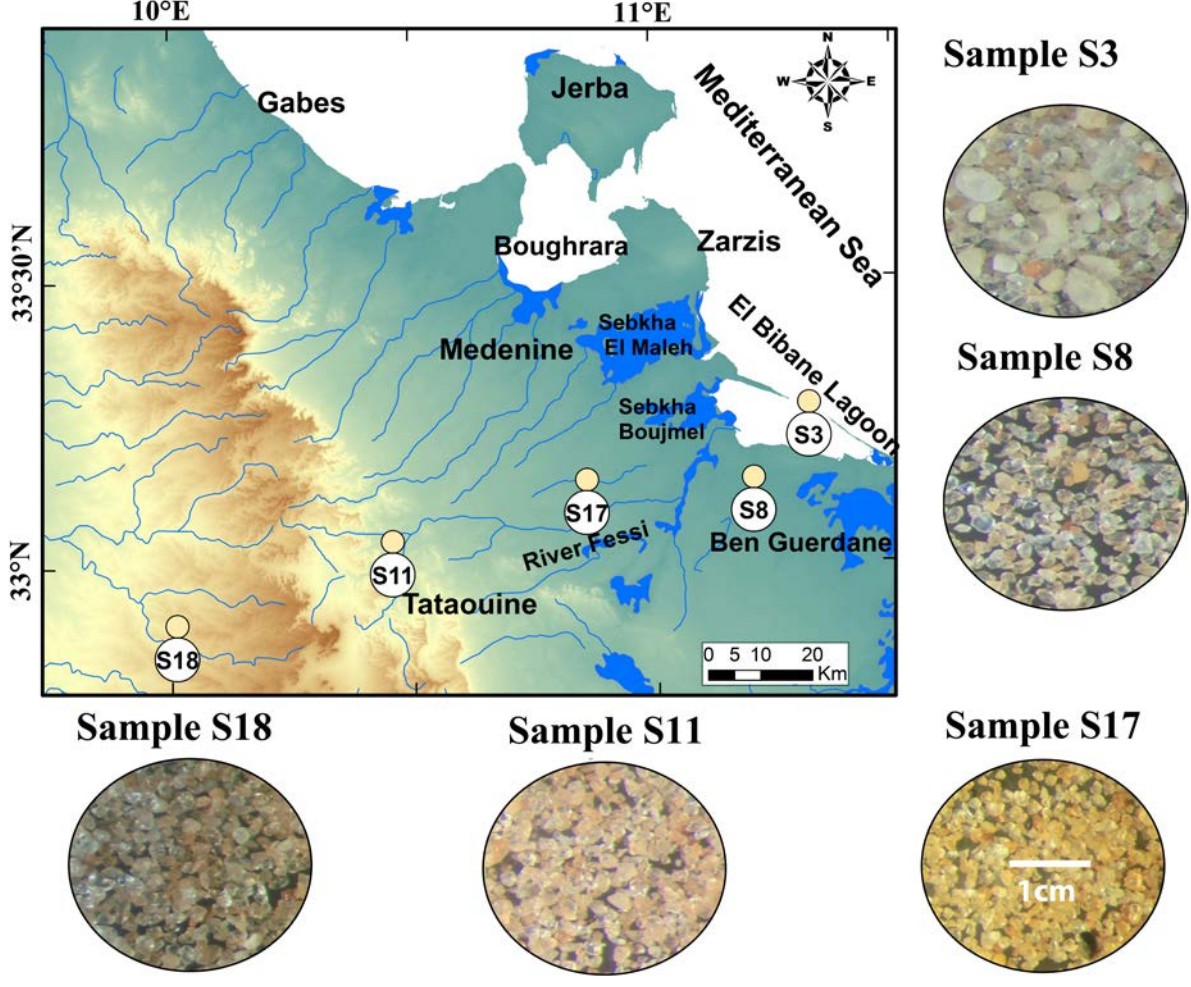

**Figure 5**

**MARINE SAMPLES**

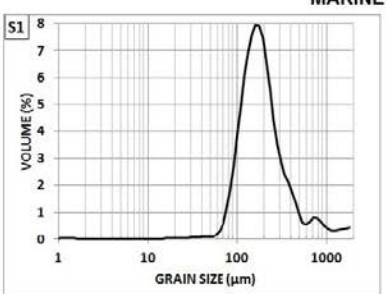
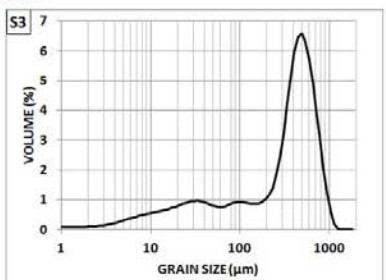

**FLUVIAL SAMPLES**

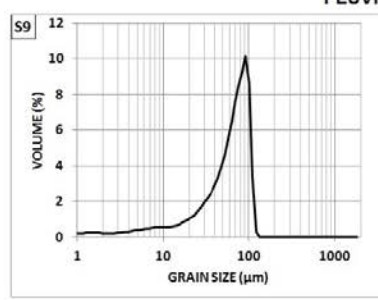
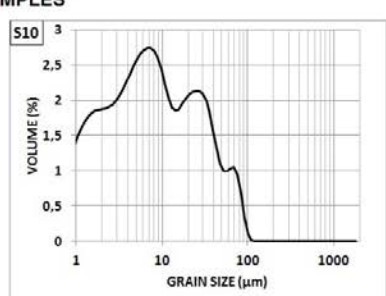

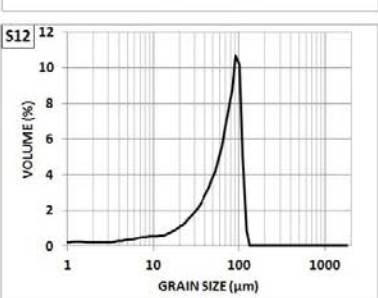
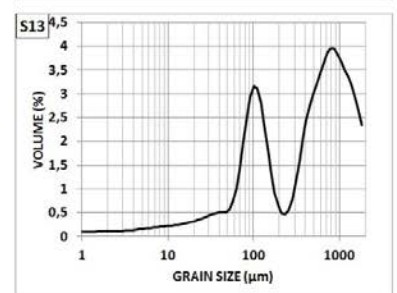

**AEOLIAN DUNES SAMPLES**

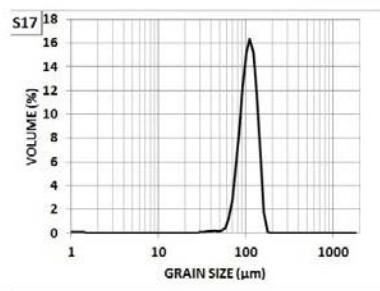
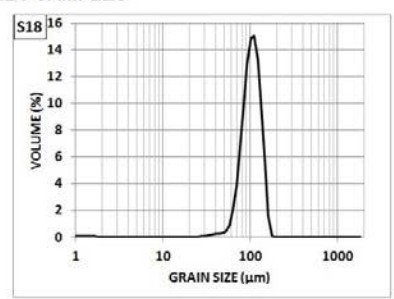

**LAGOONAL SAMPLES**

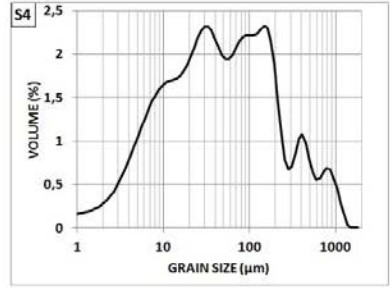
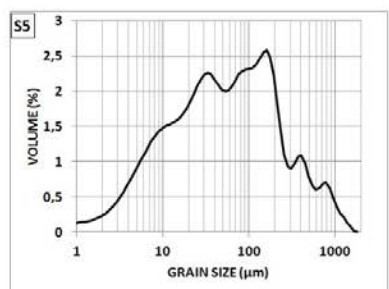


**Figure 6**



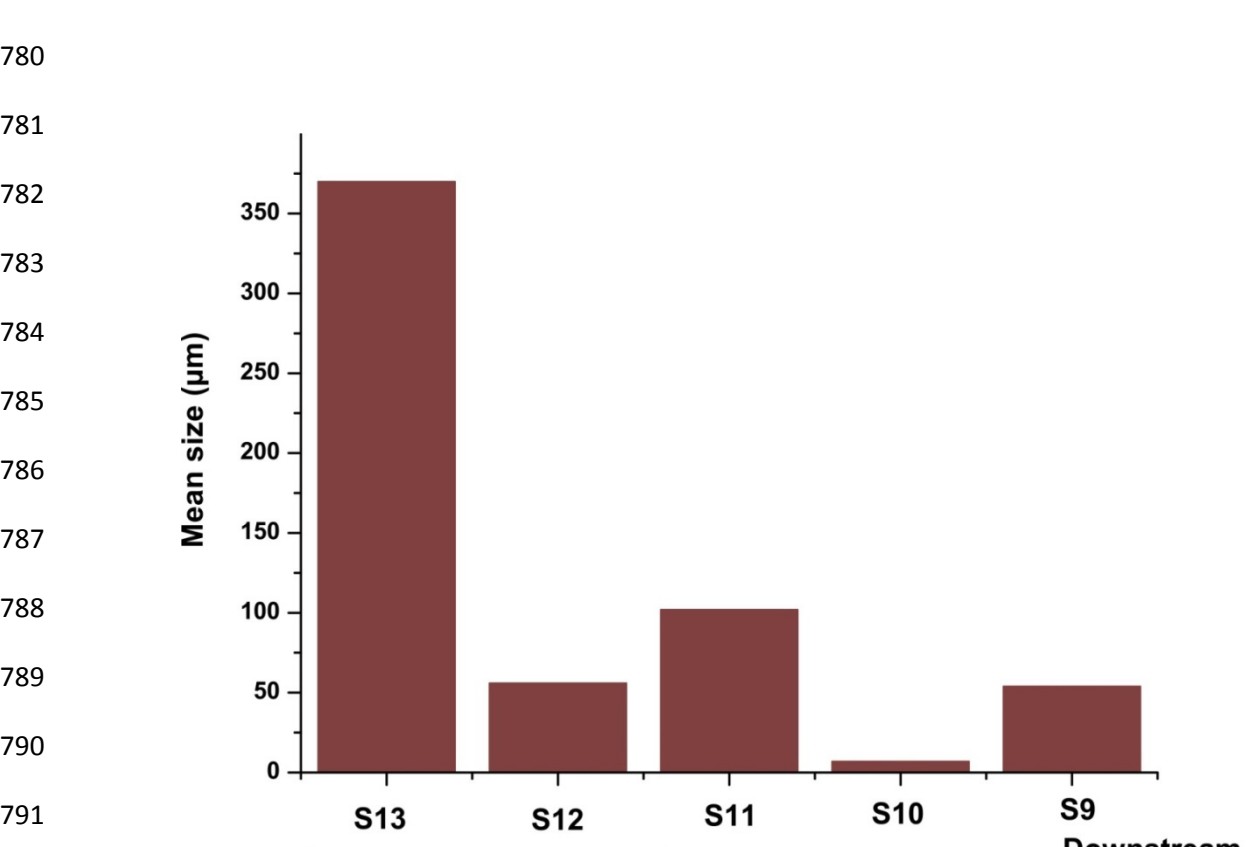

791

**Figure 7**

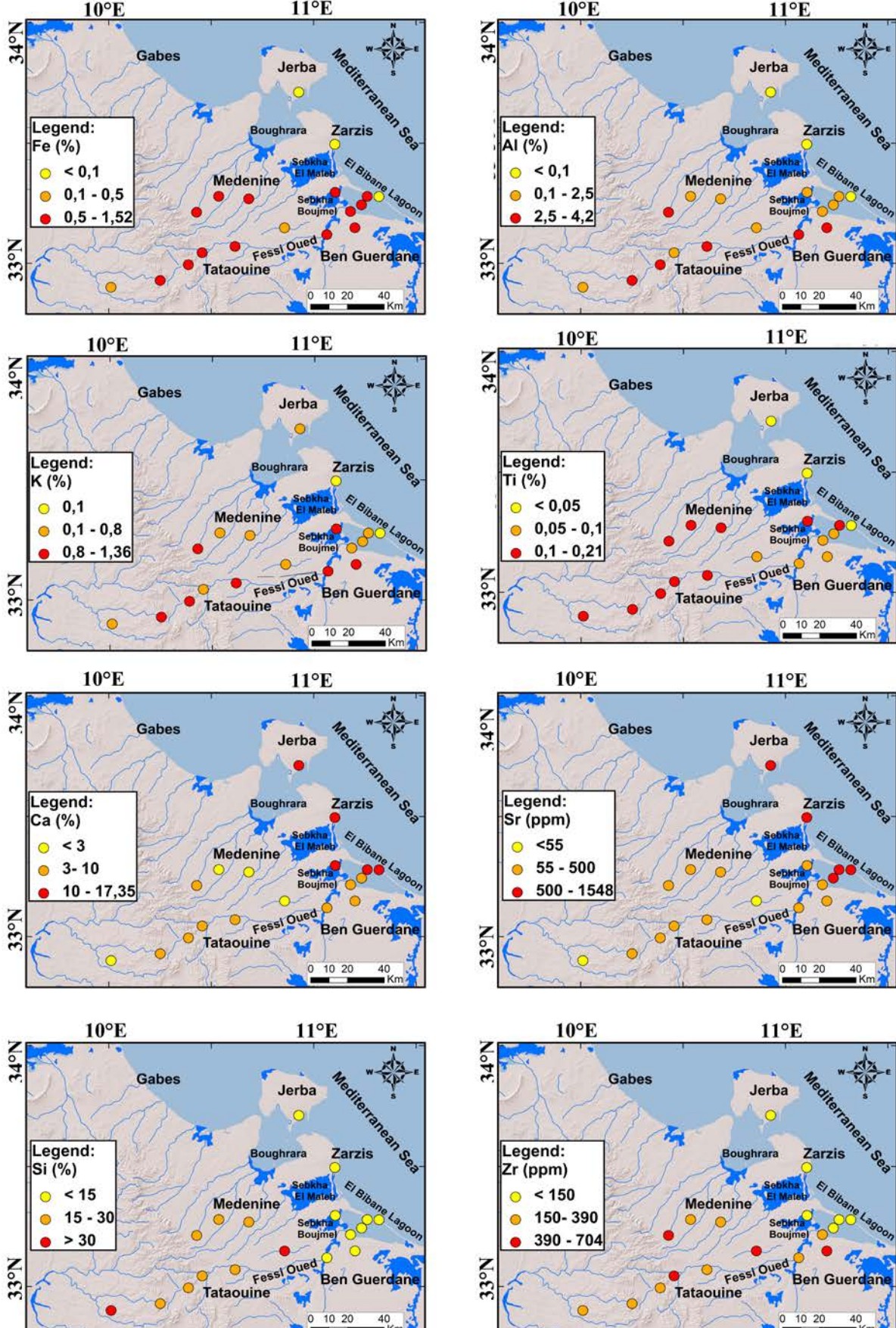

**Figure 8**

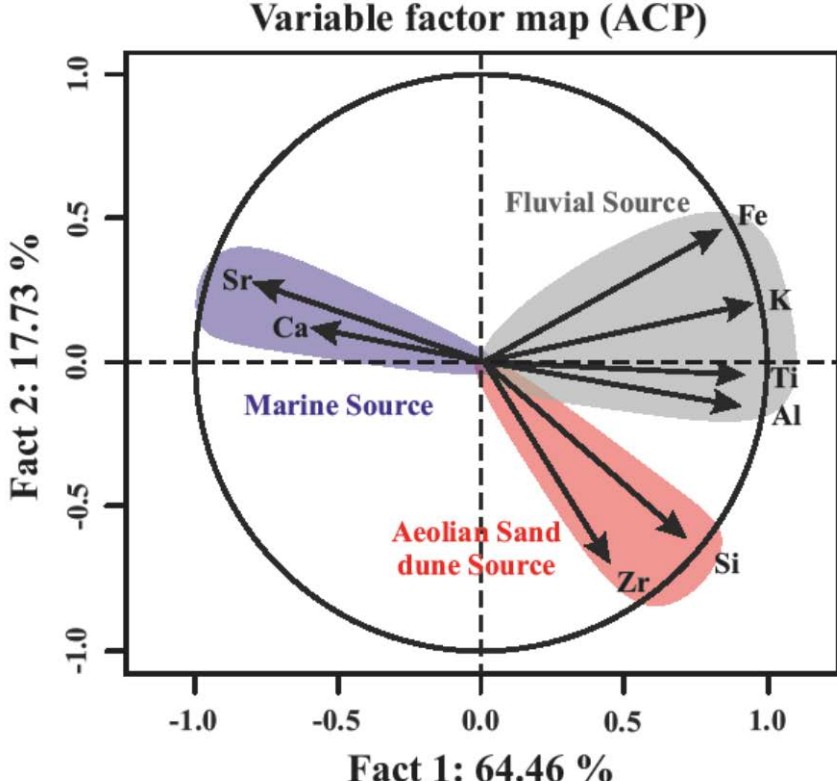














 **Figure 9**

**Figure 10**

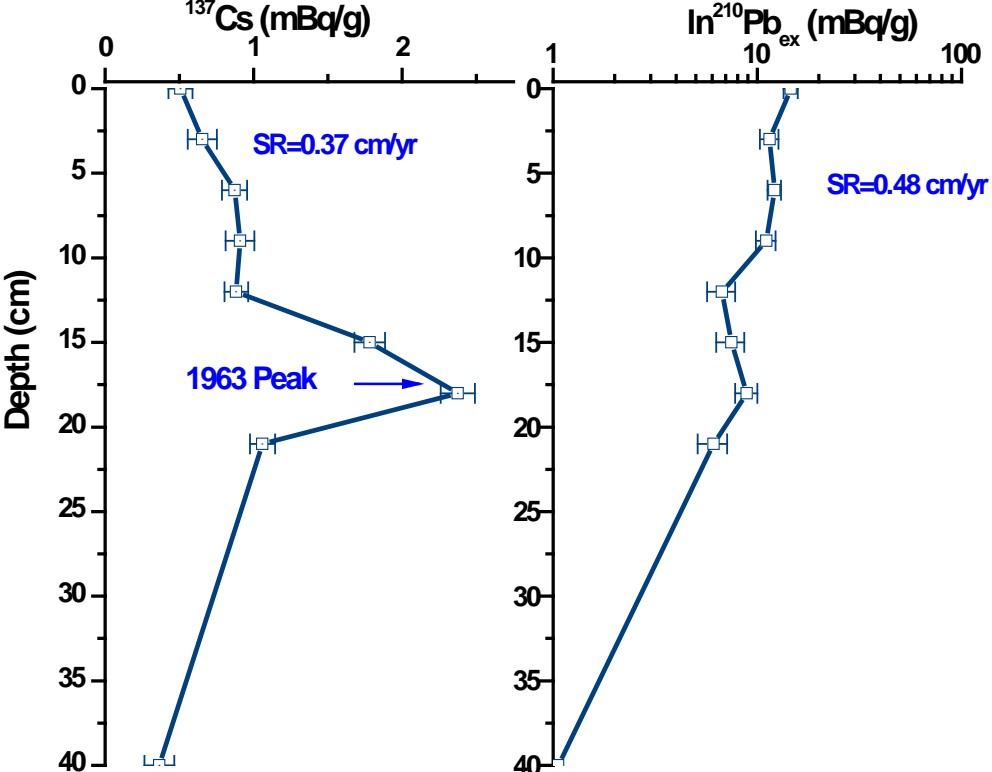















**Figure 11**

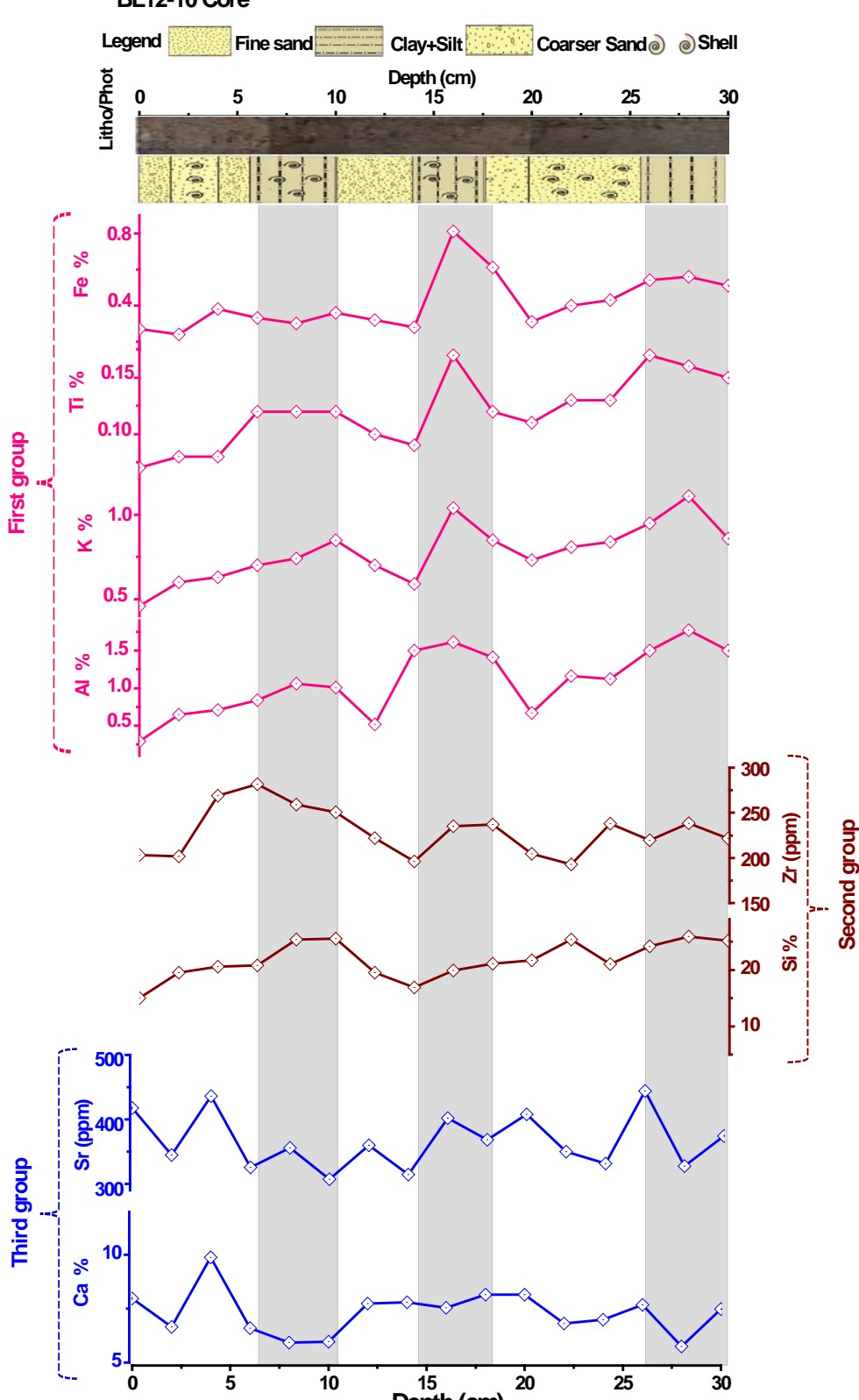

**Figure 12**

Last century histrorical rainfall of the Tataouine and Medenine stations


**Table 1**

| Sample | Locality | GPS coordinates | |
| --- | --- | --- | --- |
| | | Latitude | Longitude |
| S1 | Beach | 33°45'12.4" | 10°59'57.9" |
| S2 | Beach | 33°35'31.5" | 11°04'45.2" |
| S3 | Beach | 33°16'39.9" | 11°17'39.6" |
| S4 | Lagoon | 33°15'38.7" | 11°16'40.6" |
| S5 | Lagoon | 33°14'0.01" | 11°17'.02" |
| S6 | Lagoon | 33°13'52.3" | 11°06'31.3" |
| S7 | River | 33°16'52.3" | 11°07'31.3" |
| S8 | River | 33°08'03.0" | 11°06'51.6" |
| S9 | River | 33°03'32.1" | 11°02'00.4" |
| S10 | River | 33°04'13.6" | 10°40'56.0" |
| S11 | River | 32°59'23.4" | 10°28'12.7" |
| S12 | River | 32°55'18,0" | 10°24'15.1" |
| S13 | River | 32°55'09.7" | 10°22'35,3" |
| S14 | River | 33°03'38.0" | 10°24'05.6" |
| S15 | River | 33°09'59.2" | 10°21'35.8" |
| S16 | River | 33°12'25.37" | 10°26'46.78" |
| S17 | Aeolian | 33°07'18.9" | 10°44'58.6" |
| S18 | Aeolian | 32°50'28.4" | 10°13'43.7" |









**Table 2**

| Sample name | Sampling Locality | SAMPLE TYPE | TEXTURAL GROUP | SEDIMENT NAME |
|---|---|---|---|---|
| S1 | Beach | Unimodal, Moderately Sorted | Sand | Moderately Sorted Fine Sand |
| S2 | | Unimodal, Moderately Sorted | Sand | Moderately Sorted Fine Sand |
| S3 | | Unimodal, Very Poorly Sorted | Muddy Sand | Very Coarse Silty Coarse Sand |
| S4 | Surface sediments El Bibane Lagoon | Polymodal, Very Poorly Sorted | Sandy Mud | Very Fine Sandy Very Coarse Silt |
| S5 | | Unimodal, Moderately Sorted | Muddy Sand | Very Coarse Silty Fine Sand |
| S6 | | Bimodal, Poorly Sorted | Muddy Sand | Very Coarse Silty Very Fine Sand |
| S9 | Fessi River | Unimodal, Poorly Sorted | Muddy Sand | Very Coarse Silty Very Fine Sand |
| S10 | | Trimodal, Poorly Sorted | Mud | Fine Silt |
| S11 | | Unimodal, Well Sorted | Sand | Well Sorted Very Fine Sand |
| S12 | | Unimodal, Poorly Sorted | Muddy Sand | Very Coarse Silty Very Fine Sand |
| S13 | | Bimodal, Poorly Sorted | Muddy Sand | Very Coarse Silty Coarse Sand |
| S17 | Sand dune | Unimodal, Very Well Sorted | Sand | Very Well Sorted Very Fine Sand |
| S18 | | Unimodal, Well Sorted | Sand | Well Sorted Very Fine Sand |

| Sample name | FOLK AND WARD METHOD (µm) | | | | | | |
| | MEAN | SORTING | SKEWNESS | KURTOSIS | MODE 1 (µm) | MODE 2 (µm) | MODE 3 (µm) |
|---|---|---|---|---|---|---|---|
| S1 | 196.20 | 1.79 | 0.23 | 1.31 | 169.10 | | |
| S2 | 249.10 | 1.81 | 0.18 | 1.11 | 203.70 | | |
| S3 | 204.20 | 4.23 | -0.66 | 1.02 | 517.80 | | |
| S4 | 43.46 | 4.68 | -0.03 | 0.93 | 154.00 | 31.54 | 96.60 |
| S5 | 112.50 | 1.81 | -0.22 | 1.20 | 116.40 | | |
| S6 | 80.39 | 3.15 | -0.24 | 1.70 | 106.00 | 429.70 | |
| S9 | 54.69 | 2.24 | -0.57 | 1.49 | 96.60 | | |
| S10 | 7.13 | 3.89 | 0.00 | 0.84 | 7.09 | 26.17 | 73.02 |
| S11 | 102.50 | 1.34 | -0.24 | 1.22 | 116.40 | | |
| S12 | 56.17 | 2.25 | -0.57 | 1.42 | 96.60 | | |
| S13 | 370.90 | 3.90 | -0.41 | 0.88 | 825.40 | 106.00 | |
| S17 | 110.50 | 1.26 | -0.13 | 1.01 | 116.40 | | |
| S18 | 106.40 | 1.29 | -0.13 | 1.03 | 116.40 | | |

 **Table 3**

| Sample name | Locality | Zr (ppm) | Sr (ppm) | Ca (%) | Fe (%) | Ti (%) | K (%) | Al (%) | Si (%) |
|:---:|:---:|:---:|:---:|:---:|:---:|:---:|:---:|:---:|:---:|
| S1 | Beach | 113 | 1497 | 14.67 | 0.00 | 0.03 | 0.14 | 0.00 | 9.71 |
| S2 | Beach | 41 | 1548 | 14.51 | 0.00 | 0.01 | 0.10 | 0.00 | 6.85 |
| S3 | Beach | 24 | 899 | 13.36 | 0.00 | 0.01 | 0.10 | 0.00 | 8.38 |
| S4 | Lagoon | 133 | 1035 | 17.35 | 0.75 | 0.13 | 0.74 | 0.40 | 15.00 |
| S5 | Lagoon | 85 | 747 | 9.00 | 0.47 | 0.10 | 0.47 | 0.18 | 8.70 |
| S6 | Lagoon | 203 | 418 | 7.90 | 0.27 | 0.07 | 0.56 | 0.69 | 12.00 |
| S7 | River | 134 | 358 | 17.35 | 0.75 | 0.13 | 1.10 | 2.08 | 15.00 |
| S8 | River | 488 | 90 | 9.00 | 0.53 | 0.10 | 0.81 | 2.60 | 8.70 |
| S9 | River | 178 | 97 | 7.90 | 0.98 | 0.07 | 1.13 | 2.76 | 12.00 |
| S10 | River | 235 | 105 | 7.30 | 1.52 | 0.21 | 1.36 | 4.20 | 26.16 |
| S11 | River | 704 | 92 | 6.00 | 0.59 | 0.16 | 0.56 | 2.20 | 26.93 |
| S12 | River | 275 | 173 | 7.37 | 1.22 | 0.21 | 1.12 | 3.60 | 27.43 |
| S13 | River | 391 | 123 | 7.35 | 1.28 | 0.18 | 0.93 | 2.60 | 27.13 |
| S14 | River | 458 | 186 | 7.16 | 0.79 | 0.20 | 0.87 | 2.70 | 26.18 |
| S15 | River | 350 | 102 | 3.95 | 0.59 | 0.17 | 0.77 | 2.40 | 29.08 |
| S16 | River | 263 | 73 | 3.22 | 0.62 | 0.11 | 0.74 | 1.80 | 25.62 |
| S17 | Aeolian | 473 | 52 | 0.80 | 0.40 | 0.10 | 0.75 | 2.50 | 33.38 |
| S18 | Aeolian | 357 | 54 | 0.81 | 0.38 | 0.12 | 0.74 | 2.40 | 33.09 |















**Table 4**

| Sample name | Locality | Zr (ppm) | Sr (ppm) | Ca (%) | Fe (%) | Ti (%) | K (%) | Al (%) | Si (%) |
|---|---|---|---|---|---|---|---|---|---|
| S1 | Beach | 113 | 1497 | 14.67 | 0.00 | 0.03 | 0.14 | 0.00 | 9.71 |
| S2 | Beach | 41 | 1548 | 14.51 | 0.00 | 0.01 | 0.10 | 0.00 | 6.85 |
| S3 | Beach | 24 | 899 | 13.36 | 0.00 | 0.01 | 0.10 | 0.00 | 8.38 |
| S4 | Lagoon | 133 | 1035 | 17.35 | 0.75 | 0.13 | 0.74 | 0.40 | 15.00 |
| S5 | Lagoon | 85 | 747 | 9.00 | 0.47 | 0.10 | 0.47 | 0.18 | 8.70 |
| S6 | Lagoon | 203 | 418 | 7.90 | 0.27 | 0.07 | 0.56 | 0.69 | 12.00 |
| S7 | River | 134 | 358 | 17.35 | 0.75 | 0.13 | 1.10 | 2.08 | 15.00 |
| S8 | River | 488 | 90 | 9.00 | 0.53 | 0.10 | 0.81 | 2.60 | 8.70 |
| S9 | River | 178 | 97 | 7.90 | 0.98 | 0.07 | 1.13 | 2.76 | 12.00 |
| S10 | River | 235 | 105 | 7.30 | 1.52 | 0.21 | 1.36 | 4.20 | 26.16 |
| S11 | River | 704 | 92 | 6.00 | 0.59 | 0.16 | 0.56 | 2.20 | 26.93 |
| S12 | River | 275 | 173 | 7.37 | 1.22 | 0.21 | 1.12 | 3.60 | 27.43 |
| S13 | River | 391 | 123 | 7.35 | 1.28 | 0.18 | 0.93 | 2.60 | 27.13 |
| S14 | River | 458 | 186 | 7.16 | 0.79 | 0.20 | 0.87 | 2.70 | 26.18 |
| S15 | River | 350 | 102 | 3.95 | 0.59 | 0.17 | 0.77 | 2.40 | 29.08 |
| S16 | River | 263 | 73 | 3.22 | 0.62 | 0.11 | 0.74 | 1.80 | 25.62 |
| S17 | Aeolian | 473 | 52 | 0.80 | 0.40 | 0.10 | 0.75 | 2.50 | 33.38 |
| S18 | Aeolian | 357 | 54 | 0.81 | 0.38 | 0.12 | 0.74 | 2.40 | 33.09 |

**Table 5**

| DEPTH (cm) | Sample name | SAMPLE TYPE | TEXTURAL GROUP | SEDIMENT NAME |
|---|---|---|---|---|
| 1 | BL12-10-1 | Bimodal, Poorly Sorted | Muddy Sand | Very Coarse Silty Very Fine Sand |
| 2 | BL12-10-2 | Trimodal, Very Poorly Sorted | Muddy Sand | Very Coarse Silty Very Fine Sand |
| 3 | BL12-10-3 | Trimodal, Poorly Sorted | Muddy Sand | Very Coarse Silty Very Fine Sand |
| 4 | BL12-10-4 | Trimodal, Very Poorly Sorted | Muddy Sand | Very Coarse Silty Very Fine Sand |
| 5 | BL12-10-5 | Trimodal, Poorly Sorted | Muddy Sand | Very Coarse Silty Very Fine Sand |
| 6 | BL12-10-6 | Trimodal, Poorly Sorted | Muddy Sand | Very Coarse Silty Very Fine Sand |
| 7 | BL12-10-7 | Trimodal, Poorly Sorted | Muddy Sand | Very Coarse Silty Very Fine Sand |
| 8 | BL12-10-8 | Bimodal, Poorly Sorted | Muddy Sand | Very Coarse Silty Very Fine Sand |
| 9 | BL12-10-9 | Bimodal, Poorly Sorted | Muddy Sand | Very Coarse Silty Very Fine Sand |
| 10 | BL12-10-10 | Trimodal, Poorly Sorted | Muddy Sand | Very Coarse Silty Very Fine Sand |
| 11 | BL12-10-11 | Trimodal, Poorly Sorted | Muddy Sand | Very Coarse Silty Very Fine Sand |
| 12 | BL12-10-12 | Trimodal, Very Poorly Sorted | Muddy Sand | Very Coarse Silty Very Fine Sand |
| 13 | BL12-10-13 | Trimodal, Poorly Sorted | Muddy Sand | Very Coarse Silty Very Fine Sand |
| 14 | BL12-10-14 | Trimodal, Very Poorly Sorted | Muddy Sand | Very Coarse Silty Very Fine Sand |
| 15 | BL12-10-15 | Trimodal, Poorly Sorted | Muddy Sand | Very Coarse Silty Very Fine Sand |
| 16 | BL12-10-16 | Trimodal, Very Poorly Sorted | Muddy Sand | Very Coarse Silty Very Fine Sand |
| 17 | BL12-10-17 | Trimodal, Very Poorly Sorted | Muddy Sand | Very Coarse Silty Very Fine Sand |
| 18 | BL12-10-18 | Trimodal, Very Poorly Sorted | Muddy Sand | Very Coarse Silty Very Fine Sand |
| 19 | BL12-10-19 | Trimodal, Very Poorly Sorted | Muddy Sand | Very Coarse Silty Very Fine Sand |
| 20 | BL12-10-20 | Bimodal, Poorly Sorted | Muddy Sand | Very Coarse Silty Very Fine Sand |
| 21 | BL12-10-21 | Bimodal, Poorly Sorted | Muddy Sand | Very Coarse Silty Very Fine Sand |
| 22 | BL12-10-22 | Trimodal, Poorly Sorted | Muddy Sand | Very Coarse Silty Very Fine Sand |
| 23 | BL12-10-23 | Trimodal, Poorly Sorted | Muddy Sand | Very Coarse Silty Very Fine Sand |
| 24 | BL12-10-24 | Bimodal, Poorly Sorted | Muddy Sand | Very Coarse Silty Very Fine Sand |
| 25 | BL12-10-25 | Trimodal, Poorly Sorted | Muddy Sand | Very Coarse Silty Very Fine Sand |
| 26 | BL12-10-26 | Trimodal, Poorly Sorted | Muddy Sand | Very Coarse Silty Very Fine Sand |
| 27 | BL12-10-27 | Trimodal, Very Poorly Sorted | Muddy Sand | Very Coarse Silty Very Fine Sand |
| 28 | BL12-10-28 | Trimodal, Very Poorly Sorted | Muddy Sand | Very Coarse Silty Very Fine Sand |
| 29 | BL12-10-29 | Trimodal, Poorly Sorted | Muddy Sand | Very Coarse Silty Very Fine Sand |
| 30 | BL12-10-30 | Bimodal, Poorly Sorted | Muddy Sand | Very Coarse Silty Very Fine Sand |




