# Peer review of "Extreme flood events reconstruction spanning the last century in the El Bibane lagoon"

_Climate of the Past, 2016_

## Referee Comment (RC1) · Anonymous Referee #1 · 10 May 2016

General comments:

This is basically an interesting case study dealing with the tracking of palaeo-flood events in the El Bibane Lagoon (SE Tunisia) during the past century. The main objective of the study is to investigate sediment sources in the lagoon and to discriminate between fluvial, aeolian, marine end-members by using sedimentological and geochemical data. The patterns observed in modern sediments are expected to help deciphering ancient flood events in lagoonal deposits, as preserved in a core covering the past century based on a combined chronology using 210Pb and 137Cs data. I must be honest in saying that, if the study is relatively sound and acceptable, I've not been convinced in general by the novelty of the approach, and have in addition

several reservations regarding the interpretations (see the Specific Remarks). In particular, most of the results have been presented between the Results chapter and the Discussion, which renders the manuscript confusing and difficult to read. Alternatively some parts of the text have been totally overlooked and would benefit from further consideration/exploration before the manuscript can be accepted. The quality of the figures is overall acceptable, albeit some figures are of very poor graphical quality. The manuscript is not really well written, and should absolutely be revised by a native English before further consideration. I also regret that no tentative comparison with other regional datasets is provided in the Discussion, although I am pretty convinced that such a perspective would help to build a bigger picture of palaeoflood activity regionally. Finally, I do not believe that the manuscript provides the sort of conceptual and fundamental advance in our understanding of the processes and mechanisms governing lagoonal sedimentation and past central/southern Mediterranean climate that has been published elsewhere. For these reasons, I would not recommend this study to be published in Climate of the Past. However I leave this decision to the editorial board, who should appreciate the other reviewers' comments and recommendations.

Specific remarks:

1. Introduction :

Page 3, Lines 1-3 : Please provide more information dealing with the study of Raji et al., 2014 in Morocco, and show how the outcome of this work is related to the present study.

Page 3, end of the introduction : I would have appreciated to find here, as a foremost objective of the study, a perspective of data integration with other dataset covering the same time span, at a regional/larger scale.

3. Climate and hydrology

Page 5, lines 10-12: Please check the phrasing of that sentence. This is a regular

problem throughout the manuscript, which would highly benefit from a thorough cross-reading by a native English.

Page 5, lines 13-16 : When you refer to Figure 3, please also introduce the Medenine and Tataouine watersheds here (and not later at the beginning of chapter 4). There is a mistake with the spelling of Medenine on Figure 2.

4. Materials and Methods

Page 6, lines 11-13 : Please provide a general lithological description of core BL12-10, since we are not provided with any information with respect to the sedimentology at that stage.

In general the methods are described in an extremely concise way, and would perhaps merit more devotion. The information provided in the present version of the manuscript are indeed very limited (XRF, grain size analysis and age model using 210Pb and 137Cs). Why did you opt for a 1cm-resolution (only) with the XRF data, and not a higher resolution ? Is the sediment too homogeneous, thus rendering this perspective not promising ? Please elaborate on that.

Page 7, lines 11-12 : Please rephrase as I do not understand this sentence.

Page 7, lines 11-13 : I find this introduction for the PCA analyses far too simple ! Could you elaborate more on that ? For instance, since you use percentage values (both for grain-size and XRF data), have the raw data been square-root transformed, centred and standardized before applying the PCA analysis ? This is of great importance regarding the reliability of the results. Please clarify it.

5. Results

Page 8, lines 2-4 : On Fig. 6 the distribution of grain sizes appear different and more complex between S7 and S10 (fluvial end-member). For instance, the mode at $100\mu$ is not present on sample S10. Similarly, the mode 20-63 $\mu$ is not really obvious in S10. Is the pattern so tricky when considering other samples from the fluvial component (e.g.,

S8, S9, S1-S16) ? Please comment on that and eventually show more plots for the fluvial components.

What about the 4th group, i.e., the lagoonal samples ? It is neither presented so far in the text, nor shown on Figure 5 (although it does on Fig. 6, interestingly). The distribution looks rather complex for this fraction in Fig. 6, and obviously show a mixture between the different modal distribution (with at least a great contribution of fluvial samples).

Page 8, lines 15-24 : Here we are provided with XRF data given as percentage values. Please explain how these values have been obtained. Have the semi-quantitative XRF core-scanning data been calibrated by discrete XRF measurements as to determine linear regressions in cross plots and calculate percentage values from scanner data ? Please clarify on that, and above all, please show the raw XRF data (in cps) obtained on core BL12-10.

Another issue : Taken into account the very low ranges of variations (0-1,5% for Fe ; 0-0,2% for Ti), how can you be confident with the interpretations (i.e., the discriminations into different environmental pools) ?

Page 9, line 5 : Please change Fig. 7 into Fig. 8.

Page 9, lines 18-19 : Please rephrase here, a verb is missing.

Page 10, lines 5-6 : Do you mean mud or clay layers ? Mud is usually enriched in organic matter, whereas clayey sediments are not. What do you mean by mud layers typically composed of clay and silt sediments ? By the way, there is no mud shown on Figure 9 ;

Page 10, lines 6-7 : Where do these flood layers appear on Fig. 9 ? How did you identify it ? I regret that the quality of Fig. 9 is so poor ! Please redraw this figure accordingly.

Page 10, lines 16-18 : I do not understand this sentence ! It provides a very simplistic explanation for the discrepancies observed between 210Pb and 137Cs data. Did you also measure 241Am throughout core BL12-10, which would help in solving this apparent mismatch?

6. Discussion

Page 11, lines 20-23 and thereafter: I do not understand why the Discussion chapter still contains results/interpretation. The outcome of the PCA analysis should definitely be treated in the Results or Results/Interpretations chapter, but not in the Discussion !! Please modify this accordingly. The Discussion should be the locus where the results are integrated regionally, and at a larger scale, regarding the main scientific question identified in the introduction. Here we are provided with results in the Results chapter, followed by results in the Discussion chapter. See also my comment Page 7, lines 11-13.

Page 11, line 4 : I do not understand why this reference is suitable and of interest here. Please check this and correct accordingly.

Page 11, lines 13-15 : I do not agree that Fe and K (at least) showing negative loadings on Factor 2 !!

Overall, I am not convinced by the application of a PCA analysis here to discriminate between different sources. Please explain why the PCA analysis brings compelling useful evidence for the interpretation of environmental proxies.

Page 12, lines 3-9 : Are these results really unexpected ? What do we learn here ? Do we really need geochemical proxies, grain-size data and PCA analyses to show that lagoonal sediments are made of a mixture of continental and marine sources ? Why this still is presented in the Discussion ??

Page 12, lines 12-15 : Looking at the data, it is not really obvious that one could define genuine palaeoflood events. How do you discriminate between a background fluvial influence within the lagoon and genuine palaeoenvironmental disruptions (e.g.,

exceptional flood events recorded in the sediments = eventites) ? Is there a threshold to be considered in the data ?

Page 13, lines 17-19 : May this alternative explanation account for the apparent discrepancies observed between 210Pb and 137Cs data ? Apart from that, if the BL12-10 core consists of a background sedimentation disrupted by occurrences of flood events during the past century, it should definitely be taken into consideration when calculating average sedimentation rates. Did the FL1, FL2 and FL3 flood layers excluded for the estimation of sedimentation rates ? If not, this has to be commented.

Moreover, if the FL2 layer represents more than one flood deposit (e.g., 3 floods events as suggested), why do all sedimentological proxies (i.e., Fe/Ca and Ti/ca, clay+silt fractions) peak only once in FL2 ? What about the 1984 flood recorded in the Tataouine watershed as shown on Figure 3 ? May this correspond to the peaks observed in Fe/Ca and Ti/Ca at the lower end of FL1 (around 10 cm) ?

References :

Many references are listed in the reference list but do not occur in the text. There are listed here below, but please check the reference list in general.

Beker (1989) is missing in the text. Guelorget et al. (1982) is missing in the text. Plewa et al. (2012) is missing in the text. Prospero et al. (1981) is missing in the text. Raji (2014) is missing in the text. Richter et al. (2006) is missing in the text. Torres-Padron et al. (2002) is missing in the text.

Figures :

Figures 1, 2, 3, 4, 5, 7, 8 are of good visual and graphical quality in general. In contrast, Figures 6, 9, 10 and 11 are of poor quality (Fig. 12 acceptable) and should definitely be improved before the manuscript can be reconsidered.

---

## Referee Comment (RC2) · Anonymous Referee #2 · 4 Jun 2016

General Comments

The paper shows that this site has the potential for developing reconstructions of past flood events, as demonstrated by the correspondence between silt layers and twentieth century flood/precipitation events. It is really showing the potential for future reconstructions, rather than giving new data at the moment.

I think the paper could do more to emphasise the importance of the work. At the moment the abstract finishes with a statement that hydrological events can be reconstructed using sedimentary archives, which has already been shown elsewhere. Instead I would like to see more emphasis in the introduction, discussion and conclusion about what the wider implications are and why this site in particular may be important.

<hr>

For example: saying whether there are other reconstructions from this region that have reconstructed hydrological changes – and if not, highlighting that the paper shows that this site has the potential to provide this, which could answer questions on the recurrence intervals, magnitude changes etc...

The manuscript organisation needs improving, as there is some mixing of results, interpretation and discussion, which make it quite confusing at times.

In addition I feel the English needs better proof reading before publication to ensure it reads well. I have made some changes to this but not throughout.

Finally there are too many figures and some seem unnecessary so should be combined or removed.

Specific comments

Page 2, line 13 – maybe add here about what we can understand from geologic archives, for example the frequency, magnitude, patterns of events etc...

Page 3, line 1-3 – this should not be a paragraph on its own, so either remove or merge with previous paragraph

Page 3, line 17 – maybe use the word 'partially' before 'separated from the med...', as it looks on the map to not be completely separate.

Page 3, line 18 – are both peninsulas 12 km, make it clear if this is the combined length or not

Page 4 – there is an extensive geological description here which I am not sure is necessary. I think cut this down and make it more clear which parts are likely to influence the watershed of the lagoon, and the area on which it is situated.

Page 5, line 14-15 – this says that figure 3 shows that the precipitation events caused the flood events, however it only shows the precipitation records with no link to flooding. Remove the final part 'causing the flood events' or give more evidence.

Page 7, paragraph 1 – make this paragraph about the dating into a sub section on its own because it doesn't fit in this one about the proxies used. I am not familiar with what 210Pbex is and I think it might be better to refer to it as simply Pb210, as you do in the rest of the paper. If it is necessary it should maybe be defined when it is first mentioned.

Page 7, paragraph 2 – as you just used PCA it might be better to start the paragraph with 'Principle Component Analysis was used to understand the relationship....' because this makes the link with what you say in the next sentence. I would also merge the last two sentences in this paragraph.

Page 7, paragraph 3 – the first sentence suggests that you have made the measurements and then grouped them into 3 source areas, whereas I think you have actually taken samples from three different types of location and then present the characteristics of these. Change this first sentence to make it clear that this is what you did.

Page 9, paragraph 1 – in the first two sentences you have used 'sediment' in each and then described three different origins – make it clear which sediment you are describing. I think in the first sentence you mean the lagoon sediment and in the second sentence perhaps terrestrial sediment, but I am not sure. So make this clear and also reference each sentence. Is the third sentence about ionic radius important to know for understanding the results? I think it needs removing. Finally, the last sentence repeats the results from section 5.1.1., so I think that you could delete lines 11 and 12 (ending the sentence with 'observations') and in brackets put 'see 5.1.1'.

Page 9, paragraph 2 – the sentence about detrital quartz (line 18-19) does not fit in here and should be removed. The end of the paragraph from line 18-22 is also more of an interpretation and doesn't belong here in the results. Also would need to reference line 20.

Page 10, paragraph 1 – first sentence should be in the materials section, as it is about collecting the core – or delete this if it is already described in the methods. The final sentence in this paragraph is an interpretation, and I think would go better in an interpretation section.

Page 10, paragraph 2 – I don't think you should chose a Pb210 model based on the downcore distribution of Pb activity (and Pb210 activity often has exponential curve), I think it should be based on understanding of the deposition at the site, for example I think the CF:CS model assumes constant sedimentation, whereas the CRS model does not. Look at the assumptions of this model and then justify it in the methods section, and also make sure that this is referenced as it is not here. In line 10 you say the range is down to 0.1, but in table 3 the lowest is 1.058. Also the Pb210 profile in figure 10 shows a big gap between 21 and 40 cm with no samples dated – would it be possible to date between these? Without these depths you are not able to show where the equilibrium depth is (where Pb210 activity is around 0). I know that having this depth is important for the CRS model – I am not familiar with the CF:CS model so it may be different, but if possible further dates would help constrain the timing of the lowest flood layer.

Page 11, paragraph 1 – I would remove this section completely and move most of it (line 4-17) to the results (put it in section 5.1.2). The order of the figures would then also change.

Page 12, lines 1-9 – the term 'continental source' appears here and it is unclear what this describes, as previously only marine, fluvial and aeolian sources are described. Either change it or explain what is meant. Also what is meant by 'continental pole' and 'marine pole' is unclear, maybe use the 'sediments' or 'sources' instead of 'pole'.

Page 12, section 6.3 – I think it would help to have a paragraph or a few sentences at the start of this section clarifying the link between the sediment source characteristics and the sediment changes in the lake sediments. For example say that as the sediment from fluvial source was high in X,X,X then these characteristics in the core sediment will be used as a basis for interpreting flood events.

Page 13, line 2 - remove this sentence as it repeats the previous one.

Page 13, line 5-6 – I also feel this sentence is repetitive and has no reference, so should be removed

Page 12 and 13 - This is a very long paragraph so I would split it into three, with new paragraphs at page 12 line 23 ('From our age model...'), at page 13 line 6 ('Using the same approach') and page 13 line 14 ('Finally')

Page 13, line 11-13 – I am unclear what is meant by 'most of the sediments'. Did these studies show that sediment deposits were left by the 1969 event in particular? It is also not clear which event is being referred to as the latter event, so make it clear which it is.

Page 13, line 17-19 – I agree with this statement about the constraints on your pb210 dating, but maybe acknowledge this sooner in the results section

Figure 1 and 2: these could be combined, as the geological map could be put in place of the lower map on figure 1 which spans the same area and doesn't show much additional information.

Figure 5: This is interesting but I think it may not be important enough to require a figure. There are a lot of figures, so this might be one to remove. Also the figure caption refers to samples which are not shown on the figure (e.g.S4) so these should not be mentioned.

Figure 7 and 8: it is clear from the graph that the values are percentages, so remove this from the caption. These two figures could be combined, as they show the same type of data.

Figure 11 and 12: these are important for the interpretation so I think these figures should come earlier (before the figure 9 and 10 showing the core results). I think if the PCA section is moved to 5.1.2 as I suggest above then this should come forward, and figure 12 could also be introduced there as well. The paper might read better if there

is a clear first part looking at the catchment sources, followed by the results from the core.

Figure 13: this compares the precipitation and sediment cores using both depth and age. I understand that the pb210 model is not perfect but it might be better to have an additional x-axis on the top graph showing the estimated age, or at least ages marked on at the boundaries of the flood periods so that you can see the timings of these just by looking at the figure without needing to read the caption. Also, in black and white the bar for FL1 does not show up well.

Table 1: this could go in supplementary information. It is clear on figure 4 where they are all located and in the next table you state the type of locality they are.

Technical corrections

Page 1:

Title – 'spanning' might be better word than 'during'. Also, there is an uneven use of capital letters.

Line 10 – flood not floods

Line 11 – 'Recent studies of' not 'Recently, study of'

L12 – 'have enhanced' not 'contributed to enhance'

L14 – I think it should be 'multi-proxy approach' not 'multi-approach'. Also rethink word 'associating' and maybe use 'combining'

L16 – change to 'sediment deposits'

L17 – remove 's' at end of sediments

L22 – 'flood' not 'floods'

L23 – Chronology should not have a capital letter

L22 and L23 – merge these sentences perhaps '...137Cs chronology, and give ages of AD 1995....'

Page 2:

L1 and L2 – remove first half of sentence maybe up until suggests and then merge with previous sentence, as the part starting 'Such a good correlation...' is repeating the previous sentence

L3 – maybe add 'in this location' after 'possible' because this does not prove that this method would work everywhere. Also, 'rendered' is not necessary.

L12 – 'flood events'

Page 3:

L5 – either south of Tunisia, or Southern Tunisia

L5 to L11 – description of aims needs rewording. For example, say 'The first aim of this research was to identify...' not 'First aims'. You could also call them 'stages' rather than aims. I think the part about the Pb and Cs dating methods is not needed here.

L14 – remove word 'which' and there is no need for the first 'km'

L15 - add 'and' after axis and remove word 'up'. Km should not be capitalised. Add 'a' before 6m

L20 – is the word 'slobs' the technical word or a local word. If it is local maybe use word peninsula

Page 4:

L1 – L3 – merge these two sentences perhaps. Also change to 'has low demographic pressure'

Page 5:

L4 and L5 – reference these two weather facts. I also do not understand the term 'unequally spatiotemporal repartition' so maybe consider changing

L7 – change to 'is concentrated within 30 days/yr (...'

L8 – typo of Mars not March

L8 – change to 'while in the summer months there are drought conditions'

L10- L13 – Merge first two sentences of this paragraph. Maybe move 'obtained from the Directorate Reseach of Water Resource (ref)' to follow watershed in line 10 and remove rest of the second sentence.

L15 – change to 'precipitation events' not 'precipitations'

L16-17 – the sentence beginning 'During flood events...' needs a reference

Page 6:

L7 – be careful of the spacing between the sample names

L12 – add 'a' before combination

L13-15 – It might be better to merge these two sentences that describe the XRF analysis

Page 9:

L10 – do you mean microscope rather than binocular?

L17 – remove 'of the southern Tunisia' because you are referring to your samples rather than the wider region.

Page 10:

L12 – use 'Constant Flux: Constant Supply (CF:CS)...', so capitalise it and add brackets

Page 11:

L20 – 'To precise the modern contribution of these...' doesn't make sense so reword

L21 – change 'Lagoon' to lagoon

L23 – remove 'South-eastern Tunisia' as you are considering the sanddunes close to your site rather than those throughout this part of Tunisia.

L24 – instead of 'cf chapt 5.1.1' put 'see section 5.1.1.' Page 12:

L2 – be consistent in use of either Fe or Iron. Merge sentences on L1 and L2 Page 13:

L16 – give reference or refer to a figure if it is shown in one.

L24 – use different term than 'heavily precipitating events' e.g. high intensity precipitation events

L 25 – use 'Furthermore' rather than 'on the other hand', as you are not contradicting yourself but rather making another point.

Page 14:

L8 – change to 'discriminate between the...'

L15 – reconsider phrasing of 'have permitted to identify', maybe 'have allowed us to identify'

---

## Referee Comment (RC3) · Anonymous Referee #3 · 4 Jun 2016

The present paper by Affouri et al. deals with the identification of extreme events in the El Bibaine lagoon (SE Tunisia) based mainly on the sediment geochemical composition. The paper itself illustrates an interesting method for identifying the catastrophic events in the sediment records in lagoon settings. The study of lagoon recent sediments and its comparison with sediment series from core shows a high potential of application in the study of past flood activity. However, the result presentation and discussion need, in my opinion, to be improved before publication.

Main comments: 1) The manuscript is mainly "descriptive", focused on the work carried out at this site, but a discussion about a possible application for identification of major floods events in the past is nearly absent. It is limited to a sentence in the Con-

clusions, this aspect deserves to be developed in the Introduction and Discussion. In particular, it would be useful to stress on the importance of lagoon sediment series for reconstructing the flood activity in arid and semi-arid environment, since no other significant (and continuous) sediment series can be easily retrieved in fluvial valleys. The reconstitution of fluvial hydrology is essential for climate modelling.

2) The organization of the text should be revised (see minor remarks below)

3) The use of trace elements is unclear for the Fe content (5.1.2). It is reported that Fe is "totally absent in marine sediments", but looking at Table 2 , the sediments with no Fe are defined "beach". It is not the same; this should be clarified in the text. In addition, the authors should explain the total absence of Fe. Intuitively, it might be related to the beach sand composition (quartz and carbonate debris?), but it needs further explanation in the text.

4) The age model is problematic (5.3). Any possibility that the peak that is interpreted corresponding to the maximum of nuclear essays (1963) matches in reality the Chernobyl nuclear accident? The gap between 20 and 40 cm does not help interpreting correctly. Would it be possible to generate some more measurements?

5) I strongly recommend integrating in the core description the sedimentary structures, if visible, the nature of contacts between different layers (abrupt, gradual, etc.), and the degree of bioturbation. This would allow extending the discussion, taking in count the sedimentary processes at the origin of the coarse-grained layer. The core photo in figure 9 is useless, too small and low resolution. This figure would be much clearer if it includes: a) the granulometric profile and b) any sedimentary structure observed.

Minor remarks 1) Figures 1,2 and 4: too many. I would suggest to reduce, combining the three in one or two (maximum) figures

2) Section 4.1: in which year the core has been retrieved?

3) Section 4.2.1: change "geochemically" in "geochemical"

4) Section 4.2.1: there is an inconsistency between the XRF scanner resolution given in the methods (1 cm) and the data represented in figure 13 (rather 5 cm?)

5) Section 4.2.1: define the granulometric classes (in $\mu$m) that are represented in figure 6 and 13

6) Section 4.2.2: what type of software has been used for the Statistical Analysis?

7) Page 8, lines 9-13: it is surprising that eolian sand grains are angular, is that right? Usually, eolian sands are rather rounded;

8) The pictures in figure 5 are hard to see, the label for S18 is missing on the picture and a micrometric scale is missing for all of them. I would suggest removing the panoramic pictures, leaving and enlarging the pictures showing the microfabric. Important: add a microscale!

9) Figure 5: the S3 seems to be heterogeneous, plurimodal sand. It does not match the particle size distribution shown in figure 6. You should check if the photo really corresponds to the right sample.

10) The section 6.1 should not go in the Discussion, but in the Results

11) Section 6.3: add references to figures

12) Figure 13 seems to be incomplete (FL1 is not shaded)
* * *

---

## Referee Comment (RC4) · Anonymous Referee #4 · 2 Jul 2016

General comments

The paper focuses on the study of historical paleofloods from a high-resolution geo-chemical and sedimentological analysis of a sediment core from El Bibane Lagoon (Southern Tunisia). The paper deals with two main objectives: to identify the main sediment sources within the Tataouine and Mednine watershed areas and to decipher El Bibane lagoon sediment record in order to evidence some historical flood events.

The first part -concerning the sedimentological and geochemical characterization of potential sediment sources from the lagoon watershed -is rather convincing even if the approach remain very classical and not innovative. The second part - related to the analyse of the sediment core from El Bibane Lagoon - successfully evidences that

some fine-grained and Fe and Ti-enriched layers are likely related to historical major flood episodes according to the absolute dating of the core. Once again, the approach is fine even if rather classical, since it demonstrates the potential use of the proposed multi-proxy approach in order to identify paleoflood events in sedimentary sequences.

In general, the objectives mentioned above are somehow reached by the proposed work, but the relationship between these two objectives is not clearly demonstrated in the paper as it is written. Furthermore, these objectives are not clearly stated in the manuscript. Finally, the relationship between these two parts is not further discussed in the manuscript. The main results from the first part need to be thoughtfully used when discussing the sediment record. These particular points need to be improved before publication.

The proposed multi-proxy approach (sedimentology, elemental chemistry, statistical analysis) is adequate. Nevertheless, some major points need to be improved since the interpretations are not fully demonstrated nor convincing as they are presented: for instance, the complete description of the methods should be addressed carefully, the significance of results, including error and limit should be discussed thoughtfully.

Specific comments

Some re-organisation/modifications are recommended in order to improve the manuscript: - Section 2, p4 lines 4-25, these paragraphs may be moved to p3 line 13 - Section 4, move the sentence p5 lines 22-24 to the end of the paragraph p6 line 8 - Section 4, p6 lines 3-4: I guess that the samples from the watershed area were selected before sampling in order to characterize the main potential sediment suppliers to the lagoon. As it is written, it seems that the samples were chosen arbitrarily. I suggest to replace the sentence: "In order to characterize main sources, these surface sediments were subdivided into four regions as:" by " The main potential sediment sources were sampled in order to characterize their sedimentological and chemical signatures as follow: - three samples from the beach area (S1, S2 and S3) representing

the marine source, ten samples (S7 to S16) from Fessi Oued catchment represent-ing the fluvial/river sources, two dune samples (S17 and S18) representing the eolian component. Moreover, three surface samples (S4 to S6) from El Bibane lagoon have been selected to represent present-day sedimentation. - Section 4, Analytical Methods are not properly described. Some important information are missing: o The sediment core lithological description should be detailed, organic-rich clay (mentioned p10 line 2) are mentioned but not shown; o The XRF method should be detailed (apparatus, sample size for discrete surface sediment, error, standard deviation, etc.); o Calibra-tion of XRF data and conversion as percentages; o Grain-size analysis: size/volume of analysed samples, main parameters of the measurements, duration of the measure, reproducibility, error, effect of ultrasound on carbonate shells, etc. - Section 4, Statis-tical analyses: o The whole method should be discussed, including input and output parameters, pre-treatment of data, etc. o Explain why the grain-size parameters were not included in the dataset for PCA? - Section 5, Results, 5.1.1 sediment description. The results should be given properly: o The grain-size parameters should include the mode, median, sorting (when unimodal); o The main sediment class should be men-tioned (clay, cohesive silt, sortable silt, sand); o The photos and observations from figure 5 should be described in much more detailed since they could serve as discrim-inant (for instance the S17 and S18 observations are rather different, explain why?; the eolian particles as quartz are known to have peculiar morphology); o The signifi-cance of variations range should be discussed. The clay fraction varies between 1 and 2% (Figure 9). What about the significance of such a variation? o The results from samples S4 to S6 shown in figure 6 are not discussed within the main text? o The differences between samples S7 and S10 should be emphasized (4 modes for S7 on figure 6 and the coarsest mode for S10 being smaller than 100 $\mu$m according to fig-ure 6, the fine fraction seems over-represented for sample S18, etc.); o The sorting of samples S17 and S18 should be calculated since it appears to be discriminant in term of eolian source. - Section 5, Results, 5.1.2 Distribution of major and trace elements: o The matrix effect (carbonate vs. quartz) should be major: are there any CaCO3 measurements? It would help to evaluate this matrix effect; o p8 lines 17 to 20, the authors described the behaviour of iron: "The iron displays its highest percentages in the Fessi River samples. Lower values characterize the eolian dunes whereas this element is totally absent in marine sediments. This same distribution is also observed for Ti, K and Al...". According to figure 7, I do not agree with this sentence: Fe is indeed maximum in samples from the Fessi River but more generally Fe content is highest in samples from the Mednine and Tataouine catchment areas and from Fessi River. Ti is also highest in samples from Mednine and Tataouine watershed areas, but not in samples from the Fessi River itself (figure 7), whereas K and Al are only higher in samples from the continent compared with marine samples. o p9 lines 4 and 5: Sr concentrations are obviously lower than Ca concentrations! This is not new! o p9 lines 9 to 12, the authors write "these results corroborate the marine origin..." but this is not correct. The samples are marine samples, and the fact that Ca content is high is only consistent with that fact that samples are marine samples with a dominant biogenic component. o p9 lines 13 to 15, this sentence appears to be rather obvious: Si is a major component of alumina-silicate (obviously as silicate) and of quartz (which is pure SiO2); only the eolian samples are characterized by high values, so Si enrichment could be used as a diagnostic for eolian provenance; - Section 5: o 5.2 core description (p9 and 10), this section should rather appear in the material section 4.1; o The description of the grain-size variations is absolutely not sufficient. A complete description (including mode, median, sorting, clay fraction, silt fraction, sand fraction, etc.) should appear (with a dedicated paragraph), as this is absolutely essential for identifying potential paleoflood events! I do not understand why these results do not appear in this section; o The complete description of XRF data (with a dedicated paragraph) should also be included! o The chronological aspect should be discussed before the sedimentological and geochemical results (§5.3 should appear as §5.2.1) : o 5.3 dating: I would like the authors to discuss the impact of major flood events on the sedimentation rate; o I suggest some modification as follow: • §5.2.1 Pb and Cs dating • §5.2.2 grain size/sedimentological results • §5.2.3 XRF results - Section 6, §6.1 PCA o This

paragraph should be included in the result section (§5.1.3 Principal Component Analysis) and should not appear in the discussion o Explain why the PCA does not include grain-size data? o Is this reference adequate? (p11 line 4, Windston et al., 1989) o p11 lines 11-13, "The first component represents therefore the fine fraction od the sediment, which is mainly composed of various types of clay minerals, usually abundant in surface sediments"; this conclusion is not supported by the dataset since the grain-size analyses are not included in the PCA. To my opinion Factor 1 is mainly related to the matrix which is either calcium dominated or alumina-silicate dominated, in other words, Factor 1 depends on nature of the sediment: carbonate (i.e. biogenic component in this particular case) or alumina-silicate (i.e. detrital or terrigenous component); o p11 lines 16-17, the following conclusion "These two factors differentiate hence carbonates and both sand and clay sediments" is once again not fully supported by the PCA analyses since grain-size is not taken into account in the PCA. Actually, the fact that Zr (and Si) likely drives Factor 2 suggests that grain-size should be one forcing factor. I suggest the author to check this conclusion by including grain-size analyses in the PCA input; o I do agree with the conclusion that 1) Ca and Sr may be used to retrace the marine component, 2) Al, Fe, Ti and K may be used to retrace riverine supply and 3) Zr and Si may be used to retrace the eolian contribution, but I am not fully convinced that PCA is useful to demonstrate this commonly accepted statement. - §6 Discussion o The choice of the parameters should be better justified, for instance explain the fact that Zr is not further used? o p12 lines 3-4, Ti/Ca and Fe/Ca ratio appear to reflect solely the marine component. I suggest to use "supply" or "contribution" or "component" instead of "pole" since the paper is not dealing with end-members; o p12 lines 4-9, this part of the discussion is a bit clumsy. It is clear from Figure 12 that these ratios are efficient in discriminating the "continental source" (in this case the eolian source) and "marine source" and the text mentions that "El Bibane lagoon surface sediments are situated between marine and continental sources". But, according to Figure 12, the sediments from El Bibane are in fact situated between the Marine and Fluvial sources, while the pure "eolian" contribution is likely not significant. o §6.3: in this paragraph, it

is not clear if the paleoflood sequences were first identified thanks to their lithological aspects, or if they were identified using both grain-size and elemental ratio? This point should be clarified; o p12 lines 13-14: the sentence "...high content of the clay and silt and high content of the elemental ratio" should be replaced by "...high content in silt and high elemental ratio..." o p13 lines 20-25: the hypothesis of multi-phased flooding is not supported by the data (see figure 13); - Conclusion o p14 lines 4-6: please add "sedimentological and geochemical characterization", change "in order to reconstruct" by "in order to identify the specific signature of paleoflood events"; o p14 lines 10-11: change "...are situated between marine and continental end members" by "are situated between marine and river sources". o p14 line 12, the term "clay" should be omitted since it only represents <2 % of the sediment.

Technical corrections

The English spelling and grammar should be checked carefully.

Please check the consistency of some terms, for instance "Mednine" or "Medenine" should be used consistently throughout the text and figures.

Some sentences/wording are not correct: - p1 line 18: "high content of the clay and silt" is not correct, replaced by "high content in clay and silt" - p1 line 19 (and within the main text): "high content of the elemental ratio" is not appropriate; transform to "high elemental ratio" - p3 line 21: "Tyrrhenian" should be explained (it is explained on page 4, but should be explained on its first appearance) - p4 line 7: Matmata is missing on figure 2 - p5 line 4: change "the number of sunny days may reach 64,4%" by "the number of sunny days may reach 64%" - p5 lines 4-5 : "The rainfall ... annual average that does not exceed 200 mm". This average should be drawn on Figure 3; - etc....

Some of the references (from the references list) are not used within the text: - Prospero et al., 1981 - Raji, 1984 - Torres-Padron et al., 2002

Some references are not correctly used within the main text: - p2 lines 15-16: Becker

et al., 1989 should be replaced by Becker, 1989 - p2 line 20: Noren, 2002 should be replaced by Noren et al., 2002 - p2 lines 22-23: Liu et al., 1993 should be replaced by Liu & Fearn, 1992 - p2 line 23: Donnelly et al., 2007 should be replaced by Donelly and Woodruff, 2007 - p3 line 20: Medhioub, 1981 should be replaced by Medhioub & Perthuist, 1981 - p4 line 3: Pilkey et al., 1989 should be replaced by Pilkey, 1989 - p4 line 2: Bouougri, 2012 should be replaced by Bouougri & Parada, 2012

Figures: - Figures 2 and 4 could be gathered in a unique figure; the bottom insert in figure 4 could be removed; - Figure 2: the reference Ben Haj Ali et al., 1985 is missing in the references list, check the colour variations between Neogene and Paleogene, and between Permian and Permo-Trias; - Figure 3: I suggest to use only diagram, reference is missing; - Figures 7 and 8, I suggest to use distinct symbols for eolian (diamond), marine (square) and river samples; - Figure 9: add some parameters (mode, median, etc.) and specify the considered grain-size fraction (sortable silt, cohesive silt, fine sand or give the size range <63$\mu$m, >63$\mu$m, etc.); - Figure 13 could be associated with Figure 9. Explain the difference between Figure 13(b) and figure 3? Figure 3 could thus be removed.

---

## Editor Comment (EC2) · N. Combourieu Nebout (Editor) · 2 Sep 2016

Dear authors,

We have now recieved the four reviews concerning your paper. Both reviewers made important comments about your manuscript. I am strongly recommending you to reply quickly to both reviews and justify your response to all comments.

You have now to post your replies to all the comments on the discussion forum and explain too how you want to modify your manuscript.

Please also prepare a revised version of your paper accordingly. In the revised version, I would like to see your corrections in track change mode.

[Discussion paper]

[Figure]

Looking forward to reading your replies.

All the very best

Nathalie Combourieu-Nebout

---

## Author Comment (AC1) · 8 Sep 2016

Responses to Reviewer Comments
deciphering ancient flood events in lagoonal deposits, as preserved in a core covering the past century based on a combined chronology using 210Pb and 137Cs data. I must be honest in saying that, if the study is relatively sound and acceptable, I've not been convinced in general by the novelty of the approach, and have in addition several reservations regarding the interpretations (see the Specific Remarks). In particular, most of the results have been presented between the Results chapter and the Discussion, which renders the manuscript confusing and difficult to read. Alternatively some parts of the text have been totally overlooked and would benefit from further consideration/exploration before the manuscript can be accepted. The quality of the figures is overall acceptable, albeit some figures are of very poor graphical quality. The manuscript is not really well written, and should absolutely be revised by a native English before further consideration. I also regret that no tentative comparison with other regional datasets is provided in the Discussion, although I am pretty convinced that such a perspective would help to build a bigger picture of palaeoflood activity regionally. Finally, I do not believe that the manuscript provides the sort of conceptual and fundamental advance in our understanding of the processes and mechanisms governing lagoonal sedimentation and past central/southern Mediterranean climate that has been published elsewhere. For these reasons, I would not recommend this study to be published in Climate of the Past. However I leave this decision to the editorial board, who should appreciate the other reviewers' comments and recommendations.

Specific remarks:

1. Introduction: Page 3, Lines 1-3: Please provide more information dealing with the study of Raji et al., 2014 in Morocco, and show how the outcome of this work is related to the present study. A paragraph was added in the introduction section (page 3 line 1-3):

"Few studies have been undertaken to reconstruct past flood events from lagoon sediments (Raji, 2014). Most of the studies were interested to flooding associated with both hurricanes and tsunamis where overwash deposits preserved within backbarrier

lagoons and salt ponds can provide a means for documenting previous flooding activity. Heavy rain flooding events recorded within these environments are still poorly documented".

Page 3, end of the introduction: I would have appreciated to find here, as a foremost objective of the study, a perspective of data integration with other dataset covering the same time span, at a regional/larger scale.

A paragraph has been added at the end of the introduction:

"Reconstruction of past flood events from sedimentary archives which covers the last century has not been studied in the southern Tunisia. Moreover we show in this study the importance of lagoon sediment series for reconstructing the flood activity in arid and semi-arid environment in an area where no other significant (and continuous) sediment series can be easily retrieved in fluvial valleys".

3. Climate and hydrology

Page 5, lines 10-12: Please check the phrasing of that sentence. This is a regular problem throughout the manuscript, which would highly benefit from a thorough cross reading by native English.

In the revised version we take into account the reviewer proposal:

"The annual precipitations of Medenine and Tataouine stations during the last century were obtained from the Directorate Research of Water Resource (DGRE, 2010, Figure 2). Five major precipitation events were recorded from these two stations (i.e. A.D 1932, A.D 1969, A.D 1979, A.D 1984 and A.D 1995). These events have induced large flood events on the Fessi River watershed (Poncet, 1970; Bonvallot, 1979; Ousslati, 1999; Boujarra et Ktita 2009; Fehri, 2014)".

Page 5, lines 13-16: When you refer to Figure 3, please also introduce the Medenine and Tataouine watersheds here (and not later at the beginning of chapter 4). There is a mistake with the spelling of Medenine on Figure 2.

In the revised version this figure has been corrected accordingly.

4. Materials and Methods

Page 6, lines 11-13: Please provide a general lithological description of core BL12-10, since we are not provided with any information with respect to the sedimentology at that stage.

We agree with the reviewer's comment and we take into account the reviewer suggestion:

"The lithotological description of the first 30 cm core showed coarse-grained layers of siliciclastic sand and shell fragments inter-bedded with organic rich dark grey fine grained sediment (mud) of clay and silt. Three mud layers were identified from 6 to 10 cm, 14 to 18 cm and finally from 26 to 30 cm core depth".

In general the methods are described in an extremely concise way, and would perhaps merit more devotion. The information provided in the present version of the manuscript are indeed very limited (XRF, grain size analysis and age model using 210Pb and 137Cs). Why did you opt for a 1cm-resolution (only) with the XRF data, and not a higher resolution? Is the sediment too homogeneous, thus rendering this perspective not promising? Please elaborate on that.

We agree with the reviewer suggestion. In the revised version this part has been improved accordingly.

"For elemental analyses of the bulk sediment a portable energy dispersive X-Ray fluorescence NITON XL3t was used. This technique delivers fast and accurate elemental analysis results, from a few ppm to percentage. XRF-scanning analyses are done directly on the sediments of the BL12-10 split-core section. The split-core surfaces were first flattened and covered with a thin (4 $\mu$m) Ultralene film to avoid contamination of the measurement prism of the core scanner (Richter et al. 2006). All surface samples were prepared for XRF-bead analysis by powdering and homogenizing of the dried

samples using an agate mortar. The resulting powder was dried for 2 h at 105°C and kept in a desiccator at room temperature. Ca. 4 g of the powdered samples were placed in plastic cups and sealed with Mylar foil (0.4 $\mu$m). The prepared sample cups were placed on the XRF and measured for 120 sec with different filters for the detection of specific elements. Two filters were used with the following adjustments: main measuring 90 s at 10 kV tube voltage with 40 $\mu$A for Al, Si, S, Cl, K, Ca, Ti, Mn, Fe and 30 kV tube voltage for Zn, Br, Sr, Rb, Zr with 40 $\mu$A. The portable XRF scanner (NITON XL3t) has been calibrated and checked on all NITON XRF calibration standards and is certified as "Passed" by Thermo Scientific Portable Analytical inst. Lnc. In our study, the XRF-scan data will be presented as processed intensities expressed in ppm or in percentage. In this study, we choose a 2 cm resolution with the XRF data for two reasons: (1) we have used a field portable XRF scanner core that may not permit a continuous scan analysis. Furthermore the maximum outlet opening of the X-ray generator is 0.7 cm in diameter. Therefore, the maximum resolution is to make a measurement every 0.7 cm; (2) the sediment is not laminated; sediment rearrangement processes by the lagoon bottom currents and bioturbation homogenized the sediment up to few mm to cm thickness. Consequently, increasing the resolution is not necessary. Laser grain-size analyses were achieved with a Beckmann- Coulter LS13320 Particle Size Analyser (Geosciences Montpellier). Grain-size analyses were performed on the BL12-10 sequence with an average interval of 1 cm. For each sample, a small homogeneous amount of sediment was mixed in deionized water then sieved at 1.5 mm diameter before pouring in the Fluid Module of the Particle Sizer until to obtain an optimal obscuration rate between 7 and 12% in the Fraunhofer optical cell. The time of background and sample measurement was set to 90 s and sonication was applied during the measurement of the sample in order to improve the dispersion of fine particles in the fluid. Each sample was measured twice and the good repeatability of measurement was verified according to the statistics from the international standard ISO 13320-1. Dating of sedimentary layers was carried out using 210Pb and 137Cs methods on a centennial timescale. The 137Cs and 210Pbex activities analyses were

performed on the fraction $< 150\mu$m by gamma spectrometry using a CANBERRA Broad Energy Ge (BEGe) detector (CANBERRA BEGe 3825). The sediment was then finely crushed after drying, and transferred into small tubes (diameter 14 mm), and stored for more than 3 weeks to ensure equilibrium between 226Ra and 222Rn. Generally, counting times of 24 to 48 h were required to reach a statistical error of less than 10% for excess 210Pb in the deepest samples and for the 1963 137Cs peak. Activities of 210Pb were determined by integrating the area of the 46.5-keV photo-peak. 226Ra activities were determined from the average of values derived from the 186.2-keV peak of 226Ra and the peaks of its progeny in secular equilibrium with 214Pb (295 and 352 keV) and 214Bi (609 keV). In each sample, the (210Pb unsupported) excess activities were calculated by subtracting the (226Ra supported) activity from the total (210Pb) activity. We then used the Constant Flux/Constant Sedimentation (CFCS) model and the decrease in excess 210Pb to calculate the sedimentation rate (Goldberg, 1963). The uncertainty of the sedimentation rate obtained by this method was derived from the standard error of the linear regression of the CFCS model. 137Cs was studied on the core BL12- 10 in order to assess sediment accumulation rates and chronology of the first 30 centimetres of the core. 137Cs (t1/2 = 30.1 yr) is an anthropogenic radionuclide. It entered the environment in response to atmospheric nuclear tests from 1954 to 1980 AD that induced global fallouts (the first year of atmospheric releases was 1953 AD, whereas the maximum atmospheric production is reached in 1963 AD. 137Cs depth profiles have been extensively used in various environments to assess sediment accumulation rates (Nittrouer et al., 1984; He and Walling, 1996; Radakovitch et al., 1999; Frignani et al., 2004)".

Page 7, lines 11-12: Please rephrase as I do not understand this sentence. Page 7, lines 11-13: I find this introduction for the PCA analyses far too simple!

As suggested by the reviewer we included in the statistical analysis a paragraph about PCA:

"Statistical methods were applied to complete and refine the analysis. Principal Component Analysis (PCA) is widely used statistical techniques in environmental geochemistry. This multivariate approaches is used to reduce the large number of variable that result from XRF analysis. Principal Component Analysis (PCA) was applied to chemical elements in order to distinguish the different sediment sources of surface sediments and link them to the geochemical processes or proprieties. In the present work, the dataset contains 18 samples, each of which includes concentration of 8 elements (Ca, Sr, Fe, K, Al, Ti, Si and Zr). Data are presented in the form of elemental concentration (8 variables). In this study, a statistical analysis was performed using the STATITCF (1987) which is based on variables and it is suitable for identifying the associations of variables with a set of observations. A representation quality of the parameters (positions in the factorial plane) was then performed".

Could you elaborate more on that? For instance, since you use percentage values (both for grain-size and XRF data), have the raw data been square-root transformed, centered and standardized before applying the PCA analysis? This is of great importance regarding the reliability of the results. Please clarify it.

The raw data (in percentage values) have not been square-root transformed, centered and standardized before applying the PCA analysis.

5. Results

Page 8, lines 2-4: On Fig. 6 the distribution of grain sizes appear different and more complex between S7 and S10 (fluvial end-member). For instance, the mode at $100\mu$ is not present on sample S10. Similarly, the mode 20-63 $\mu$ is not really obvious in S10. Is the pattern so tricky when considering other samples from the fluvial component (e.g., S8, S9, S10-S16)? Please comment on that and eventually show more plots for the fluvial components.

In the revised version this part has been modified:

"We showed that the fluvial source has a bi to multimodal distribution with two or even

three modes. In order to obtain the best resolution in the identification of the fluvial source, we choose to use the sediment samples which were collected only along the River Fessi: S9, S10, S12 and S13. These surface sediment samples show a decrease in the mean grain size from upstream to downstream of the River Fessi watershed (Fig .6). The decrease in the mean grain size could be explained by a strong change of the topographic slope around Tataouine. Here, the coarser material is deposited and the finer material is transported away by the river. These finer sediments are deposited in the low plain of the river and in the El Bibane lagoon. Therefore, we suggest that S9 and S10 (collected between Tataouine and the lagoon) characterize our fluvial component in the lagoon. The grain size distribution for S9 is unimodal with a mean grain size around 96 $\mu$m and moderately sorted muddy sand named very coarse silty very fine sand and sample S10 is fine silt with trimodal distribution in 7$\mu$m, 26$\mu$m and 73$\mu$m, and poorly sorted mud sediment type. These characteristics will serve to identify the fluvial source into the lagoon".

What about the 4th group, i.e., the lagoonal samples? It is neither presented so far in the text, nor shown on Figure 5 (although it does on Fig. 6, interestingly). The distribution looks rather complex for this fraction in Fig. 6, and obviously shows a mixture between the different modal distributions (with at least a great contribution of fluvial samples).

The samples S4 and S5 are represented in the figure 6 (see also figure 4). These samples show a grain size distribution similar to those of the fluvial samples.

Page 8, lines 15-24: Here we are provided with XRF data given as percentage values. Please explain how these values have been obtained. Have the semi-quantitative XRF core-scanning data been calibrated by discrete XRF measurements as to determine linear regressions in cross plots and calculate percentage values from scanner data? Please clarify on that, and above all, please show the raw XRF data (in cps) obtained on core BL12-10.

[Figure]

The elemental analyses from XRF measurement were performed in mining type ModCF prolene mode. XRF provide a semi-quantitative measurement which shows directly concentrations in ppm or in percentage values. International powder standards (NIST2702 and NIST2781) were used to assess the analytical error and accuracy of measurement, which are lower than 5% for Ti, Cr, Fe, Zn and Pb, between 5 and 15% for Ca, Mn, As, Rb, Sr, and between ca. 15 and 25% for K and Co.

Another issue: Taken into account the very low ranges of variations (0-1.5% for Fe; 0-0.2% for Ti), how can you be confident with the interpretations (i.e., the discriminations into different environmental pools)?

We distinguished and classified these surface samples into three components. We discriminated these sources by the elemental analysis of Fe and Ti. For example, even though the low ranges of variations we could clearly see that marine source have Fe<2680ppm (equivalent to Fe<0.3%), the aeolien samples have Fe in the range to 3793-4980ppm (0.3<Fe<0.5%) and finally the fluvial samples have Fe in the range to 5350-15250 ppm (0.5<Fe<1.5%). Moreover, the gradient of Fe values from the up-stream of the watershed to the littoral is well pronounced. These results are significant and are higher than the error limits of the analytical measurement. We can assume from these data that the Fe and Ti can easily distinguish the three components (eolian, marine and fluvial).

Page 9, line 5: Please change Fig. 7 into Fig. 8. Done

Page 9, lines 18-19: Please rephrase here, a verb is missing. Done

Page 10, lines 5-6: Do you mean mud or clay layers? Mud is usually enriched in organic matter, whereas clayey sediments are not. What do you mean by mud layers typically composed of clay and silt sediments? By the way, there is no mud shown on Figure 9.

We mean mud. The BL12-10 core showed three dark grey layers of clay and silt (6 to

10 cm, 14 to 18 cm and 26 to 30 cm). These three dark grey layers are mud layers because they are rich in organic matter.

Page 10, lines 6-7: Where do these flood layers appear on Fig. 9? How did you identify it ? I regret that the quality of Fig. 9 is so poor! Please redraw this figure accordingly.

We agree the review and this figure was redrawn.

Page 10, lines 16-18: I do not understand this sentence! It provides a very simplistic explanation for the discrepancies observed between 210Pb and 137Cs data. Did you also measure 241Am throughout core BL12-10, which would help in solving this apparent mismatch?

The 241Am was too low to be measured. We think that the difference between the two methods could be explained by a change of the accumulation rate between the beginning and the last part of the 20 century or due to the low number of 210Pbex measures that do not allow us to use the CFCS model to its optimum. However, the accumulation rate estimated by 210Pbex and 137Cs in this dynamic environment is not too bad (0, 37 cm/y with 137Cs and 0,48 cm/y with 210Pbex).

6. Discussion

Page 11, lines 20-23 and thereafter: I do not understand why the Discussion chapter still contains results/interpretation. The outcome of the PCA analysis should definitely be treated in the Results or Results/Interpretations chapter, but not in the Discussion!! Please modify this accordingly.

In the revised version this part has been modified accordingly.

The Discussion should be the locus where the results are integrated regionally, and at a larger scale, regarding the main scientific question identified in the introduction. Here we are provided with results in the Results chapter, followed by results in the Discussion chapter. See also my comment Page 7, lines 11-13.

Please see reply to Page 7, line 11-13.

Page 11, line 4: I do not understand why this reference is suitable and of interest here. Please check this and correct accordingly.

This reference has been removed

Page 11, lines 13-15: I do not agree that Fe and K (at least) showing negative loadings on Factor 2!!

In the revised version this sentence has been modified. Indeed, Fe and K show a positive loading on Factor2.

Overall, I am not convinced by the application of a PCA analysis here to discriminate between different sources. Please explain why the PCA analysis brings compelling useful evidence for the interpretation of environmental proxies.

PCA was performed on geochemical analysis to distinguish the different sediment components. We applied PCA to regroup these elements and to identify the main factor controlling the chemical compositions of sediments from the catchment and the El Bibane lagoon. Thereafter, elements have been regrouped as fluvial, marine and Aeolian sources with respect to the two factor loading. This method has the particularity to highlight in the same figure the different components. Such figure is easily understandable.

Page 12, lines 3-9: Are these results really unexpected? What do we learn here? Do we really need geochemical proxies, grain-size data and PCA analyses to show that lagoonal sediments are made of a mixture of continental and marine sources? Why this still is presented in the Discussion??

Yes. We think that this approach is necessary to identify extreme events of fluvial, marine or Aeolian sources which could be recorded in the lagoon sedimentary archives. At regional scale, any paleoenvironmental reconstruction would be impossible without this classical approach. The objective of this paper was not to verify that lagoon deposits were a mixture of marine and continental sediments. Our paper deals with the calibration of different environmental tracers for paleoenvironmental reconstructions.

Page 12, lines 12-15: Looking at the data, it is not really obvious that one could define genuine palaeoflood events. How do you discriminate between a background fluvial influence within the lagoon and genuine palaeoenvironmental disruptions (e.g., exceptional flood events recorded in the sediments = eventites)? Is there a threshold to be considered in the data?

The multi-proxy approach using sedimentological and geochemical analysis has permitted the identification of three flood deposits. 210Pb and 137Cs geochronology have been used to date these flood deposits. These dated three flood deposits correspond to three historical heavy precipitation events. These events are extreme events. We suggest that the El Bibane lagoon may record past extreme flood events.

Page 13, lines 17-19: May this alternative explanation account for the apparent discrepancies observed between 210Pb and 137Cs data? Apart from that, if the BL12-10 core consists of a background sedimentation disrupted by occurrences of flood events during the past century, it should definitely be taken into consideration when calculating average sedimentation rates. Did the FL1, FL2 and FL3 flood layers excluded for the estimation of sedimentation rates? If not, this has to be commented.

Numerous studies have used 210Pbex profiles to identify abrupt events. In these studies, the 210Pbex activity versus depth curve is nonlinear and thus cannot be explained by radioactive decay alone. The activity within these disturbed layers is particularly low and is linked to sediment deposits that have been reworked (Arnaud et al., 2002; Dezileau et al., 2006). Thus, the 210Pbex profile may permit us to identify the disturbed areas in a sediment core. In our study it is not the case. The FL2 flood deposit is the most obvious sedimentological and geochemical event. This event is not manifested by a decrease of 210Pbex. This result can be explained by the fact that: 1) the sediment supply from the watershed with the 210Pbex is not deeply mixed with old material.

There is little remobilization of sediment, which is not specious given the thin lateral terraces along the River Fessi. 2) FL2 flood deposit constitutes a series of large floods between 1969 and 1979. This FL2 deposit receives 210Pbex from the watershed over a 10 years period. This study shows that we do not reconstruct a single flood event but rather a succession of events concentrated in a period of time. And 3) it is possible that the bioturbation smooth the profile of 210Pbex. In this case it is difficult to identify disturbed levels in a sediment core.

Moreover, if the FL2 layer represents more than one flood deposit (e.g., 3 floods events as suggested), why do all sedimentological proxies (i.e., Fe/Ca and Ti/Ca, clay+silt fractions) peak only once in FL2 ?

The geochronology of the FL2 flood deposit extends from AD.1965 to AD.1980. Between these dates, two historical extreme flood events are known (AD.1969 and AD.1979) and one flood event of lower magnitude (AD.1972). Only one deposit occurs in the case of the BL12-10 core. Consequently, we assume that this unique flood deposit is linked to these three high precipitation events (i.e. AD.1969, AD.1972 and AD.1979). The sedimentary supply from the different rivers in relationship to these heavy precipitation events has been trapped in the inundation plain, in the Lagoon and probably transported to the Mediterranean Sea through the passes. The sedimentation rate belonging to these events in the lagoon is not very high. Otherwise, these events are sedimentologically and geochemically recorded. Bioturbation and bottom currents in the lagoon have probably smooth the signal. Finally, the three extreme flood events are registered as only one deposit in our sedimentary archive.

What about the 1984 flood recorded in the Tataouine watershed as shown on Figure 3 ? May this correspond to the peaks observed in Fe/Ca and Ti/Ca at the lower end of FL1 (around 10 cm)?

The 1984 flood cannot correspond to the lower end of FL1 flood deposit. Taking into account our age model it is difficult to link FL1 to 1984. Moreover the 1984 flood

event was a weak magnitude event. It is possible that this low magnitude event is not recorded in our sedimentary archive.

References: Many references are listed in the reference list but do not occur in the text. There are listed here below, but please check the reference list in general. Done Beker (1989) is missing in the text. Guelorget et al. (1982) is missing in the text. Plewa et al. (2012) is missing in the text. Prospero et al. (1981) is missing in the text. Raji (2014) is missing in the text. Richter et al. (2006) is missing in the text. Torres-Padron et al. (2002) is missing in the text. Done Figures: Figures 1, 2, 3, 4, 5, 7, 8 are of good visual and graphical quality in general. In contrast, Figures 6, 9, 10 and 11 are of poor quality (Fig. 12 acceptable) and should definitely be improved before the manuscript can be reconsidered. These figures have been modified accordingly.

---

## Author Comment (AC2) · 8 Sep 2016

Responses to Reviewer Comments

We thank the reviewer for his thoughtful comments and suggestions. His comments have improved the manuscript considerably. We have included almost all of the raised suggestions and below we present a point-by-point response to the comments.

Clim. Past Discuss., doi:10.5194/cp-2016-40-RC2, 2016 © Author(s) 2016. CC-BY 3.0 License. Interactive comment on "Extreme flood events reconstruction during the last century in the El Bibane lagoon (Southeast of Tunisia): a Multi-proxy Approach " by A. Affouri et al.

[Figure]

Anonymous Referee #2

General Comments The paper shows that this site has the potential for developing reconstructions of past flood events, as demonstrated by the correspondence between silt layers and twentieth century flood/precipitation events. It is really showing the potential for future reconstructions, rather than giving new data at the moment. I think the paper could do more to emphasise the importance of the work. At the moment the abstract finishes with a statement that hydrological events can be reconstructed using sedimentary archives, which has already been shown elsewhere. Instead I would like to see more emphasis in the introduction, discussion and conclusion about what the wider implications are and why this site in particular may be important. For example: saying whether there are other reconstructions from this region that have reconstructed hydrological changes – and if not, highlighting that the paper shows that this site has the potential to provide this, which could answer questions on the recurrence intervals, magnitude changes etc... The manuscript organization needs improving, as there is some mixing of results, interpretation and discussion, which make it quite confusing at times. In addition I feel the English needs better proof reading before publication to ensure it reads well. I have made some changes to this but not throughout. Finally there are too many figures and some seem unnecessary so should be combined or removed. Specific comments

Page 3, line 1-3 – this should not be a paragraph on its own, so either remove or merge with previous paragraph

A paragraph has been included in the introduction accordingly:

"Few studies have been undertaken to reconstruct past flood events from lagoon sediments (Raji, 2014). Most of the studies were interested to flooding associated with both hurricanes and tsunamis where overwash deposits preserved within backbarrier lagoons and salt ponds can provide a means for documenting previous flooding activity. Heavy rain flooding events recorded within these environments are still poorly documented. Moreover we show in this study the importance of lagoon sediment series for reconstructing the flood activity in arid and semi-arid environment in an area where no other significant (and continuous) sediment series can be easily retrieved in fluvial valleys.

Page 3, line 17 – maybe use the word 'partially' before 'separated from the med...', as it looks on the map to not be completely separate.

We agree with the reviewer remark. We added in the revised version the word "partially".

Page 3, line 18 – are peninsulas 12 km, make it clear if this is the combined length or not

"The Eastern periphery of the EBL is partially separated from the Mediterranean Sea (Gulf of Gabes) by two peninsulas, each of about twelve kilometers long, namely Slob El Gharbi and Slob Ech Chargui (Medhioub, 1979)".

Page 4 – there is an extensive geological description here which I am not sure is necessary. I think cut this down and make it more clear which parts are likely to influence the watershed of the lagoon, and the area on which it is situated.

The Fessi river watershed covers all these geological formations outcropping in southeastern Tunisia. It is necessarily to describe all of these geological formations however we have reduced this part of the manuscript.

Page 5, line 14-15 – this says that figure 3 shows that the precipitation events caused the flood events, however it only shows the precipitation records with no link to flooding. Remove the final part 'causing the flood events' or give more evidence.

We fully agree with the reviewer's suggestion.

Figure 2 shows that the watershed from Fessi River was affected by period of heavy

precipitation events causing floods. Five major precipitation events were recorded from these two stations (i.e. A.D 1932, A.D 1969, A.D 1979, A.D 1984 and A.D 1995). These events have induced large flood events on the Fessi River watershed (Poncet, 1970; Bonvallot, 1979; Ousslati, 1999; Boujarra et Ktita 2009; Fehri, 2014).

Page 7, paragraph 1 – make this paragraph about the dating into a sub section on its own because it doesn't fit in this one about the proxies used. I am not familiar with what 210Pbex is and I think it might be better to refer to it as simply Pb210, as you do in the rest of the paper. If it is necessary it should maybe be defined when it is first mentioned.

Conventionally we used 210Pbex as 210Pbex=210Pb measured- 226Ra. In the revised version, a newly extended sub-section (see section 4.2.2) has been dedicated to the dating. All over the text we have changed 210Pb as 210Pbex.

Page 7, paragraph 2 – as you just used PCA it might be better to start the paragraph with 'Principle Component Analysis was used to understand the relationship....' because this makes the link with what you say in the next sentence. I would also merge the last two sentences in this paragraph.

In the revised version we take into account the reviewer's suggestion: Principal component analysis (PCA)

Page 7, paragraph 3 – the first sentence suggests that you have made the measurements and then grouped them into 3 source areas, whereas I think you have actually taken samples from three different types of location and then present the characteristics of these. Change this first sentence to make it clear that this is what you did.

This part has been changed in the revised version accordingly:

"The collected surface sediment samples have been taken from three different types of location and the characteristics of these samples are then presented".

Page 9, paragraph 1 – in the first two sentences you have used 'sediment' in each

and then described three different origins – make it clear which sediment you are describing. I think in the first sentence you mean the lagoon sediment and in the second sentence perhaps terrestrial sediment, but I am not sure. So make this clear and also reference each sentence. Is the third sentence about ionic radius important to know for understanding the results? I think it needs removing. Finally, the last sentence repeats the results from section 5.1.1., so I think that you could delete lines 11 and 12 (ending the sentence with 'observations') and in brackets put 'see 5.1.1'.

In the revised version we take into account this suggestion.

Page 9, paragraph 2 – the sentence about detrital quartz (line 18-19) does not fit in here and should be removed. The end of the paragraph from line 18-22 is also more of an interpretation and doesn't belong here in the results. Also would need to reference line 20.

We agree with the reviewer comment. We modified the text accordingly.

Page 10, paragraph 1 – first sentence should be in the materials section, as it is about collecting the core – or delete this if it is already described in the methods. The final sentence in this paragraph is an interpretation, and I think would go better in an interpretation section.

We agree with the reviewer comment. We modified the text accordingly.

Page 10, paragraph 2 – I don't think you should chose a Pb210 model based on the downcore distribution of Pb activity (and Pb210 activity often has exponential curve), I think it should be based on understanding of the deposition at the site, for example I think the CF:CS model assumes constant sedimentation, whereas the CRS model does not. Look at the assumptions of this model and then justify it in the methods section, and also make sure that this is referenced as it is not here. In line 10 you say the range is down to 0.1, but in table 3 the lowest is 1.058. Also the Pb210 profile in figure 10 shows a big gap between 21 and 40 cm with no samples dated – would it

be possible to date between these? Without these depths you are not able to show where the equilibrium depth is (where Pb210 activity is around 0). I know that having this depth is important for the CRS model – I am not familiar with the CF:CS model so it may be different, but if possible further dates would help constrain the timing of the lowest flood layer.

The CF:CS model (Golberg, 1963; Krishnaswami et al., 1971) supposes a constant 210Pb flux and a constant sedimentation rate. Although the sedimentation rate in the lagoon is clearly variable, the CF:CS model can still be applied when typical lagoonal conditions prevail (Sabatier et al., 2008). It is the case for BL12-10 all along the core. In a logarithmic diagram, 210Pbex data define a straight regression line which allows us to calculate an average sedimentation rate. It is not necessarily to have intermediate points and to find the 210Pbex activity around 0 with this method. Indeed, this method differs from CRS method which must be performed with continuous measurements all along the sediment core. Furthermore, the accumulation rate estimated by 210Pbex (CFCS model) and 137Cs in this dynamic environment is not too bad (0, 37 cm/y with 137Cs and 0, 48 cm/y with 210Pbex).

Page 11, paragraph 1 – I would remove this section completely and move most of it (line 4-17) to the results (put it in section 5.1.2). The order of the figures would then also change.

We agree with the reviewer. In the revised version this part has been displaced accordingly. In the revised version, a newly extended sub-section (see section 5.1.3) has been dedicated to the PCA.

Page 12, lines 1-9 – the term 'continental source' appears here and it is unclear what this describes, as previously only marine, fluvial and aeolian sources are described. Either change it or explain what is meant. Also what is meant by 'continental pole' and 'marine pole' is unclear, maybe use the 'sediments' or 'sources' instead of 'pole'.

In the revised version we take into account the suggestion of the reviewer.

Page 12, section 6.3 – I think it would help to have a paragraph or a few sentences at the start of this section clarifying the link between the sediment source characteristics and the sediment changes in the lake sediments. For example say that as the sediment from fluvial source was high in X,X,X then these characteristics in the core sediment will be used as a basis for interpreting flood events.

We agree with the reviewer's comment. This section has been changed (see section 6.4) and a paragraph was added in the revised version:

Our multi-proxy analysis on the BL12-10 core shows that sediments with high content of clay+silt, as well as high Fe/Ca and Ti/Ca elemental ratios would be the sedimentological signature of the paleoflood levels identified in the lagoon sequence. Dating of the three most recent flood deposits provides ages of AD 1995 $\pm$ 6, AD 1970 $\pm$ 9, and AD 1945 $\pm$ 9. The results show striking temporal correspondence of these layers to heavy precipitation events recorded in the region since the 20th century. Such a good correlation between flood deposits and historical and instrumental heavy rain precipitation events suggests that flood deposit paleohydrology constitutes a valuable tool for flood assessment in the southeastern Tunisia.

Page 13, line 2 - remove this sentence as it repeats the previous one.

This sentence has been removed accordingly.

Page 13, line 5-6 – I also feel this sentence is repetitive and has no reference, so should be removed

In the revised version this sentence has been removed accordingly.

Page 12 and 13 - This is a very long paragraph so I would split it into three, with new paragraphs at page 12 line 23 ('From our age model...'), at page 13 line 6 ('Using the same approach') and page 13 line 14 ('Finally')

In the revised version we take into account this suggestion.

Page 13, line 11-13 – I am unclear what is meant by 'most of the sediments'. Did these studies show that sediment deposits were left by the 1969 event in particular? It is also not clear which event is being referred to as the latter event, so make it clear which it is.

The geochronology of the FL2 flood deposit extends from AD.1965 to AD.1980. Between these dates, two historical extreme flood events are known (AD.1969 and AD.1979) and one flood event of lower magnitude (AD.1972). Only one deposit occurs in the case of the BL12-10 core. Consequently, we assume that this unique flood deposit is linked to these three high precipitation events (i.e. AD.1969, AD.1972 and AD.1979). The sedimentary supply from the different rivers in relationship to these heavy precipitation events has been trapped in the inundation plain, in the Lagoon and probably transported to the Mediterranean Sea through the passes. The sedimentation rate belonging to these events in the lagoon is not very high. Otherwise, these events are sedimentologically and geochemically recorded. Bioturbation and bottom currents in the lagoon have probably smooth the signal. Finally, the three extreme flood events are registered as only one deposit in our sedimentary archive.

Page 13, line 17-19 – I agree with this statement about the constraints on your pb210 dating, but maybe acknowledge this sooner in the results section

We agree with the reviewer remark.

Figure 1 and 2: these could be combined, as the geological map could be put in place of the lower map on figure 1 which spans the same area and doesn't show much additional information.

We agree with the reviewer. These figures 1 and 2 have been combined as figure 1.

Figure 5: This is interesting but I think it may not be important enough to require a figure. There are a lot of figures, so this might be one to remove. Also the figure caption refers to samples which are not shown on the figure (e.g.S4) so these should

not be mentioned.

As suggested by the reviewer 1, the figure has been modified.

Figure 7 and 8: it is clear from the graph that the values are percentages, so remove this from the caption. These two figures could be combined, as they show the same type of data.

In the revised version we take into account this suggestion. These figures have been combined as figure 6.

Figure 11 and 12: these are important for the interpretation so I think these figures should come earlier (before the figure 9 and 10 showing the core results). I think if the PCA section is moved to 5.1.2 as I suggest above then this should come forward, and figure 12 could also be introduced there as well. The paper might read better if there is a clear first part looking at the catchment sources, followed by the results from the core.

We agree with the reviewer. These figures have been reordered accordingly.

Figure 13: this compares the precipitation and sediment cores using both depth and age. I understand that the pb210 model is not perfect but it might be better to have an additional x-axis on the top graph showing the estimated age, or at least ages marked on at the boundaries of the flood periods so that you can see the timings of these just by looking at the figure without needing to read the caption. Also, in black and white the bar for FL1 does not show up well.

This figure has been changed as suggested by the reviewer (see figure 13).

Table 1: this could go in supplementary information. It is clear on figure 4 where they are all located and in the next table you state the type of locality they are.

In the revised version this table has been removed.

Technical corrections

We thank the reviewer for his help in the text editing. All of the technical corrections have been changed accordingly.

Page 1: Title – 'spanning' might be better word than 'during'. Also, there is an uneven use of capital letters. Changed

Line 10 – flood not floods Changed

Line 11 – 'Recent studies of' not 'Recently, study of' Changed

L12 – 'have enhanced' not 'contributed to enhance' Changed

L14 – I think it should be 'multi-proxy approach' not 'multi-approach'. Also rethink word 'associating' and maybe use 'combining' Changed

L16 – change to 'sediment deposits' Changed

L17 – remove 's' at end of sediments Removed

L22 – 'flood' not 'floods' Changed

L23 – Chronology should not have a capital letter Changed

L22 and L23 – merge these sentences perhaps '...137Cs chronology, and give ages of AD 1995....' Done

Page 2:

L1 and L2 – remove first half of sentence maybe up until suggests and then merge with previous sentence, as the part starting 'Such a good correlation...' is repeating the previous sentence. Removed

L3 – maybe add 'in this location' after 'possible' because this does not prove that this method would work everywhere. Also, 'rendered' is not necessary. Changed L12 – 'flood events' Changed

Page 3:

L5 – either south of Tunisia, or Southern Tunisia

Changed

L5 to L11 – description of aims needs rewording. For example, say 'The first aim of this research was to identify...' not 'First aims'. You could also call them 'stages' rather than aims. I think the part about the Pb and Cs dating methods is not needed here. Changed L14 – remove word 'which' and there is no need for the first 'km' Changed

L15 - add 'and' after axis and remove word 'up'. Km should not be capitalised. Add 'a' before 6m Changed

L20 – is the word 'slobs' the technical word or a local word. If it is local maybe use word peninsula Changed

Page 4: L1 – L3 – merge these two sentences perhaps. Also change to 'has low demographic pressure' Changed

Page 5:

L4 and L5 – reference these two weather facts. I also do not understand the term 'unequally spatiotemporal repartition' so maybe consider changing Changed

L7 – change to 'is concentrated within 30 days/yr (...' Changed

L8 – typo of Mars not March Changed

L8 – change to 'while in the summer months there are drought conditions' Changed

L10- L13 – Merge first two sentences of this paragraph. Maybe move 'obtained from the Directorate Reseach of Water Resource (ref)' to follow watershed in line 10 and remove rest of the second sentence. Changed

L15 – change to 'precipitation events' not 'precipitations' Changed

L16-17 – the sentence beginning 'During flood events...' needs a reference

References added as (Poncet, 1970; Bonvallot, 1979; Ousslati, 1999; Boujarra et Ktita 2009; Fehri, 2014).

Page 6:

L7 – be careful of the spacing between the sample names

Done

L12 – add 'a' before combination Done

L13-15 – It might be better to merge these two sentences that describe the XRF analysis Done

Page 9:

L10 – do you mean microscope rather than binocular?

We mean "Binocular observation"

L17 – remove 'of the southern Tunisia' because you are referring to your samples rather than the wider region. Removed

Page 10:

L12 – use 'Constant Flux: Constant Supply (CF:CS)...', so capitalize it and add brackets Done

Page 11: L20 – 'To precise the modern contribution of these...' doesn't make sense so reword

This sentence has been modified. Ca, Ti and Fe elements have been chosen in order to recognize the contribution of these sources to the surface sediments of the Lagoon (see section 6.3).

L21 – change 'Lagoon' to lagoon Done

L23 – remove 'South-eastern Tunisia' as you are considering the sand dunes close to

your site rather than those throughout this part of Tunisia. Done

L24 – instead of 'cf chapt 5.1.1' put 'see section 5.1.1.' Done

Page 12:

L2 – be consistent in use of either Fe or Iron. Merge sentences on L1 and L2

We agree with the reviewer. In the revised version we take into account this suggestion: "On the other hand, Fe is present as a maximum in the river samples and as a trace element in marine samples".

Page 13:

L16 – give reference or refer to a figure if it is shown in one.

The 1932 flood event registered in south Tunisian historical archives and data (Fehri, 2014).

L24 – use different term than 'heavily precipitating events' e.g. high intensity precipitation events. Done

L 25 – use 'Furthermore' rather than 'on the other hand', as you are not contradicting yourself but rather making another point. Done

Page 14:

L8 – change to 'discriminate between the...' Done

L15 – reconsider phrasing of 'have permitted to identify', maybe 'have allowed us to identify' Done

---

## Author Comment (AC3) · 8 Sep 2016

Responses to Reviewer Comments

We thank the reviewer for his thoughtful comments and suggestions. His comments have improved the manuscript considerably. We have included almost all of the raised suggestions and below we present a point-by-point response to the comments.

Anonymous Referee #3

The present paper by Affouri et al. deals with the identification of extreme events in the El Bibane lagoon (SE Tunisia) based mainly on the sediment geochemical composition. The paper itself illustrates an interesting method for identifying the catastrophic events in the sediment records in lagoon settings. The study of lagoon recent sediments and its comparison with sediment series from core shows a high potential of application in the study of past flood activity. However, the result presentation and discussion need, in my opinion, to be improved before publication.

Main comments:

1) The manuscript is mainly "descriptive", focused on the work carried out at this site, but a discussion about a possible application for identification of major floods events in the past is nearly absent. It is limited to a sentence in the Conclusions; this aspect deserves to be developed in the Introduction and Discussion. In particular, it would be useful to stress on the importance of lagoon sediment series for reconstructing the flood activity in arid and semi-arid environment, since no other significant (and continuous) sediment series can be easily retrieved in fluvial valleys. The reconstitution of fluvial hydrology is essential for climate modelling.

We agree with the reviewer. A paragraph was added in the introduction and in the discussion sections which deal with the importance of lagoon sediments for reconstructing the past flood activity in arid and semi-arid environments:

"Few studies have been undertaken to reconstruct past flood events from lagoon sediments (Raji, 2014). Most of the studies were interested to flooding associated with both hurricanes and tsunamis where overwash deposits preserved within back-barrier lagoons and salt ponds can provide a mean for documenting previous flooding activity. Heavy rain flooding events recorded within these lagoon environments are still poorly documented. Moreover, reconstruction of past flood events from sedimentary archives has been poorly studied in Tunisia. Zielhofer et al. (2014) have used fluvial archives

to reconstruct past fluvial activity in the northern part of Tunisia. However, these sedimentary sequences are often neither continuous nor complete. In our study we tried to reveal the importance of lagoonal archives to reconstruct past flood activities under a semi-arid environment in southern part of Tunisia, an area where significant sedimentary sequences are absent or not continuous in time.

2) The organization of the text should be revised (see minor remarks below)

In the revised version we take into account this suggestion. The organization of the text has been revised.

3) The use of trace elements is unclear for the Fe content (5.1.2). It is reported that Fe is "totally absent in marine sediments", but looking at Table 2, the sediments with no Fe are defined "beach". It is not the same; this should be clarified in the text. In addition, the authors should explain the total absence of Fe. Intuitively, it might be related to the beach sand composition (quartz and carbonate debris?), but it needs further explanation in the text.

Indeed, macroscopic observation of marine sediments revealed that they are composed mainly of quartz grain and shell debris. Fe is associated with clay minerals. These clay minerals are absent in the marine samples. The finer fraction (clay + silt) is minor in these samples because marine waves and currents are strong along the coast. This material is redistributed far away from the littoral. Such characteristics have been observed in other littoral environments (Sabatier, 2008; Raji, 2014).

4) The age model is problematic (5.3). Any possibility that the peak that is interpreted corresponding to the maximum of nuclear essays (1963) matches in reality the Chernobyl nuclear accident? The gap between 20 and 40 cm does not help interpreting correctly. Would it be possible to generate some more measurements?

The Chernobyl nuclear accident has not affected the Southern part of Tunisia and probably the northern part. Therefore, the increase in 137Cs in our sediment core was

associated to 1963.

5) I strongly recommend integrating in the core description the sedimentary structures, if visible, the nature of contacts between different layers (abrupt, gradual, etc.), and the degree of bioturbation. This would allow extending the discussion, taking in count the sedimentary processes at the origin of the coarse-grained layer. The core photo in figure 9 is useless, too small and low resolution. This figure would be much clearer if it includes: a) the granulometric profile and b) any sedimentary structure observed.

We are fully agree with the reviewer's suggestion. This photo was taken at low resolution. In order to improve this figure we added the granulometric profiles and retraced the stratigraphic log.

Minor remarks

1) Figures 1, 2 and 4: too many. I would suggest to reduce, combining the three in one or two (maximum) figures

As suggested by the reviewer, in the revised version we have combined these figures (1, 2 and 4) as Figure 1

2) Section 4.1: in which year the core has been retrieved?

The core has been retrieved in May, 2012.

3) Section 4.2.1: change "geochemically" in "geochemical"

In the revised version we change "geochemically" in "geochemical".

4) Section 4.2.1: there is an inconsistency between the XRF scanner resolution given in the methods (1 cm) and the data represented in figure 13 (rather 5 cm?)

XRF scanner resolution was performed each 2 cm (see table 3 and figure)

5) Section 4.2.1: define the granulometric classes (in $\mu$m) that are represented in figure 6 and 13

We agree with the reviewer's. We define here granulometric classes in $\mu$m.

6) Section 4.2.2: what type of software has been used for the Statistical Analysis?

We used the software STATITCF (1987) (see section 4.2.3)

7) Page 8, lines 9-13: it is surprising that eolian sand grains are angular, is that right? Usually, eolian sands are rather rounded;

The macroscopic observation of our aeolian samples show that they are composed of homogenous dark yellow sand with angular grains; some of them are coated by iron oxide. The quartz grains from the S17 and S18 samples showed angular quartz which could be related to mechanisms of particles transport (proximal vs distal from the source).

8) The pictures in figure 5 are hard to see, the label for S18 is missing on the picture and a micrometric scale is missing for all of them. I would suggest removing the panoramic pictures, leaving and enlarging the pictures showing the microfabric. Important: add a microscale!

As suggested by the reviewer 3, a microscale has been added on the figure.

9) Figure 5: the S3 seems to be heterogeneous, plurimodal sand. It does not match the particle size distribution shown in figure 6. You should check if the photo really corresponds to the right sample.

Indeed, there is a difference between the sedimentary material S3 (Fig. 4) and the grain size distribution (Fig. 5). We explain this difference by the fact that the binocular observation was performed on raw samples whereas Laser Grain size analysis was performed on fraction < 1.5 mm. The fraction >1.5 mm is composed mainly of shell debris which were eliminated by sieving at 1.5 mm before Laser Grain size analysis.

10) The section 6.1 should not go in the Discussion, but in the Results

In the revised version we take into account this suggestion:

[Figure]

The section 6.1 was displaced in the Result section (see section 5.1.3)

11) Section 6.3: add references to figures

As suggested by the reviewer we added a reference to figures.

12) Figure 13 seems to be incomplete (FL1 is not shaded)

We thank the reviewer for his remark. FL1 has been added in the Figure 13.

---

## Author Comment (AC4) · 8 Sep 2016

Responses to Reviewer Comments

We thank the reviewer for his thoughtful comments and suggestions. His comments have improved the manuscript considerably. We have included almost all of the raised suggestions and below we present a point-by-point response to the comments.

Clim. Past Discuss., doi:10.5194/cp-2016-40-RC4, 2016 © Author(s) 2016. CC-BY 3.0 License. Interactive comment on "Extreme flood events reconstruction during the last century in the El Bibane lagoon (Southeast of Tunisia): A Multi-proxy Approach" by A. Affouri et al. Anonymous Referee #4 General

comments

The paper focuses on the study of historical paleofloods from a high-resolution geo-chemical and sedimentological analysis of a sediment core from El Bibane Lagoon (Southern Tunisia). The paper deals with two main objectives: to identify the main sediment sources within the Tataouine and Mednine watershed areas and to decipher El Bibane lagoon sediment record in order to evidence some historical flood events. The first part -concerning the sedimentological and geochemical characterization of potential sediment sources from the lagoon watershed -is rather convincing even if the approach remain very classical and not innovative. The second part - related to the analyse of the sediment core from El Bibane Lagoon - successfully evidences that some fine-grained and Fe and Ti-enriched layers are likely related to historical major flood episodes according to the absolute dating of the core. Once again, the approach is fine even if rather classical, since it demonstrates the potential use of the proposed multi-proxy approach in order to identify paleoflood events in sedimentary sequences. In general, the objectives mentioned above are somehow reached by the proposed work, but the relationship between these two objectives is not clearly demonstrated in the paper as it is written. Furthermore, these objectives are not clearly stated in the manuscript. Finally, the relationship between these two parts is not further dis-cussed in the manuscript. The main results from the first part need to be thought-fully used when discussing the sediment record. These particular points need to be improved before publication. The proposed multi-proxy approach (sedimentology, el-emental chemistry, statistical analysis) is adequate. Nevertheless, some major points need to be improved since the interpretations are not fully demonstrated nor convincing as they are presented: for instance, the complete description of the methods should be addressed carefully, the significance of results, including error and limit should be discussed thoughtfully.

Specific comments

Some re-organisation/modifications are recommended in order to improve the

manuscript: - Section 2, p4 lines 4-25, these paragraphs may be moved to p3 line 13

We are fully agree with the reviewer's proposal. In the revised version we take into account this suggestion and these paragraphs have been moved.

- Section 4, move the sentence p5 lines 22-24 to the end of the paragraph p6 line 8

This sentence has been moved in the revised version.

- Section 4, p6 lines 3-4: I guess that the samples from the watershed area were selected before sampling in order to characterize the main potential sediment suppliers to the lagoon. As it is written, it seems that the samples were chosen arbitrarily. I suggest to replace the sentence: "In order to characterize main sources, these surface sediments were subdivided into four regions as:" by " The main potential sediment sources were sampled in order to characterize their sedimentological and chemical signatures as follow: - three samples from the beach area (S1, S2 and S3) representing the marine source, ten samples (S7 to S16) from Fessi Oued catchment representing the fluvial/river sources, two dune samples (S17 and S18) representing the eolian component. Moreover, three surface samples (S4 to S6) from El Bibane lagoon have been selected to represent present-day sedimentation.

In the revised version this part has been replaced.

- Section 4, Analytical Methods are not properly described. Some important information are missing: o The sediment core lithological description should be detailed, organic-rich clay (mentioned p10 line 2) are mentioned but not shown;

In the revised version, we describe the analytical methods in detail (see section 4.2)

o The XRF method should be detailed (apparatus, sample size for discrete surface sediment, error, standard deviation, etc.); Calibration of XRF data and conversion as percentages;

We agree the reviewer and we take into account this suggestion in the revised version:

"For elemental analyses of the bulk sediment a portable energy dispersive X-Ray fluorescence NITON XL3t was used. This technique delivers fast and accurate elemental analysis results, from a few ppm to percentage. XRF-scanning analyses are done directly on the sediments of the BL12-10 split-core section. The split-core surfaces were first flattened and covered with a thin (4 $\mu$m) Ultralene film to avoid contamination of the measurement prism of the core scanner (Richter et al., 2006). All surface samples were prepared for XRF-bead analysis by powdering and homogenizing of the dried samples using an agate mortar. The resulting powder was dried for 2 h at 105°C and kept in a desiccator at room temperature. Ca. 4 g of the powdered samples were placed in plastic cups and sealed with Mylar foil (0.4 $\mu$m). The prepared sample cups were placed on the XRF and measured for 120 sec with different filters for the detection of specific elements. Two filters were used with the following adjustments: main measuring 90 s at 10 kV tube voltage with 40 $\mu$A for Al, Si, S, Cl, K, Ca, Ti, Mn, Fe and 30 kV tube voltage for Zn, Br, Sr, Rb, Zr with 40 $\mu$A. The portable XRF scanner (NITON XL3t) has been calibrated and checked on all NITON XRF calibration standards and is certified as "Passed" by Thermo Scientific Portable Analytical inst. Lnc. In our study, the XRF-scan data will be presented as processed intensities expressed in ppm or in percentage. The elemental analyses from XRF measurement were performed each 2 cm step in mining type ModCF prolene mode. These data show directly concentrations in ppm or percentage values. This is a semi-quantitative measurement. International powder standards (NIST2702 and NIST2781) were used to assess the analytical error and accuracy of measurement, which are lower than 5% for Ti, Cr, Fe, Zn, Pb, between 5 and 15% for Ca, Mn, As, Rb, Sr, and between ca. 15 and 25% for K and Co".

o Grain-size analysis: size/volume of analysed samples, main parameters of the measurements, duration of the measure, reproducibility, error, effect of ultrasound on carbonate shells, etc.

We agree the reviewer and this part have been modified:

[Figure]

"Laser grain-size analyses were achieved with a Beckmann- Coulter LS13320 Particle Size Analyser (Geosciences Montpellier). Grain-size analyses were performed on the BL12-10 sequence with an average interval of 1 cm. For each sample, a small homogeneous amount of sediment was mixed in deionized water then sieved at 1.5 mm diameters before pouring in the Fluid Module of the Particle Sizer until to obtain an optimal obscuration rate between 7 and 12% in the Fraunhofer optical cell. The time of background and sample measurement was set to 90 s and sonication was applied during the measurement of the sample in order to improve the dispersion of fine particles in the fluid. Each sample was measured twice and the good repeatability of measurement was verified according to the statistics from the international standard ISO 13320-1. GRADISTAT program version 4.0 (Blott, 2000) was used for grain size statistical analysis. Sample statistics are calculated using the Method of Moments in Microsoft Visual Basic programming language: mean, mode(s), sorting (standard deviation), skewness, kurtosis, D10, D50, D90, D90/D10, D90-D10, D75/D25 and D75-D25. Grain size parameters are calculated arithmetically and geometrically (in microns) and logarithmically (using the phi scale) (Krumbein and Pettijohn, 1938). Linear interpolation is also used to calculate statistical parameters by the Folk and Ward (1957) graphical method and derive physical descriptions (such as "very coarse sand" and "moderately sorted".

- Section 4, Statistical analyses:

o The whole method should be discussed, including input and output parameters, pre-treatment of data, etc.

The whole method has been concisely and suitably described in the section 4.2.3 in order to shorten the manuscript.

We are fully agree with the reviewer's suggestion. See paragraph below which has been added in the statistical analyses section:

"Statistical methods were applied to complete and refine the analysis. Principal Component Analysis (PCA) is widely used statistical techniques in environmental geochemistry. This multivariate approaches is used to reduce the large number of variable that result from XRF analysis. Principal Component Analysis (PCA) was applied to chemical elements in order to distinguish the different sediment sources of surface sediments and link them to the geochemical processes or proprieties. In the present work, the dataset contains 18 samples, each of which includes concentration of 8 elements (Ca, Sr, Fe, K, Al, Ti, Si and Zr). Data are presented in the form of elemental concentration (8 variables). In this study, a statistical analysis was performed using the STATITCF (1987) which is based on variables and it is suitable for identifying the associations of variables with a set of observations. A representation quality of the parameters (positions in the factorial plane) was then performed".

o Explain why the grain-size parameters were not included in the dataset for PCA?

The grain size parameters were not been included in the dataset for PCA because we think that the granulometric profiles were sufficient to discriminate the different sediment sources (aeolien, fluvial and marine). Furthermore the PCA application to grain size is redundant.

- Section 5, Results,

5.1.1 sediment description.

The results should be given properly:

o The grain-size parameters should include the mode, median, sorting (when uni-modal);

We are agree with reviewer and we take into account this suggestion in the revised version Grain size parameters have been calculated using the GRADISTAT program (see tables 1 and 3)

o The main sediment class should be mentioned (clay, cohesive silt, sortable silt, sand);

In the revised version, please see tables 1 and 3 below which added.

o The photos and observations from figure 5 should be described in much more detailed since they could serve as discriminant (for instance the S17 and S18 observations are rather different, explain why?; the eolian particles as quartz are known to have peculiar morphology);

We think that the difference between S17 and S18 could be explained by a different mechanism of eolian abrasion/transport. There is probably a short transport of aeolian material from the source into S17 dune fields and a long transport of aeolian material from the source into the S18 dune fields. This interpretation is supported by the sorting character of the dune sands from each of these two sites (very well sorted to well sorted sands respectively). Moreover, sources are in the proximity of alluvial deposits of the River Fessi and Matmata loess in the Southeast of Tunisia. This could also explain a part of these differences.

o The significance of variations range should be discussed. The clay fraction varies between 1 and 2% (Figure 9). What about the significance of such a variation?

The BL12-10 core was retrieved in the El Bibane Lagoon from the nearest part of the Fessi River delta. Surface sediments samples show that the finer fraction $<2\mu$m (clay) vary between 1 and 2%. This low content could be explained by the redistribution of this finer material by hydrodynamic waves and bottom currents in the lagoon. The fine fraction is redistributed by bottom currents and transported into the middle part of the Lagoon and/or toward the sea. Medhioub (1979) showed that the clay fraction do not exceed 2% in the surface sediment of El Bibane Lagoon around Fessi river delta. Our results corroborate this observation.

o The results from samples S4 to S6 shown in figure 6 are not discussed within the main text?

We agree the reviewer. In the revised version, a paragraph has been added:

[Figure]

"The El Bibane Lagoon surface sediments samples S4, S5 and S6 are characterized by multimodal grain size distribution (Fig. 5). The grain size distributions of sample S4 show polymodal very poorly sorted sandy mud named very fine sandy very coarse silt with trimodal distribution at 154$\mu$m, 31$\mu$m and 96$\mu$m. The sample S5 is unimodal, with mode in 116$\mu$m, moderately sorted type named very coarse silty fine sand sediment with a muddy sand texture (Folk, 1954; Folk and Ward, 1957). The sample S6 is very coarse silty very fine sand sediment, with a bimodal distribution in 106$\mu$m and 429$\mu$m, poorly sorted muddy sand".

o The differences between samples S7 and S10 should be emphasized (4 modes for S7 on figure 6 and the coarsest mode for S10 being smaller than 100 $\mu$m according to figure 6, the fine fraction seems over-represented for sample S18, etc.);

We added a new paragraph as well as a new figure in the text:

"In order to obtain the best resolution in the identification of the fluvial source, we choose to use the sediment samples which were collected only along the River Fessi: S9, S10, S12 and S13. These surface sediment samples show a decrease in the mean grain size from upstream to downstream of the River Fessi watershed (Fig .6). The decrease in the mean grain size could be explained by a strong change of the topographic slope around Tataouine. Here, the coarser material is deposited and the finer material is transported away by the river. These finer sediments are deposited in the low plain of the river and in the El Bibane lagoon. Therefore, we suggest that S9 and S10 (collected between Tataouine and the lagoon) characterize our fluvial component in the lagoon. The grain size distribution for S9 is unimodal with a mean grain size around 96 $\mu$m and moderately sorted muddy sand named very coarse silty very fine sand and sample S10 is fine silt with trimodal distribution in 7$\mu$m, 26$\mu$m and 73$\mu$m, and poorly sorted mud sediment type. These characteristics will serve to identify the fluvial source into the lagoon".

o The sorting of samples S17 and S18 should be calculated since it appears to be

discriminant in term of eolian source.

In the revised version we take into account this suggestion:

"Unimodal distributions in 116$\mu$m characterize the aeolien samples S17 and S18. These samples are very well sorted (S17) and well sorted (S18) very fine sand".

- Section 5, Results,

5.1.2 Distribution of major and trace elements:

o The matrix effect (carbonate vs. quartz) should be major: are there any CaCO3 measurements? It would help to evaluate this matrix effect;

The results could be affected by matrix effect, sediment humidity, etc. Even though these effects exist, they have not major influence on the results. We have performed numerous XRF measurements on different standards. Please, see paragraph added in the revised version:

"The elemental analyses from XRF measurement were performed in mining type ModCF prolene mode. These data show directly concentrations in ppm or percentage values. This is a semi-quantitative measurement. International powder standards (NIST2702 and NIST2781) were used to assess the analytical error and accuracy of measurement, which are lower than 5% for Ti, Cr, Fe, Zn, Pb, between 5 and 15% for Ca, Mn, As, Rb, Sr, and between ca. 15 and 25% for K and Co".

o p8 lines 17 to 20, the authors described the behaviour of iron: "The iron displays its highest percentages in the Fessi River samples. Lower values characterize the eolian dunes whereas this element is totally absent in marine sediments. This same distribution is also observed for Ti, K and Al: : :". According to figure 7, I do not agree with this sentence: Fe is indeed maximum in samples from the Fessi River but more generally Fe content is highest in samples from the Mednine and Tataouine catchment areas and from Fessi River. Ti is also highest in samples from Mednine and Tataouine watershed areas, but not in samples from the Fessi River itself (figure 7), whereas K

and Al are only higher in samples from the continent compared with marine samples.
o p9 lines 4 and 5: Sr concentrations are obviously lower than Ca concentrations! This
is not new!

We are fully agree with the reviewer's remark. In the revised version this part has been
modified

o p9 lines 9 to 12, the authors write "these results corroborate the marine origin: : :"
but this is not correct. The samples are marine samples, and the fact that Ca content is
high is only consistent with that fact that samples are marine samples with a dominant
biogenic component.

In the revised version this part has been modified:

"Ca in the marine samples is high. The high percentage of Ca in these samples is
related to both the significant presence of biogenic material, but also probably the
precipitation of authigenic carbonate."

o p9 lines 13 to 15, this sentence appears to be rather obvious: Si is a major component
of alumina-silicate (obviously as silicate) and of quartz (which is pure SiO2); only the
eolian samples are characterized by high values, so Si enrichment could be used as a
diagnostic for eolian provenance;

- Section 5:

o 5.2 core description (p9 and 10), this section should rather appear in the material
section 4.1;

We change this section in the revised version (See section 4.2).

o The description of the grain-size variations is absolutely not sufficient. A complete
description (including mode, median, sorting, clay fraction, silt fraction, sand fraction,
etc.) should appear (with a dedicated paragraph), as this is absolutely essential for
identifying potential paleoflood events! I do not understand why these results do not

appear in this section;

We added a complete description of the grain-size variations for the core BL12-10:

"The sediment sequence from El Bibane lagoon presented in this study come from the core BL12-10 retrieved from the nearest part of the delta of Fessi River in May 2012 (Fig. 3). The lithotological description of the first 30 cm core showed coarse-grained layers of siliciclastic sand and shell fragments inter-bedded with organic rich dark grey fine grained sediment (mud) of clay and silt. Three mud layers were identified from 6 to 10 cm, 14 to 18 cm and finally from 26 to 30 cm core depth. The high-resolution grain-size analysis of BL12-10 core showed several thin, fine grained and sand sediments layers (Fig. 8). The more prominent mud layers are typically composed of clay and silt sediments. Grain size parameters are calculated by Statistical analysis (GRADISTAT program version 4.0; Blott, 2000) and the nomenclature of grain size classifications follows Folk and Ward (1975). Analysis of BL12-10 samples for sediment grain size demonstrated that sediments were composed of muddy sand as a mixture of fine and medium grains (e.g. very coarse silty very fine sand). The BL12-10 core is dominated by the bimodal and trimodal grain size distributions. These distributions were labeled as very coarse silty to very fine sand, poorly to very poorly sorted, fine skewed with leptokurtic distribution (Table 3)"

o The chronological aspect should be discussed before the sedimentological and geo-chemical results (§5.3 should appear as §5.2.1) :

For the coherence of the manuscript we prefer to discuss the core description before the chronological data because these latter were studied on the sedimentary sequences. For this reason the section "core description" must appear before.

o 5.3 dating: I would like the authors to discuss the impact of major flood events on the sedimentation rate;

The examination of the 210Pbex profile shows that the FL2 was characterized, more

or less, by constant 210Pbex values. This suggests that FL2 represents a succession of extreme flood events close in time. However, the low measurement resolution as well as the bioturbation could smooth the signal. That is why it is difficult to discuss accurately in this paper the effect of major flood events on the sedimentation rate. This could be a perspective for a future work at high resolution.

o I suggest some modification as follow: âËŸA ′c §5.2.1 Pb and Cs dating âËŸA ′c §5.2.2 grain size/sedimentological results âËŸA′c §5.2.3 XRF results

In the revised version, this section has been modified:

"5.2 Core BL12-10

5.2.1. Core description and grain size analysis

5.2.2. 210Pb and 137Cs dating"

o The complete description of XRF data (with a dedicated paragraph) should also be included!

We do not discuss in this paper about all XRF elemental data. We believe this is not necessary. It will take a whole page to discuss and the article is already long. We only discuss the geochemical ratios Fe/Ca and Ti/Ca of surface samples. These geochemical ratios have permitted the characterization of the three sediment sources (aeolian, marine and fluvial) and have been applied to the BL12-10 core samples. This approach was used to identify the variation of the sediment source supply to the lagoon and the past flood events in BL12-10 core.

- Section 6, §6.1 PCA

o This paragraph should be included in the result section (§5.1.3 Principal Component Analysis) and should not appear in the discussion

As suggested by the reviewer 4, this section has been moved as §5.1.3 in the result section.

o Explain why the PCA does not include grain-size data?

In this study, the grain size parameters were not included in the dataset for PCA because we think that the granulometric profiles were sufficient to discriminate the different sediment sources (aeolien, fluvial and marine).

o Is this reference adequate? (p11 line 4, Windston et al., 1989)

We agree with the reviewer's remark. This reference has been removed in the revised version

o p11 lines 11-13, "The first component represents therefore the fine fraction of the sediment, which is mainly composed of various types of clay minerals, usually abundant in surface sediments"; this conclusion is not supported by the dataset since the grain-size analyses are not included in the PCA. To my opinion Factor 1 is mainly related to the matrix which is either calcium dominated or alumina-silicate dominated, in other words, Factor 1 depends on nature of the sediment: carbonate (i.e. biogenic component in this particular case) or alumina-silicate (i.e. detrital or terrigenous component); o p11 lines 16-17, the following conclusion "These two factors differentiate hence marines carbonates to continental components" is once again not fully supported by the PCA analyses since grain-size is not taken into account in the PCA. Actually, the fact that Zr (and Si) likely drives Factor 2 suggests that grain-size should be one forcing factor. I suggest the author to check this conclusion by including grain-size analyses in the PCA input;

We agree the reviewer that grain-size is not taken into account in the PCA. Each sample which has been analyzed by XRF and PCA has been analyzed by Laser granulometry. Therefore, we could associate watershed samples to specific granulometric fractions. i.e samples from the watershed which were collected in the River Fessi. These samples are characterized by multi-modal grain size distribution, whereas, those from the Aeolian source show unimodal distribution.

[Figure]

o I do agree with the conclusion that 1) Ca and Sr may be used to retrace the marine component, 2) Al, Fe, Ti and K may be used to retrace riverine supply and 3) Zr and Si may be used to retrace the eolian contribution, but I am not fully convinced that PCA is useful to demonstrate this commonly accepted statement. - §6 Discussion o The choice of the parameters should be better justified, for instance explain the fact that Zr is not further used?

Statistical analyses of geochemical data have permitted to characterise and to distinguish the different sediment components around El Bibane lagoon. Ca, Ti and Fe elements have been chosen in order to recognize the contribution of these sources to the surface sediments of the Lagoon. Zr is present in the three components. In our region, Zr concentrations are not sufficiently discriminant between these different sources. o p12 lines 3-4, Ti/Ca and Fe/Ca ratio appear to reflect solely the marine component. I suggest to use "supply" or "contribution" or "component" instead of "pole" since the paper is not dealing with end-members;

We agree with the reviewer: we used "contribution" or "component" instead "pole"

o p12 lines 4-9, this part of the discussion is a bit clumsy. It is clear from Figure 12 that these ratios are efficient in discriminating the "continental source" (in this case the eolian source) and "marine source" and the text mentions that "El Bibane lagoon surface sediments are situated between marine and continental sources". But, according to Figure 12, the sediments from El Bibane are in fact situated between the Marine and Fluvial sources, while the pure "eolian" contribution is likely not significant.

We are fully agree with the reviewer's suggestion. In the revised version this has been modified.

o §6.3: in this paragraph, it is not clear if the paleoflood sequences were first identified thanks to their lithological aspects, or if they were identified using both grain-size and elemental ratio? This point should be clarified;

We agree with the reviewer. The paleoflood sequences were first identified using both grain-size and elemental ratio. This paragraph has been changed as:

"The BL12-10 core shows different mud layers (clay and silt mixture) preserved in the core which seems to be flood layers, i.e., coming from fluvial incursions during intense flood events. Multiproxy analysis on these mud layers show that they are characterized by high content in clay+silt, as well as high Fe/Ca and Ti/Ca elemental ratios which represent the sedimentological origin of the watershed. Consequently, these layers register past flood activities of the River Fessi".

o p12 lines 13-14: the sentence ": : :high content of the clay and silt and high content of the elemental ratio" should be replaced by ": : :high content in silt and high elemental ratio: : :"

In the revised version we take into account this suggestion and this sentence has been replaced.

o p13 lines 20-25: the hypothesis of multi-phased flooding is not supported by the data (see figure 13);

The geochronology of the FL2 flood deposit extends from AD.1965 to AD.1980. Between these dates, two historical extreme flood events are known (AD.1969 and AD.1979) and one flood event of lower magnitude (AD.1972). Only one deposit occurs in the case of the BL12-10 core. Consequently, we assume that this unique flood deposit is linked to these three high precipitation events (i.e. AD.1969, AD.1972 and AD.1979). The sedimentary supply from the different rivers in relationship to these heavy precipitation events has been trapped in the inundation plain, in the Lagoon and probably transported to the Mediterranean Sea through the passes. The sedimentation rate belonging to these events in the lagoon is not very high. Otherwise, these events are sedimentologically and geochemically recorded. Bioturbation and bottom currents in the lagoon have probably smooth the signal. Finally, the three extreme flood events are registered as only one deposit in our sedimentary archive.

- Conclusion

o p14 lines 4-6: please add "sedimentological and geochemical characterization", change "in order to reconstruct" by "in order to identify the specific signature of paleoflood events";

In the revised version we take into account the reviewer suggestion.

o p14 lines 10-11: change ": : :are situated between marine and continental end members" by "are situated between marine and river sources".

We agree with the reviewer. The continental end members have been change in the revised version.

o p14 line 12, the term "clay" should be omitted since it only represents <2 % of the sediment.

We prefer to keep this term even if it represents only 2% or less because it defines the contribution of the watershed.

Technical corrections

The English spelling and grammar should be checked carefully. Please check the consistency of some terms, for instance "Mednine" or "Medenine" should be used consistently throughout the text and figures. Done

Some sentences/wording are not correct: - p1 line 18: "high content of the clay and silt" is not correct, replaced by "high content in clay and silt"

As suggested by the reviewer, some sentences/wording has been corrected.

- p1 line 19 (and within the main text): "high content of the elemental ratio" is not appropriate; transform to "high elemental ratio"

In the revised version, this sentence has been transformed.

- p3 line 21: "Tyrrhenian" should be explained (it is explained on page 4, but should be

explained on its first appearance)

Done (See section.2)

- p4 line 7: Matmata is missing on figure 2

Matmata has been added in Fig. 2

- p5 line 4: change "the number of sunny days may reach 64,4%" by "the number of sunny days may reach 64%" Done

- p5 lines 4-5 : "The rainfall : : : annual average that does not exceed 200 mm". This average should be drawn on Figure 3; - etc.: : : Done

Some of the references (from the references list) are not used within the text:

- Prospero et al., 1981 - Raji, 1984 - Torres-Padron et al., 2002 Done

Some references are not correctly used within the main text: Done

- p2 lines 15-16: Becker et al., 1989 should be replaced by Becker, 1989 Done

- p2 line 20: Noren, 2002 should be replaced by Noren et al., 2002 Done.

- p2 lines 22-23: Liu et al., 1993 should be replaced by Liu & Fearn, 1992 Done

- p2 line 23: Donnelly et al., 2007 should be replaced by Donelly and Woodruff, 2007. Done

- p3 line 20: Medhioub, 1981 should be replaced by Medhioub & Perthuist, 1981 Done

- p4 line 3: Pilkey et al., 1989 should be replaced by Pilkey, 1989 Done

- p4 line 2: Bouougri, 2012 should be replaced by Bouougri & Parada, 2012 Done

Figures:

- Figures 2 and 4 could be gathered in a unique figure; the bottom insert in figure 4 could be removed; Done

- Figure 2: the reference Ben Haj Ali et al., 1985 is missing in the references list, check the colour variations between Neogene and Paleogene, and between Permian and Permo-Trias; Done

- Figure 3: I suggest to use only diagram, reference is missing; Done

- Figures 7 and 8, I suggest to use distinct symbols for eolian (diamond), marine (square) and river samples; Done

- Figure 9: add some parameters (mode, median, etc.)  and specify the considered grain-size fraction (sortable silt, cohesive silt, fine sand or give the size range <63_m, >63_m, etc.); Done

- Figure 13 could be associated with Figure 9.  Explain the difference between Figure 13(b) and figure 3? Figure 3 could thus be removed.

As suggested by the reviewer, figure 13 b has been changed as figure 2.

––––––––––––––––––––––––––––

---

## Referee Report (RR1)

[revised manuscript text omitted]

---

## Author Response (AR2)

**Responses to Reviewer Comments**

We thank the reviewer for his thoughtful comments and suggestions. We have included almost all of the raised suggestions and below we present a point-by-point response to the comments.

**Report #1**

I have revised the paper for the second time, I find it ameliorated with respect to the previous version.

However, I would like to address some more suggestions:

- Inaccuracies are still present in the text, I have tried to identify those but I could miss some. Please look carefully at the text;

-concerning the material and data, one table is missing: you should provide the list of the interface core sites, their lengths and collecting depths, with geographical coordinates. However, three tables are already given, please consider to provide this type of data in the supplementary material

- Check the correspondence between "Results" and "Discussion". For instance, in the discussion related to the short core, you use element ratios that you have not presented before in the "Results" Other minor details can be found in the annotated manuscript (pdf).

As last point, I would like to rise a major issue: this work can be useful for application in the paleo-flood reconstructions or extreme events records, certainly. However, the discussion is really poor in this respect (lines 18-20, pag. 17). I would expect a larger discussion on the previous work in Tunisian and Mediterranean regions (arid or not), the approach known in this field, the results obtained and found in the previous literature and the contribution of this work for making progress in this field.

Page 3 Line 5

We agree with the reviewer and the sentence" However, these sedimentary sequences are often continuous nor complete." is removed

Page 3 Line 7

As suggested, we removed the sentence "an area where significant sedimentary sequences are absent or not continuous in time"

Page 3 Line 8-9

We agree with the reviewer. This sentence is rephrased: "In this study we tried to reveal the importance of lagoonal archives to reconstruct past flood activities under a semi-arid environment in southern part of Tunisia, studying the paleo-floods from high resolution geochemical and sedimentogical analyses"

Page 3 Line 11-13

We modify the sentence as proposed:

"The second aim was to reconstruct flood events from the lagoonal archives during the last century".

Page 3 Line 17

We agree with the reviewer and we take into account his suggestion:

"The Djeffara (Inner domain) and the Dahar (Outer domain)"

Page 3 Line 20

We replace "belonging" by "belongs".

Page 3 line 21

The term "outcrop" is removed.

Page 3 Line 22

We agree with the reviewer. "Fig. 1B" was added in the paragraph.

Page 4 Line 10

We thank the reviewer for his comment and a sentence was added:

"(under marine isotopic stage 5e: last interglacial)".

Page 4 Line 14

A box around El Bibane lagoon is added in the Figure 1.

Page 4 Line 18

El Gharbi (western) and Ech Chargui (eastern) were added in the Figure 1.

Page 5 Line 14

Figure.2 is replaced by Fig. 2.

Page 5 Line 23-24

A table was added in the supplementary data as asked.

Page 6 Line 9-10

We agree with the reviewer and a sentence was added:

"S6 represents the surface sediment sample of a lagoon sediment core (BL12-10)".

Moreover, to reconstruct the recent flood events occurred in the studied area, a short sediment core (BL12-10, 40 cm length; Latitude: 33°14'58.7"; Longitude: 11°10'3.7" Fig.3) was recovered from the El Bibane Lagoon (EBL) by a hand corer 75mm diameter PVC tube.

Page 9 Line 15

The sentence "Surface sediment samples have been collected from three different types of location"is removed.

Page 9 Line 16

We agree with the reviewer and the sentence is added:

"of the surface sediment samples".

Page 9 Line 17

 "depending on the environmental setting" is added in the text as asked

Page 10 Line 3

"fluvial" is removed.

Page 10 Line 6

"even" is removed

Page 10 Line 14

We agree with the reviewer.

"Our" is replaces by "the"

Page 12 line 14

We agree with the reviewer and we take into account his suggestion. A paragraph describing the geochemical results was added:

"Our results display that these three mud layers preserved in the core are also characterized by high Fe/Ca and Ti/Ca elemental ratios (Fig. 12)".

Page 12 line 18

We thank the reviewer for his remark.

In this study, we undertake the calibration of the sedimentary archive and the hydrological data during the last century. The historical hydrological data begun in the early 1930, that's why we select the sedimentological and the geochemical analysis up to 30cm depth of the core which cover this time period.

The whole BL12-10 core length is 90 cm. Results of this total 90cm length core will be reported in a next paper (Affouri et al., in prep.) discussing the climatic change during the warm periods (Antiquity and Medieval) and the cold periods (LIA) and its impact on the flood variability of the Fessi River.

Page 16 Line 25

The flood events recorded in 1969 and 1979 in the sedimentary archive correspond to 14 and 17 cm core depth. The down core distribution of the $^{210}$Pb does not show a clear split in the profile indicating that no significant erosion happened in the lagoon at the location of the core during the studied period.

Page 17 Line 4-5

We agree with the reviewer and we take into account his suggestion

The term bioturbation is removed. The composition of the sedimentary archive is homogeneous and linked to bottom currents. These sediments are suspended and re-deposited.

The deposits thickness associated with these flood events is low (<5cm).

Page 18-19

We thank the reviewer for his remark and we take into account his suggestion. Text have been changed asked.

"These results indicate that finer material, high content of mud (clay+silt), as well as high ratios of Fe/Ca and Ti/Ca are associated to flood events in the lagoonal sequence. The association of these proxies in the sedimentary sequence of the El Bibane lagoon can therefore be used to reconstruct flood activities in Southeastern Tunisia during the upper Holocene".

**Responses to Reviewer Comments**

We thank the reviewer for his thoughtful comments and suggestions.

**Report #2**

1) Additionally, the semi-quantitative Niton XRF data might be not really robust for pure quantitative comparisons. However, the authors must show at least the same XRF proxies for the catchment, PCA and core stratigraphy.

Indeed, the XRF scanner is a semi-quantitative measurement. Geochemical data are expressed in ppm or percentage values. Analytical error on replicate standard are lower than 5% for Ti, Cr, Fe, Zn, Pb, between 5 and 15% for Ca, Mn, As, Rb, Sr, and between 15 and 25% for K and Co. (The technique is described in the chapter materials and methods).

However, variability of Fe/Ca and Ti/Ca ratios in surface sediments and along the lagoonal core are higher than analytical errors indicating that this semi-quantitative approach is sufficient to mark the past floods of the Fessi River.

**Abstract**

Page 1 Line 19: It is mention in the abstract that surface samples in the lagoon's catchment trace the origin of lagoon deposits. Do the data really provide this information? High Fe/Ca and Ti/Ca and high clay and silt contents indicate paleofloods in the lagoon but is there a coupling with the surface samples in the catchment?

Our data suggest that the surface samples characteristics of the catchment can be used to identify the origin of the lagoon deposits. The aim of our study is to identify and to calibrate the proxies which can be used to reconstruct the paleofloods archived in the sediments of El Bibane lagoon.

In this context, geochemical data of major and trace elements have been used to display their distribution on the whole studied area, from the upper watershed to the coast. To a first approximation, this approach allowed us to distinguish between the different sources of sediments (Aeolian, marine and fluvial). However, it is difficult to make palaeoenvironmental reconstitutions using only geochemical elements because of the dilution processes which can strongly affect the geochemical profiles. That's why we used elemental ratio to overwhelm the dilution effect. Consequently, elemental ratios such as Fe/Ca and Ti/Ca, (Fig. 11) are proposed in this study to identify the modern sediment sources in the studied area and to reconstruct past changes in the sources of the lagoon sediment. Figure 11 have permitted to identify the different sediment sources.

For instance, it displays that the fluvial end member is characterized Fe/Ca> 0.08 and Ti/Ca> 0.02. When applying this proxy along the core BL12-10, we can see that the three mud layers which were interpreted as flood events present Fe/Ca values close to 0.08 and Ti/Ca close to 0.02 (fig. 12), close to the present day fluvial end member.

Page 2 Lines 1ff The authors mention historical flood events. For me, a historical flood event is a short event that is not comparable with an enhanced annual mean in precipitation. The authors do not show these event data but only annual means in precipitation.

We agree with the reviewer.

The southern of Tunisian suffered a real catastrophic event. The mean annual rainfall is about 200 mm in the southern Tunisia but precipitation is very irregular with episodic catastrophic flooding followed by long periods of drought which may extend over several years (Medhioub et al, 1981). Flood events were linked to heavy precipitation events which were very close in time. For example: the flood of 1969 (24 to 48 hours; Pias, 1970; kallel.R and Colombani.J, 1972), the flood of 1979 in Medenine (4 days; Bonvallot, 1979), the flood of 1995 in Tataouine (11 to 24 hours; Boujarra and Kttita; 2009, Fehri, 2014) and the flood of 1932 (A heavy rainfall has been recorded in Medenine region; 449 mm during few days; Bonvallot, 1979).

**Manuscript**

Page 2 Line 13: I am not sure about this statement. There were already a network of gauges and meteorological stations in Mediterranean North Africa during the colonial period. However, the easiest option to support your statement might be significant reference.

We thank the reviewer for his comment. In fact, no reference is available to support our statement, but the official web page of the Tunisian National Institute of Meteorology indicates that the establishment of the first synoptic observation network by FMO (French Meteorological office) was in 1926. For Southern Tunisia, available data start in the early 1930.

Page 3 Line 3ff. The knowledge about centennial-scale Holocene flood phases in Tunisia is by far better than mentioned in the introduction. Please consider Faust et al 2004 QSR, Zielhofer and Faust 2008 QSR and references therein. The approach of cumulative probability plots that is applied in Zielhofer and Faust 2008 does not need continuous archives. Hence, I cannot follow your argumentation at this point.

We thank the reviewer for his remark. We specified in the text that our statement about the poor knowledge of centennial-scale Holocene flood phases concerns only Southern Tunisia.

Page 3 Line 16 to Page 4 line 12. This paragraph is somewhat complicate to read. It might be better to put these statements into a figure or a table.

This paragraph describes the geology setting of the studied area and these statements were placed on the figure (Please see Fig. 1) in order to become easy to reading.

Page 5 Line 15ff You cannot deduce short-term flood events from mean annual data. Maybe, you just exclude the term "flood events" in the manuscript and emphasise peaks in mean annual precipitations.

We agree the reviewer comment.  The higher annual data recorded in 1969 was associated with heavy precipitation events which were close in time (September and October).

Page 7 Line 14ff Please mention only parameters that are shown in the result chapter as well.

Granulometry parameters that are not used in this work have been removed from the text, as asked.

**Chapter 5.1.1.**

Please do not use references in the result chapter.

All references of the result chapter were removed.

Why do you present the data of the dunes? Do you have dunes in the lagoon record? It might be better to emphasise the S9 and S10 fluvial reference samples.

The dunes represent the main source of aeolian material. We have decided to sample the dunes in the study area in order to characterize the aeolian supply to the lagoon.

Chapters 5.1.2 and 5.1.3 are not convincing to me. Within the lagoon core you point to Fe/Ca and Ti/Ca ratios as proxies for alluvial fills. Please use the same proxies (ratios) for the catchment analysis.

The elemental ratios Fe/Ca and Ti/Ca are also analyzed for both surface samples (Fig. 11) and along the BL12-10 core (Fig. 12).

Chapter 6.2 It would be much better to discuss element ratios than simple elements.

We thank the reviewer for his remark.

In this section we choose to discuss simple elements by statistical analyses (PCA) with the aim to separate simple elements in link with their origin (Marine, fluvial and Aeolian end members). The statistical analysis of the some elemental ratios (Fe/Ca and Ti/Ca) demonstrated that they can be used to identify the principal sediment of the El Bibane lagoon.  By so doing, we avoid problems linked to possible dissolutions in the past.

We thank the reviewer for his thoughtful comments and suggestions.

**Report #3**

The revised manuscript by Affouri and co-authors describes a way to reconstruct past flood events from the El Bibane lagoon from Southeast Tunisia. The sedimentary records from lagoons are definitely an underexplored archive for the reconstruction of flood events and therefore this manuscript could be of wider interest for the paleo-flood community. Affouri et al. applies classical methods like grain size analysis and elemental fingerprinting of different potential source lithologies in order to identify the flood layer deposits in the El Bibane lagoon. Although the conclusions draw from the analysis are at first sight convincing, I'm puzzled by the fact that the high-energy flood deposits are actually smaller in grain size than the 'normal' sedimentation in this lagoonal setting. Typically, lagoons are associated with low energy depositional areas and therefore I would expect a courser grain size for the flood deposits compared to the marine/lagoonal sediments. Unfortunately, it is not clear from the manuscript what are the expected depositional processes at the coring site.

Here, the past flood activity was investigated using a multi-proxy approach combining sedimentological and geochemical analysis of surfaces sediments from the Southeast of Tunisia catchment in order to trace the origin of sediment deposits in the El Bibane lagoon. Three sediment sources were identified: marine, aeolian, and fluvial. The grain size analysis showed that the fluvial source has a bi to multimodal distribution with two or even three modes. In order to obtain the best resolution in the identification of the fluvial source, we choose to use the sediment samples which were collected only along the River Fessi: S9, S10, S12 and S13. These surface sediment samples show a decrease in the mean grain size from the upstream to the downstream of the River Fessi watershed. The decrease in the mean grain size could be explained by a strong change of the topographic slope around Tataouine region. Here, the coarser material is deposited and the finer material is transported away by the river. These finer sediments are deposited in the low plain of the river and in the El Bibane lagoon. Therefore, we suggest that S9 and S10 (collected between Tataouine and the lagoon) characterize the fluvial component in the lagoon. The grain size distribution for S9 is unimodal with a mean grain size around 96 μm indicating moderately sorted muddy sand. The corresponding size range is very coarse silty/very fine sand. Sample S10 is fine silt with trimodal distribution in 7μm, 26μm and 73μm, and poorly sorted mud sediment type. These characteristics will serve to identify the fluvial source into the lagoon.

For this reason, I would suggest to clarify the following three aspects for a convincing flood deposit identification:

1. Definition of the marine end member sediment source
The authors want to define the marine sediment source by sampling beach sands, but I would not call these sediments as the marine source for the lagoon. These are shoreline deposits and should be labeled as such. I would relate the marine sediment source of the lagoon to particles that are produced within the water column (which are rather small in grain size). Is there anything known about this marine sediment source? What about the productivity within the lagoon and the deposition of organic matter?

We agree with the reviewer.

Numerous studies have been carried out on the coastal environments. These environments correspond to the watershed, the lagoonal and the marine area near the sea. Therefore, we preferred to name these samples as marine sediments because they were coherent with other studies dealing with coastal environments such as Raji et al. (2014) and Dezileau et al. (2016). Moreover, we demonstrated clearly in this work that the fine fraction found in the lagoon sediments is associated with periods of higher discharge of Fessi River (flood events).

The productivity within the lagoon has been treated in this study on the distribution of the mollusk species (Affouri et al; in prep) and according the previous studies (Medhioub et al, 1977, Zouali , 1982, Capapé et al, 2004 and jouili el 2016), the El Bibane lagoon was considered to be among the most biologically diverse and productive ecosystems and a nursery for several species.

The organic matter deposition in the lagoon has not been studied in this work. Therefore Medhioub et al, 1981; show in his study on the El Bibane lagoon, an anomalous repartition of the organic matter in the sediment. This anomalous could be explained by a local mortality of marine organisms related to the junction of the inflow and the more saline and confined waters of the lagoon.

2. How were the lagoonal samples collected? Related to the first aspect above is the question about the definition of the lagoonal sediment. The authors need to explain the method used for collecting the lagoonal sediments (grab sampler, coring,..?) and which sediment depth interval is covered in this surface samples? If only the topmost cm was sampled then this could be related to the marine end member source. In addition, the authors should show the grain size distribution of S6 in Fig. 5 as this surface sample is closest to the coring site and therefore the most relevant one of the 3 lagoonal samples.

The first three centimeters of the core were used to characterize the surface samples (Cf line 9-11, page 6). The studied core BL12-10 was recovered in the southern part of the lagoon, far from the connection with the Sea. Thus it cannot used to characterize the marine source.

We agree with the reviewer that more surface sediment covering the whole area of the lagoon are needed to better characterize the modern influence of the different sources.

3. Shell fragments in the flood deposits
According to the Figures 9 and 11, the finer flood deposits FL1 and FL2 contain shells or maybe shell fragments. How is it possible that a fluvial deposit, which is likely deposited within hours or days can contain – I assume – marine shells? Would this not contradict with the interpretation of fluvial deposit?

Internal circulation within the lagoon can cause a re-suspension of the material. Biorturbation can also play a role which may explain the occurrence of these shells in the flood deposits. In any case, the occurrence of these fragments are negligible in these flood deposits.

Overall:
The manuscript need major clarification concerning the flood layer recognition. The number of

figures needs to be consolidated and the quality of the figures should partly be improved. Furthermore, the manuscript needs careful evaluation for consistency (see comments below) and some language polishing.

Specific comments and technical corrections (changes are marked within 'apostrophes')

For the entire manuscript: Consistent use of abbreviation throughout the manuscript

Write the name of the study object in capital letters: 'El Bibane Lagoon'. Occasionally it is abbreviated (->EBL), but not consistently. I would advise not to do it. Check throughout the manuscript.

We thank the reviewer for his remark and we take account his suggestion.

The same is with the term Principal component Analysis (PCA) on Page 9, Line 2, as well as in other locations within the manuscript.
Done
Page 2, Line 16: …floods 'in' the future.

Done

Page 2, Line 22: … where overwash deposits 'are' preserved within

Done

Page 3, Line 5: In 'this' study',' we tried…

Done

Page 3, Line 6: … 'the semi-arid environment of Southern Tunisia' (delete the rest of this sentence)

Done

Page 3, Line 9: …lagoonal sequence in Southern Tunisia (delete 'the')

This sentence is removed

Page 3, Line 13: …, we 'undertook a' calibration

Done

Page 3, Line 16: Morphological, (delete the) 'S'outhern Tunisia '-' known as the Tunisian platform '-' includes…

Done

Page 4, Line 2: The Cretaceous series 'represents' a general 'succession' from neritic,…

Done

Page 4, Line 15: It has 'a maximum water depth of' 6m in the…

Done

Page 5, Line 10: …30 days/''years (remove space)

Done

Page 5, Line 11: … October to Mar'ch'…

Page 5, Line 23: I would like to see a table with all the coordinates of the surface samples. This was the case for the original submitted manuscript, but such a table was then removed due to a comment of reviewer 2, although reviewer 2 only requested to move this table to the supplement. This sentence here ("The locations of all…") could then be moved to the description of this table.

A table of the coordinates of the surface samples is added and moved in the supplementary data

Page 5, Line 24: Sentence: "Sediments were returned to the laboratory for analysis." is not needed.

Done

Page 6, Line 15: …split, photographed, 'and' logged…

Done

Page 6, Line 17: ….with a 'hand-held' Niton….

Done

Page 6, Line 19: 4'µ'm (micrometer, not millimeter)

Done

Page 7, Line 1: remove space between Beckmann-''Coulter
Done
Page 7, Line 18: …calculated arithmetically','  (delete: and) geometrically (in microns)',' and logarithmically….
Done
Page 8, Line 2: 210Pbex: ex should be subscripted through the manuscript; check also the figures.
We thank the reviewer for his comment.
Page 9, Line 4: remove the word chemical

Done

Page 9, Line 15: …types of location's'

This sentence "Surface sediment samples have been collected from three different types of location "was removed

Page 9, Line 17: Use the same order to mention the different sources (i.e Aeolien, Marine and Fluvial) here as they are discussed subsequently (Pages 9 and 10)!

Done

Page 9, Line 18 …encompass'es' sediment samples…

Done

Page 10, Lines 5-8: The authors claim that the fluvial sediments have a bi- to multimodal distribution, but the samples S9 and S12 (Fig. 5) are unimodal. It is mentioned further down, but this section needs to be revised!
Done
Page 10, Line 12: transported 'further' by the river.

Done

Page 12, Lines 16-17: Sentence can be removed. Redundant information.

Page 13, Line 12: ..shows 'a maximum' at…

Done
Page 14, Line 7: Rephrase to: 'Furthermore, the lagoon samples showed a higher variability in the grain size due to the presence of shell fragments.'

Done

Page 15, Line 25: Identification of flood 'layers' in the…
Done
Page 17, Line 4: The authors state that the sedimentation rate of these flood events is not high, but by definition these event layers are deposited within hours to day. These event deposits need to have a high sedimentation rate! Clarify.

We agree with the reviewer comment. The deposits thickness associated to these flood events is low important (<5cm).

Fig. 2: Could be combined easily with Fig. 12!

We thank the reviewer for his comment.

Fig. 3: It is confusing that the water bodies are colored blue in the upper figure, whereas the landmass is colored blue in the lower figure. Please correct!

Done

Fig. 4, Line 3: S3 (and not S1) is shown in the figure

Done

Fig. 4, Line 4: According to the image of S17 1cm can NOT be the diameter of the photos!
Done
Fig. 8, Line 5: …and 'a'eolian sand dune.

Done

Figs. 9 and 12: Combine these to figures as they have a very similar content. Show the grain size date in the full resolution (as in the Fig 9 now). And the blue bars for the flood deposits are not consistent between Fig 9 and Fig. 12. Revise!

We agree with the reviewer. The grain size of the BL12-10 was performed at high resolution each 1 cm but the geochemical analysis were each 2 cm. When we correlated the two proxies we choose to place the grain size results each 2cm which match the geochemical analysis.

Table 3 continued: Check the values for MODE 1; it is highly unlikely that all values are identical for mode 1 (i.e. 106.0)! Please correct!

We thank the reviewer for his remark. Indeed, GRADISTAT program version 4.0 (Blott, 2000) was used for grain size statistical analysis. The following sample statistics are calculated using the Method of Moments in Microsoft Visual Basic programming language: mean, mode(s), sorting (standard deviation), skewness and kurtosis. Grain size parameters are calculated arithmetically, geometrically (in microns) and logarithmically (using the phi scale). The results of the grain statistical analysis (Table 3 continued) were determined automatically using this grain size program. All of the samples, except one, have Mode 1 at 106μm. These samples are different in their Mean values.

---

## Author Response (AR3)

**Cover Letter**

**Dear Editor,**

First of all on behalf of all co-authors, I would like to thank the reviewers and the editor for their helpful comments and suggestions. I believe that we have successfully integrated all the reviewers' comments and requests, and that these changes have greatly improved the manuscript. Please find the revised version of our manuscript CP-2016-40 entitled: **"Extreme floods events reconstruction spanning the last century in the El Bibane lagoon (Southeast of Tunisia): A Multi-proxy Approach" by A. Affouri, L. Dezileau and N. Kallel**.

In this revised MS the text has been modified taking into account the reviewers and the editor comments. Please find in this MS three text colors as follows: The pine text color reflects the modifications as suggested in the first revised version (RC1, RC2, RC3 and RC4); the blue text color reflects the second revisions to the reviewers (RC1, RC2 and RC3) and finally the red text color reproduces the MS modifications as suggested by the editor comments. In addition, figures qualities had been enhanced and some figures were modified as recommended by the reviewers.

I look forward to hearing from you and please do not hesitate to contact me for any further requests.

Sincerely yours,

Aida Affouri

**Response to the Editor comments**

*We thank the Editor for his thoughtful comments and suggestions. We have included almost all of the raised suggestions and below we present a point-by-point response to the comments.*

1-      The reviewers underlined problems in the structure of your paper. They asked you to modify the organization by moving some paragraphs. Corrections are not only remove and added some rare sentences or correct some typographic or language errors. Especially as pointed by the reviewers, some paragraphs are not related to the discussion but to the results. So please, move all descriptions of data in results. For example, paragraphs 6.1, 6.2 and probably part of 6.3 are related to results and not to discussion. This will helps to clarify your manuscript.

We thank the reviewer for his comment and we take into account his suggestion. The structure of the manuscript is changed.

**5. Results**

**5.1. Surface sediments**

**5.1.1. Sediment description:  grain size and morphology**

**5.1.2. Distribution of major and trace elements**

**5.1.3. Principal component analysis (PCA)**

**5.1.4. El Bibane lagoon: Main sediment sources**

**5.2 Core BL12-10**

**5.2.1. $^{210}$Pb and $^{137}$Cs dating**

**5.2.2 Sedimentary and geochemistry**

**6. Discussion**

**6.1 Paleoflood reconstructions**

**6.2. The El Bibane lagoon: A key region for paleohydrological reconstructions**

2-      As underlined by report 1, you have to present in the methods all types of data you use on the core and do not present data that will not be described in the results and used in the discussion. Ex: XRF data.

We agree with the report 1.  A paragraph was added in the text with corresponding data reported on figure 11:

"Down-core profiles of heavy and light elements through the depth also delineate the different units distinguished by sedimentological analysis. Based on their profiles, the first group composed by Fe, Ti, K and Al exhibit similar variations, concentration values are mainly high in fine-grained intervals and are low in coarse-grained intervals. These high values are probably due to high inputs from the Fessi River. The Si and Zr which characterized the second group display a different behaviour than the first group. These two elements are high in the fine sandy intervals. This probably suggests that their highest values are related either to aeolian inputs in the lagoon. The Ca and Sr characterised the third group show a reverse distribution pattern by comparison to the first group with higher values in the coarse grained intervals and lower values in the fine grained intervals. (Fig.11). In order to obtain the best resolution in the geochemical analysis, a similar elemental ratios used in the surface sediments were applied on the core to identify the geochemical signatures of the different sedimentological events. Our data displays that the elemental ratios Fe/Ca and Ti/Ca show the most down core significant variations and that the three mud layers (high content of clay+silt) preserved in the core are characterized by high Fe/Ca and Ti/Ca ratios (Fig.12a)".

3-     In the response to reviewer 3 you add some information on the core but you did not write anything in the text. For example, explanations on the upper centimetres or on the grain size may be helpful. Why didn't you add that in the text?

We thank the editor for his remark and **requested** information was added in the text:

    "The S6 representing the first three centimeters of a lagoon sediment core BL12-10 was used to characterize the surface sediments samples".
Moreover, to reconstruct recent flood events occurred in the studied area, a short sediment core (BL12-10, 40 cm length; Latitude: 33°14'58.7"; Longitude: 11°10'3.7" Fig.3) was recovered from the El Bibane Lagoon (EBL) by a hand corer 75mm diameter PVC tube in the southern part of the lagoon, at 35 km from the Fessi River delta and 14 Km for the connection with the sea (Please see: 4. **Materials and Methods; 4.1  Materials).**

    The whole BL12-10 core length is 90 cm but in this study we used only the first 30 cm (Please see **5.2.2 Sedimentary and geochemistry).**

4-     The true discussion is definitively too short and still more after moving what is relevant to results. This has been underlined by reviewers in the two rounds of reviews and I agree with that. You have to develop and debate on what your work is interesting and brings concerning the flood reconstructions comparing to previous works and not only in the studied area but in a larger context in Tunisia and in Mediterranean. In addition, it will be useful and

essential to develop in the discussion how your work helps to understand local and regional climate changes, especially on what concerns extreme events, through such floods analyses. We thank the editor for comment and we take into account his suggestion. A paragraph is added in the text:

"**6.2. The El Bibane lagoon: A key region for paleohydrological reconstructions**

According to many studies on lagoons, these environments are good study areas to record past climatic and environmental changes, and extreme sea events. These fields of research were successfully applied in the western North Atlantic (Donnelly and Woodruff, 2007), Northwest Florida (Liu and Fearn, 2000; Lane et al., 2011; Das et al., 2013), the Northeastern United States (Parris et al., 2010), the Central Pacific (Toomey et al., 2013), Southern Japan (Woodruff et al., 2009), Western Australia (Nott, 2011), Northeastern New Zealand (Page et al., 2010), Northern Europe (Sorrel et al., 2012), or the Western Mediterranean (Dezileau et al., 2011, 2016; Sabatier et al., 2012; Raji et al., 2015; Degeai et al., 2015). Despite, the importance of these topics, such studies are still scarce in southern Tunisia. The El Bibane lagoon is different from the other studied lagoons because it cannot record coastal overwash events. Such particularity is linked to the morphology of barriers that separate this lagoon from the open sea. These barriers consist of two narrow fossil carbonate consolidated peninsula formed during the last interglacial period and reaching 10 m elevation (Medhioub et al; 1997 and Jedoui; 2000). Thus they are not recent sand bars or spits as the geographical map could suggest and they cannot not be over-washed during extreme sea events. However, we have demonstrated from this study that this lagoon could record past flood events during exceptional heavy precipitation episodes that punctuated the recent meteorological and climatic history of Tunisia and North Africa. Tramblay et al., (2013) have analysed the influence of large-scale atmospheric circulation, including the North Atlantic Oscillation (NAO), Mediterranean Oscillation (MO), El Nino-Southern Oscillation (ENSO) and Western Mediterranean Oscillation (WEMO) on precipitations and extreme events in 22 stations located in Algeria, Morocco and Tunisia for the last 50 years. Although some spatial patterns for the different precipitation indices have been identified over Maghreb countries the southern part of Tunisia was only represented by one meteorological station (Gabes). This clearly avoid to identify an homogeneous climatic region, there is a need to include more stations with longer record length. El Bibane lagoon paleoflood record can be of great importance to better understand the physical mechanism responsible for the changes in the frequency and/or the intensity of extreme events in the southern part of Tunisia. It will be

interesting to study the natural variability of past flood events in this semi-arid environment through contrasting climatic periods (cold and warm periods). Further coming investigations on long core sediments could clarify the relationship between large-scale atmospheric circulation reconstructions and the major flood periods (Affouri et al., in prep). Additionally, such studies could be a crucial tool to evaluate the role of Mediterranean paleo-climate on the development and growth of human society".

5-      One of the reviewers ask you to gather the figures 9 and 12. I agree with that. Why did not you do that? You write you agree and you did not change that!!!

These two figures were combined as one. (Please see Figure 12)

6-      It is not acceptable to produce figures with a so bad resolution as what is offered in the corrected manuscript. Please take care about what you sent. Furthermore, the characters in many of them are too little and often unreadable. Some figures are blurred with unreadable legends. Please enlarge the characters in the maps and correct the readability and sharpness of all figures.

We thank the editor for his remark. All the figures had been improved with high resolution.

---

## Author Response (AR4)

**Dear Editor**

First of all on behalf of all co-authors, I would like to thank the editor for his helpful comments and suggestions. I believe that we have successfully integrated all the editor comments and requests, and that these changes have greatly improved the manuscript. Please find the revised version of our manuscript CP-2016-40 entitled: **"Extreme floods events reconstruction spanning the last century in the El Bibane lagoon (Southeast of Tunisia): A Multi-proxy Approach" by A. Affouri, L. Dezileau and N. Kallel**.

In this revised MS the text has been modified taking into account the editor comments. Please find in this MS one text color. The green text color reflects the modifications as suggested by the editor comments. In addition, figures qualities had been enhanced and some figures were modified as recommended by the reviewers.

I look forward to hearing from you and please do not hesitate to contact me for any further requests.

Sincerely yours,

Aida Affouri

**Response to the Editor**

*We thank the Editor for his thoughtful comments and suggestions. We have included almost all of the raised suggestions and below we present a point-by-point response to the comments.*

In acknowledgements, please do not forget to thank the reviewers. They took time to read your paper and provide comments. Thank you.

We thank the editor for his remark. An acknowledgement to the editor and the reviewers was added.

"We are particularly grateful for editor comments of Dr Nathalie Combourieu-Nebout. We thank Maria-Angella Bassetti, MCF-HDR Assistant Professor at the university of Perpignan, France and anonymous reviewers for their helpful comments and their criticism, which led to a considerable improvement of the manuscript".

1- The text will benefit by a reading by an English native could be done before acceptance. I propose some primary corrections.

Line 50: please verify Raji et al, 2014. It does not exist in the reference list

We agree with the editor. We correct this reference as follows:

"Raji et al., 2015"

Line 69: add and before "is constituted"

We have placed "and" before "is constituted".

Line 70: there is no Figure 1 B

We agree with the editor.

Fig.1B is replaced by Fig.1

Line 71: replace as following by as follows

We replace "following" by "as follows"

Line 137: replace for by from "the connection..."

We replace "for" by "from"

Line 154: add "on" before "the BL12-10..." and correct the sentence "Each sample....". Perhaps it will be better to write something like "sieved through a 1cm mesh" at the end.

We thank the editor for his remark and we take into account his suggestion.

We added "on" before "the BL12-10…"

This sentence is modified as follows:

"Each sample was primary sieved through a 1mm mesh,"….

Line 213: delete these before "three groups" and add an "s" a to "follow"

We have deleted "these" and we added an "s" as to follow.

Line 262: write "The same distribution"

We write "The same distribution".

Line 285-287: please write a single sentence

We agree with the editor and we take into account his suggestion. This sentence is modified as follows:

"It is characterized by high positive loadings for Fe, Ti, K, and Al which indicates the dominance of alumino-silicates minerals in surface sediments (Spagnoli et al., 2008; Plewa et al., 2012)."

Line 287: delete thus as the prevalence of these elements is not a consequence of what you say in the previous sentence.

We thank the editor for his comment and we delete "thus".

Lines 292 to 294: Rephrase from "silicon... to 1989)" it is not clear. You explain that you may have two origins first and choose the second why?

We agree with the editor and the sentence is rephrased as follows:

"Zr and Si are associated to silicates originating either from adjacent desert areas by erosion or from western Saharan dunes by storms."

Line 297: write Ca is high in the marine samples

We thank the editor for his comment and we take into account his suggestion. The sentence is modified as follows:

"Ca is high in the marine samples".

Line 298-299: "both.... Material" and (not but)"also."

We replaced "not but" by "and also".

At the end of the 5.1.3 we have no real conclusion after the last sentence we need more explanations!! What was the real input of this component analysis? It is not clear. And we need a sentence to link with the following paragraph or at least an opening to the paleorecord. In which way does such study will serve you for the paleoreconstruction?

We agree with the editor comment and we take into account his suggestion. A sentence was added in the end of this paragraph:

"This method allowed us to label elements of terrigeneous source (Fe, Ti, K and Al) from those from in situ marine origin (Ca and Sr). These proxies will be used to reconstruct past flood and storm events with the help of sedimentary archives."

Line 324: delete "we have seen that" it is not necessary for me

We thank the editor for his remark and we take into account his suggestion.

"We have seen that" is deleted from the sentence.

Line 330: replace the ":" by /

":" is replaced by "/"

Line 341: rephrase the sentence. You can write "this study proposed the preliminary analyses performed on the first 30cm only although "the whole..... cm"

This sentence is rephrased as follow:

"This study proposes the preliminary analyses performed on the first 30 cm only although the whole BL12-10 core length is 90 cm."

Line 342: replace made by composed

We replace "made" by" composed"

Line 351: where are represented the profiles cited in the sentence? There are not values in the Table 5. I suppose it is in figure 11 but the citation is nine lines below!!

We agree with the editor and Fig.11 is added in this paragraph.

It's ok in the table 5 we have not values because it shows the statistical analysis of grain size of BL12-10 core samples.

Line359: "a similar ratio (without s) .... is applied" or "the similar ... ratios ....were"

We thank the editor for his remark and change this sentence as follows:

"Single element concentrations may be sensitive to dilution effects to allow reliable reconstructions of terrestrial climate, elemental ratios often better reflect the origin of the sedimentary material".

Line 361-363; is it necessary to write "our data" you may begin directly with the measured elemental....ratios...show ....and the mud.....12a)

We agree with the editor. This sentence is modified as follow:

"The measured elemental ratios Fe/Ca and Ti/Ca will be used to reconstruct pas flood events (Fig. 9). A higher Fe/Ca and Ti/Ca ratio in the lagoon sediments would be a signal of more sediment contribution from the Fessi River during flooding."

Line 371 to 374: could you please link the two sentences as one is the conclusion of the other.it is what I have understand. Or link the second to the third, it will be easier to follow.

We thank the editor for his remark and we take into account his suggestion. This sentence was rephrased in the text as

"The combination of geochemical and grain size data suggest that the BL12-10 core deposits had registered three flood events namely FL1, FL2 and FL3 (Fig. 12). These flood deposits have a thickness of 5cm, 4cm and 2.5cm respectively. "

Line 381: do not mark an automatic wrap return and continue the previous paragraph. Or you have to do the same thing at line 387 and 407. Perhaps the best thing will be to have a unique paragraph on the flood chronology. Without, it is not so long to read.

We agree with the editor. We have change as one paragraph.

Line 394-398: rephrase the sentence and simplify the text.

We thank the editor for his remark and we take into account his suggestion. This sentence is rephrased as follow:

"The activity of $^{210}$Pb in this flood deposit is not disturbed; it is homogeneous (Fig. 10). For this reason we assume that no significant erosion happened in the lagoon during this period. During these heavy precipitation events, most of the sedimentary material was deposited in the floodplain, in the lagoon and probably transported to the Mediterranean Sea through the passes."

Line 401-405: it will be better to link the sentences as at that time the facts shown appear not complementary to end in your conclusion.

We thank the editor for his comment and we change this paragraph as follows:

"The thickness of the sediment layer associated with these flood events is low, i.e. about 5 cm. The grain size and geochemical values of this flood deposit are rather homogeneous. This homogeneity is probably linked to the action of weak bottom currents within the El Bibane lagoon. Finally, since these ...archive".

Line 410: write "El Bibane flood record shows"... instead of "the results"

We agree with the editor and we modify this sentence:

"El Bibane flood record shows .........events".

Line 418: add at the beginning "moreover these data demonstrate that...". And in the next sentence "Thus the association...."

We thank the editor for his comment and we take into account his suggestion. This paragraph is rephrased as follows:

"Moreover these data demonstrate that finer material with a high content of mud (clay+silt), and high ratios of Fe/Ca and Ti/Ca are associated to flood events in the lagoonal sequence. The association of these proxies in the sedimentary sequence of the El Bibane lagoon can therefore be used to reconstruct flood activities in Southeastern Tunisia."

Line 425: I suggest "Lagoon records shows that such costal environments..." and place "despite the importance of these topics in Mediterranean coastal areas" at the end if you want.

We thank the editor for his remark and we take into account his suggestion:

"Lagoon records shows that such costal environments are good study areas to record past climatic and environmental changes, and extreme sea events."

Line 433: delete "despite ...topics" and begin you sentence with "such..."

We thank the editor for his comment and we take into account his suggestion. This sentence is modified as follows:

"Such studies are still scarce in southern Tunisia, despite the importance of these topics in Mediterranean coastal areas."

Line 438: verify the references. There are no Medhiou et al 1997 and medhiou and Jedoui, 2000.

We agree with the editor we correct these two references in the text.

"Medhioub, 1979; Jedoui, 2000"

Line 456: I think that it will be better to mark "Affouri et al., data in progress" instead of "in prep"

We thank the editor for his remark and we take into account his suggestion.

"Affouri et al., data in progress".

Sometimes the citations of legend are Fig. sometimes fig., please use the CP.

Done

2-In the figures.

I saw that the quality is considerably improved than in you previous versions. Nevertheless it remains that you have to take into account that the figures will not be published al in full page and will be largely reduced in the published document produced after acceptation.

So it is very important to enlarge the characters in figures to be sure that they will be readable after reduction. It is particularly important in the maps where the names of the localities sometimes cited in the text are definitively too little. See figure 1, 2, 3 and 7.

We agree with the editor and we have enlarged the characters to the names of the localities for these figures

• On Fig. 3 and 5 what did you represent by the numbers around the map (570900 to 72000 and 3640000 to 3740000). I think that they represent the latitudes and longitudes. Please write them in a classic format, I never saw such format in papers. Take care about the all numbers and names if the figure is in reduced format

We thank the editor for his remark and change all these figures in a classic format.

• Figure 6 is was not labelled in your document

We labelled in the document Figure 6.

• Figure 7: that one could be probably one conserved in full page, but enlarge text and the legends too, you have the place to do that on each small maps. There no indications of the number of the samples on the maps on this figure and you only put the location of five samples on Fig.4. so we have not the link with the Table 3.

We thank the editor for his comment.

Figure 7 is modified with high resolution. All the indication of the number and the location of the samples studied are putted in figure 3: S1 to S18. In figure 4 we just put only the samples studied by binocular observation.

The table 3 show the XRF analysis results of the major and trace element in studied samples (S1 to S18) in ppm (parts per million) and in percentage (%). Indeed, we have use three intervals for each elemental analysis: lower values by yellow colour, intermediate values by orange colour and red colour for higher values.

• Fig. 9: Names are still fuzzy

We modify this figure.

• There was not the name on the fig 12. On it, you have some indications in landscape and some indications in portrait. Could you please put the legend of the stratigraphy near the log and all the names. Same remark for the fig 11

We thank the editor for his comment and we have putted the legend of the stratigraphy near the log.